# Do we need rebalancing strategies?
# A theoretical and empirical study around SMOTE and its variants

## Abstract

*Synthetic Minority Oversampling Technique* (SMOTE) is a common rebalancing strategy for handling imbalanced tabular data sets. However, few works analyze SMOTE theoretically. In this paper, we prove that SMOTE (with default parameter) tends to copies the original minority samples asymptotically. We also prove that SMOTE exhibits boundary artifacts, thus justifying existing SMOTE variants. Then we introduce two new SMOTE-related strategies, and compare them with state-of-the-art rebalancing procedures. Surprisingly, for most data sets, we observe that applying no rebalancing strategy is competitive in terms of predictive performances, with tuned random forests, logistic regression or LightGBM. For highly imbalanced data sets, our new methods, named CV-SMOTE and Multivariate Gaussian SMOTE, are competitive. Besides, our analysis sheds some lights on the behavior of common rebalancing strategies, when used in conjunction with random forests.

## 1 Introduction

Imbalanced data sets for binary classification are encountered in various fields such as fraud detection (Hassan & Abraham, 2016), medical diagnosis (Khalilia et al., 2011) and even churn detection (Nguyen & Duong, 2021). In our study, we focus on imbalanced data in the context of binary classification on tabular data, for which most machine learning algorithms have a tendency to predict the majority class. This leads to biased predictions, so that several rebalancing strategies have been developed in order to handle this issue, as explained by Krawczyk (2016) and Ramyachitra & Manikandan (2014). These procedures can be divided into two categories: model-level and data-level.

Model-level approaches modify existing classifiers in order to prevent predicting only the majority class. Among such techniques, Class-Weight (CW) works by assigning higher weights to minority samples. Another related proposed by Zhu et al. (2018) assigns data-driven weights to each tree of a random forest, in order to improve aggregated metrics such as F1 score or ROC AUC. Another model-level technique is to modify the loss function of the classifier. For instance, Cao et al. (2019) and Lin et al. (2017) introduced two new losses, respectively LDAM and Focal losses, in order to produce neural network classifiers that better handle imbalanced data sets. However, model-level approaches are not model agnostic, and thus cannot be applied to a wide variety of machine learning algorithms. Consequently, we focus in this paper on data-level approaches.

Data-level approaches can be divided into two groups: synthetic and non-synthetic procedures. Non-synthetic procedures works by removing or copying original data points. Mani & Zhang (2003) explain that Random Under Sampling (RUS) is one of the most used resampling strategy and design new adaptive versions called Nearmiss. RUS produces the prespecified balance between classes by dropping uniformly at random majority class samples. The Nearmiss1 strategy (Mani & Zhang, 2003) includes a distinction between majority samples by ranking them with their mean distance to their nearest neighbor from the minority class. Then, low-ranked majority samples are dropped until a given balancing ratio is reached. In contrast, Random Over Sampling (ROS) duplicates original minority samples. The main limitation of all these sampling strategies is the fact that they either remove information from the data or do not add new information.

On the contrary, synthetic procedures generate new synthetic samples in the minority class. One of the most famous strategies in this group is *Synthetic Minority Oversampling Technique* (SMOTE, see Chawla et al., 2002)[1]. In SMOTE, new minority samples are generated via linear interpolation between an original minority sample and one of its nearest neighbor in the minority class. Other approaches are based on Generative Adversarial Networks (GAN Islam & Zhang, 2020), which are computationally expensive and mostly designed for specific data structures, such as images. Random Over Sampling Examples (see Menardi & Torelli, 2014) is a variant of ROS that produces duplicated samples and then add a noise in order to get these samples slightly different from the original ones. This leads to the generation of new samples on the neighborhood of original minority samples. The main difficulty of these strategies is to synthesize relevant new samples, which must not be outliers nor simple copies of original points.

**Contributions** We place ourselves in the setting of imbalanced classification on tabular data, which is very common in real-world applications (see Shwartz-Ziv & Armon, 2022). In this paper:

- We prove that, without tuning the hyperparameter $K$ (usually set to 5), SMOTE asymptotically copies the original minority samples, therefore lacking the intrinsic variability required in any synthetic generative procedure. We provide numerical illustrations of this limitation (Section 3).

- We also establish that SMOTE density vanishes near the boundary of the support of the minority distribution, therefore justifying the introduction of SMOTE variants such as BorderLine SMOTE (Section 3).

- Our theoretical analysis naturally leads us to introduce two SMOTE alternatives, CV-SMOTE and Multivariate Gaussian SMOTE (MGS). In Section 4, we evaluate our new strategies and state-of-the-art rebalancing strategies on several real-world data sets using random forests/logistic regression/LightGBM. Through these experiments [2] we show that applying no strategy is competitive for most data sets. For the remaining data sets, our proposed strategies, CV-SMOTE and MGS, are among the best strategies in terms of predictive performances. Our analysis also provides some explanations about the good behavior of RUS, due to an implicit regularization in presence of random forests classifiers.

## 2 RELATED WORKS

In this section, we focus on the literature that is the most relevant to our work: long-tail learning, SMOTE variants and theoretical studies of rebalancing strategies.

Long-tailed learning (see, e.g., Zhang et al., 2023) is a relatively new field, originally designed to handle image classification with numerous output classes. Most techniques in long-tailed learning are based on neural networks or use the large number of classes to build or adapt aggregated predictors. However, in most tabular classification data sets, the number of classes to predict is relatively small, usually equal to two (Chawla et al., 2004; He & Garcia, 2009; Grinsztajn et al., 2022). Therefore, long-tailed learning methods are not intended for our setting as $(i)$ we only have two output classes and $(ii)$ state-of-the-art models for tabular data are not neural networks but tree-based methods, such as random forests or gradient boosting (see Grinsztajn et al., 2022; Shwartz-Ziv & Armon, 2022).

SMOTE has seen many variants proposed in the literrature. Several of them focus on generating synthetic samples near the boundary of the minority class support, such as ADASYN (He et al., 2008), SVM-SMOTE (Nguyen et al., 2011) or Borderline SMOTE (Han et al., 2005). Many other variants exist such as SMOTEBoost (Chawla et al., 2003), Adaptive-SMOTE (Pan et al., 2020), Xie et al. (2020) or DBSMOTE (Bunkhumpornpat et al., 2012). From a computational perspective, several synthetic methods are available in the open-source package *imb-learn* (see Lemaître et al., 2017). Several papers study experimentally some specificities of the sampling strategies and the impact of hyperparameter tuning. For example, Kamalov et al. (2022) study the optimal sampling ratio for imbalanced data sets when using synthetic approaches. Aguiar et al. (2023) realize a survey

---

[1]More than 25.000 papers found in GoogleScholar with a title including "SMOTE" over the last decade.

[2]All our experiments and our newly proposed methods can be found at `https://github.com/anonymous8880/smote_study`.

on imbalance data sets in the context of online learning and propose a standardized framework in order to compare rebalancing strategies in this context. Furthermore, Wongvorachan et al. (2023) aim at comparing the synthetic approaches (ROS, RUS and SMOTE) on educational data.

Several works study theoretically the rebalancing strategies. Xu et al. (2020) study the weighted risk of plug-in classifiers, for arbitrary weights. They establish rates of convergence and derive a new robust risk that may in turn improve classification performance in imbalanced scenarios. Then, based on this previous work, Aghbalou et al. (2024) derive a sharp error bound of the balanced risk for binary classification context with severe class imbalance. Using extreme value theory, Chaudhuri et al. (2023) show that applying Random Under Sampling in binary classification framework improve the worst-group error when learning from imbalanced classes with tails. Wallace & Dahabreh (2014) study the class probability estimates for several rebalancing strategies before introducing a generic methodology in order to improve all these estimates. Dal Pozzolo et al. (2015) focus on the effect of RUS on the posterior probability of the selected classifier. They show that RUS affect the accuracy and the probability calibration of the model. To the best of our knowledge, there are only few theoretical works dissecting the intrinsic machinery in SMOTE algorithm, with the notable exception of Elreedy & Atiya (2019) and Elreedy et al. (2023) who established the density of synthetic observations generated by SMOTE, the associated expectation and covariance matrix.

# 3 A STUDY OF SMOTE

**Notations** We denote by $\mathcal{U}([a,b])$ the uniform distribution over $[a,b]$. We denote by $\mathcal{N}(\mu, \Sigma)$ the multivariate normal distribution of mean $\mu \in \mathbb{R}^d$ and covariance matrix $\Sigma \in \mathbb{R}^{d \times d}$. For any set $A$, we denote by $Vol(A)$, the Lebesgue measure of $A$. For any $z \in \mathbb{R}^d$ and $r > 0$, let $B(z, r)$ be the ball centered at $z$ of radius $r$. We note $c_d = Vol(B(0,1))$ the volume of the unit ball in $\mathbb{R}^d$. For any $p, q \in \mathbb{N}$, and any $z \in [0, 1]$, we denote by $\mathcal{B}(p, q; z) = \int_{t=0}^{z} t^{p-1}(1-t)^{q-1} \mathrm{d}t$ the incomplete beta function.

## 3.1 SMOTE ALGORITHM

We assume to be given a training sample composed of $(X_i, Y_i)$ $N$ pairs independent and identically distributed as $(X, Y)$, where $X$ and $Y$ are random variables that take values respectively in $\mathcal{X} \subset \mathbb{R}^d$ and $\{0, 1\}$. We consider a class imbalance problem, in which the class $Y = 1$ is under-represented, compared to the class $Y = 0$, and thus called the minority class. We assume that we have $n$ minority samples in our training set. We define the imbalance ratio as $n/N$. In this paper, we

---

**Algorithm 1** SMOTE iteration.

**Input:** Minority class samples $X_1, \ldots, X_n$, number $K$ of nearest-neighbors
Select uniformly at random $X_c$ (called **central point**) among $X_1, \ldots, X_n$.
Denote by $I$ the set composed of the $K$ nearest-neighbors of $X_c$ among $X_1, \ldots, X_n$ (w.r.t. $L_2$ norm).
Select $X_k \in I$ uniformly.
Sample $w \sim \mathcal{U}([0, 1])$
$Z_{K,n} \leftarrow X_c + w(X_k - X_c)$
**Return** $Z_{K,n}$

---

consider continuous input variables only, as SMOTE was originally designed such variables only.

In this section, we study the SMOTE procedure, which generates synthetic data through linear interpolations between two pairs of original samples of the minority class. SMOTE algorithm has a single hyperparameter, $K$, by default set to 5, which stands for the number of nearest neighbors considered when interpolating. A single SMOTE iteration is detailed in Algorithm 1. In a classic machine learning pipeline, SMOTE procedure is repeated in order to obtain a prespecified ratio between the two classes, before training a classifier.

## 3.2 THEORETICAL RESULTS ON SMOTE

SMOTE has been shown to exhibit good performances when combined to standard classification algorithms (see, e.g., Mohammed et al., 2020). However, there exist only few works that aim at understanding theoretically SMOTE behavior. In this section, we assume that $X_1, \ldots, X_n$ are i.i.d samples from the minority class (that is, $Y_i = 1$ for all $i \in [n]$), with a common density $f_X$ with bounded support, denoted by $\mathcal{X}$.

**Lemma 3.1** (Convexity). *Given $f_X$ the distribution density of the minority class, with support $\mathcal{X}$, for all $K, n$, the associated SMOTE density $f_{Z_{K,n}}$ satisfies*

$$Supp(f_{Z_{K,n}}) \subseteq Conv(\mathcal{X}). \tag{1}$$

By construction, synthetic observations generated by SMOTE cannot fall outside the convex hull of $\mathcal{X}$. Equation equation 1 is not an equality, as SMOTE samples are the convex combination of only two original samples. For example, in dimension two, if $\mathcal{X}$ is concentrated near the vertices of a triangle, then SMOTE samples are distributed near the triangle edges, whereas $Conv(\mathcal{X})$ is the surface delimited by the triangle.

SMOTE algorithm has only one hyperparameter $K$, which is the number of nearest neighbors taken into account for building the linear interpolation. By default, this parameter is set to 5. The following theorem describes the behavior of SMOTE distribution asymptotically, as $K/n \to 0$.

**Theorem 3.2.** *For all Borel sets $B \subset \mathbb{R}^d$, if $K/n \to 0$, as $n$ tends to infinity, we have*

$$\lim_{n \to \infty} \mathbb{P}[Z_{K,n} \in B] = \mathbb{P}[X \in B]. \tag{2}$$

The proof of Theorem 3.2 can be found in B.2. Theorem 3.2 proves that the random variables $Z_{K,n}$ generated by SMOTE converge in distribution to the original random variable $X$, provided that $K/n$ tends to zero. From a practical point of view, Theorem 3.2 guarantees asymptotically the ability of SMOTE to regenerate the distribution of the minority class. This highlights a good behavior of the default setting of SMOTE ($K = 5$), as it can create more data points, different from the original sample, and distributed as the original sample. Note that Theorem 3.2 is very generic, as it makes no assumptions on the distribution of $X$.

SMOTE distribution has been derived in Theorem 1 and Lemma 1 in Elreedy et al. (2023). We provide here a slightly different expression for the density of the data generated by SMOTE, denoted by $f_{Z_{K,n}}$. Although our proof shares the same structure as that of Elreedy et al. (2023), our starting point is different, as we consider random variables instead of geometrical arguments. The proof can be found in Section B.3. When no confusion is possible, we simply write $f_Z$ instead of $f_{Z_{K,n}}$.

**Lemma 3.3.** *Let $X_c$ be the central point chosen in a SMOTE iteration. Then, for all $x_c \in \mathcal{X}$, the random variable $Z_{K,n}$ generated by SMOTE has a conditional density $f_{Z_{K,n}}(.|X_c = x_c)$ which satisfies*

$$f_{Z_{K,n}}(z|X_c = x_c) = (n - K - 1)\binom{n-1}{K} \int_0^1 \frac{1}{w^d} f_X\left(x_c + \frac{z - x_c}{w}\right) \tag{3}$$
$$\times \mathcal{B}(n - K - 1, K; 1 - \beta_{x_c,z,w})\, \mathrm{d}w,$$

*where $\beta_{x_c,z,w} = \mu_X(B(x_c, \|z - x_c\|/w))$ and $\mu_X$ is the probability measure associated to $f_X$. Using the following substitution $w = \|z - x_c\|/r$, we have,*

$$f_{Z_{K,n}}(z|X_c = x_c) = (n - K - 1)\binom{n-1}{K} \int_{r=\|z-x_c\|}^{\infty} f_X\left(x_c + \frac{(z - x_c)r}{\|z - x_c\|}\right)$$
$$\times \frac{r^{d-2}\mathcal{B}(n - K - 1, K; 1 - \mu_X(B(x_c, r)))}{\|z - x_c\|^{d-1}}\, \mathrm{d}r. \tag{4}$$

A close inspection of Lemma 3.3 allows us to derive more precise bounds about the behavior of SMOTE, as established in Theorem 3.5.

**Assumption 3.4.** *There exists $R > 0$ such that $\mathcal{X} \subset B(0, R)$. Besides, there exist $0 < C_1 < C_2 < \infty$ such that for all $x \in \mathbb{R}^d$, $C_1 \mathbb{1}_{x \in \mathcal{X}} \leq f_X(x) \leq C_2 \mathbb{1}_{x \in \mathcal{X}}$.*

**Theorem 3.5.** *Grant Assumption 3.4. Let $x_c \in \mathcal{X}$ and $\alpha \in (0, 2R)$. For all $K \leq (n - 1)\mu_X(B(x_c, \alpha))$, we have*

$$\mathbb{P}(\|Z_{K,n} - X_c\|_2 \geq \alpha | X_c = x_c) \leq \eta_{\alpha,R,d} \exp\left(-2(n-1)\left(\mu_X(B(x_c, \alpha)) - \frac{K}{n-1}\right)^2\right) \tag{5}$$

*with $\eta_{\alpha,R,d} = C_2 c_d R^d \times \begin{cases} \ln\left(\frac{2R}{\alpha}\right) & \text{if } d = 1, \\ \frac{1}{d-1}\left(\left(\frac{2R}{\alpha}\right)^{d-1} - 1\right) & \text{if } d > 1. \end{cases}$*

*Consequently, if $\lim_{n \to \infty} K/n = 0$, we have, for all $x_c \in \mathcal{X}$, $Z_{K,n}|X_c = x_c \to x_c$ in probability.*

The proof of Theorem 3.5 can be found in B.4. Theorem 3.5 establishes an upper bound on the distance between an observation generated by SMOTE and its central point. Asymptotically, when $K/n$ tends to zero, the new synthetic observation concentrates around the central point. Recall that, by default, $K = 5$ in SMOTE algorithm. Therefore, Theorem 3.2 and Theorem 3.5 prove that, with the default settings, SMOTE asymptotically targets the original density of the minority class and generates new observations very close to the original ones. The following result establishes the characteristic distance between SMOTE observations and their central points.

**Corollary 3.6.** *Grant Assumption 3.4. For all $d \geq 2$, for all $\gamma \in (0, 1/d)$, we have*

$$\mathbb{P}\left[\|Z_{K,n} - X_c\|_2^2 > 12R(K/n)^\gamma\right] \leq \left(\frac{K}{n}\right)^{2/d - 2\gamma}. \tag{6}$$

The proof of Corollary 3.6 can be found in B.5 and is an adaptation of Theorem 2.4 in Biau & Devroye (2015). The characteristic distance between a SMOTE observation and the associated central point is of order $(K/n)^{1/d}$. As expected from the curse of dimensionality, this distance increases with the dimension $d$. Choosing $K$ that increases with $n$ leads to larger characteristic distances: SMOTE observations are more distant from their central points. Corollary 3.6 leads us to choose $K$ such that $K/n$ does not tend too fast to zero, so that SMOTE observations are not too close to the original minority samples. However, choosing such a $K$ can be problematic, especially near the boundary of the support, as shown in the following theorem.

**Theorem 3.7.** *Grant Assumption 3.4 with $\mathcal{X} = B(0, R)$. Let $\varepsilon \in (0, R)$ such that $\left(\frac{\varepsilon}{R}\right)^{1/2} \leq \frac{c_d}{\sqrt{2}dC_2}$. Then, for all $1 \leq K < n$, and all $z \in B(0, R) \backslash B(0, R - \varepsilon)$, and for all $d > 1$, we have*

$$f_{Z_{K,n}}(z) \leq C_2^{3/2} \left(\frac{2^{d+2}c_d^{1/2}}{d^{1/2}}\right) \left(\frac{n-1}{K}\right) \left(\frac{\varepsilon}{R}\right)^{1/4}. \tag{7}$$

The proof of Theorem 3.7 can be found in B.6. Theorem 3.7 establishes an upper bound of SMOTE density at points distant from less than $\varepsilon$ from the boundary of the minority class support. More precisely, Theorem 3.7 shows that SMOTE density vanishes as $\varepsilon^{1/4}$ near the boundary of the support. Choosing $\varepsilon/R = o((K/n)^4)$ leads to a vanishing upper bound, which proves that SMOTE density is unable to reproduce the original density $f_X \geq C_1$ in the peripheral area $B(0, R) \backslash B(0, R - \varepsilon)$. Such a behavior was expected since the boundary bias of local averaging methods (kernels, nearest neighbors, decision trees) has been extensively studied (see, e.g. Jones, 1993; Arya et al., 1995; Arlot & Genuer, 2014; Mourtada et al., 2020).

For default settings of SMOTE (i.e., $K = 5$), and large sample size, this area is relatively small ($\varepsilon = o(n^{-4})$). Still, Theorem 3.7 provides a theoretical ground for understanding the behavior of SMOTE near the boundary, a phenomenon that has led to introduce variants of SMOTE to circumvent this issue (see Borderline SMOTE in Han et al., 2005). While increasing $K$ leads to more diversity in the generated observations (as shown in Theorem 3.5), it increases the boundary bias of SMOTE. Indeed, choosing $K = n^{3/4}$ implies a boundary effect in the peripheral area $B(0, R) \backslash B(0, R - \varepsilon)$ for $\varepsilon = o(1/n)$, which may not be negligible. Finally, note that constants in the upper bounds are of reasonable size. Letting $d = 3$, $K = 5$, $X \sim \mathcal{U}(B_d(0, 1))$, the upper bound turns into $0.89n\varepsilon^{1/4}$.

### 3.3 NUMERICAL ILLUSTRATIONS

Through Section 3, we highlighted the fact that SMOTE asymptotically regenerates the distribution of the minority class, by tending to copy the minority samples. The purpose of this section is to numerically illustrate the theoretical limitations of SMOTE, typically with the default value $K = 5$.

**Simulated data** In order to measure the similarity between any generated data set $\mathbf{Z} = \{Z_1, \ldots, Z_m\}$ and the original data set $\mathbf{X} = \{X_1, \ldots, X_n\}$, we compute $C(\mathbf{Z}, \mathbf{X}) = \frac{1}{m}\sum_{i=1}^{m}\|Z_i - X_{(1)}(Z_i)\|_2$, where $X_{(1)}(Z_i)$ is the nearest neighbor of $Z_i$ among $X_1, \ldots, X_n$. Intuitively, this quantity measures how far the generated data set is from the original observations: if the new data are copies of the original ones, this measure equals zero. We apply the following protocol: for each value of $n$,

1. Generate $\mathbf{X}$ composed of $n$ i.i.d samples distributed as $\mathcal{U}([-3, 3]^2)$.

2. Generate $\mathbf{Z}$ composed of $m = 1000$ new i.i.d observations by applying SMOTE procedure on the original data set $\mathbf{X}$, with different values of $K$. Compute $C(\mathbf{Z}, \mathbf{X})$.

3. Generate $\tilde{\mathbf{X}}$ composed of $m$ i.i.d new samples distributed as $\mathcal{U}([-3, 3]^2)$. Compute $C(\tilde{\mathbf{X}}, \mathbf{X})$, which is a reference value in the ideal case of new points sampled from the same distribution.

Steps 1-3 are repeated 75 times. The average of $C(\mathbf{Z}, \mathbf{X})$ (resp. $C(\tilde{\mathbf{X}}, \mathbf{X})$) over these repetitions is computed and denoted by $\bar{C}(\mathbf{Z}, \mathbf{X})$ (resp. $\bar{C}(\tilde{\mathbf{X}}, \mathbf{X})$). We consider the metric $\bar{C}(\mathbf{Z}, \mathbf{X})/\bar{C}(\tilde{\mathbf{X}}, \mathbf{X})$, depicted in Figure 1 (see also Figure 3 in Appendix for $\bar{C}(\mathbf{Z}, \mathbf{X})$).

**Results.** Figure 1 shows the renormalized quantity $\bar{C}(\mathbf{Z}, \mathbf{X})/\bar{C}(\tilde{\mathbf{X}}, \mathbf{X})$ as a function of $n$. We notice that the asymptotic for $K = 5$ is different since it is the only one where the distance between SMOTE data points and original data points does not vary with $n$. Besides, this distance is smaller than the other ones, thus stressing out that the SMOTE data points are very close to the original distribution for $K = 5$. Note that, for the other asymptotics in $K$, the diversity of SMOTE observations increases with $n$, meaning $\bar{C}(\mathbf{Z}, \mathbf{X})$ gets closer from $\bar{C}(\tilde{\mathbf{X}}, \mathbf{X})$. This behavior in terms of average distance is ideal, since $\tilde{\mathbf{X}}$ is drawn from the same theoretical distribution as $\mathbf{X}$. On the contrary, $K = 5$ keeps a lower average distance,

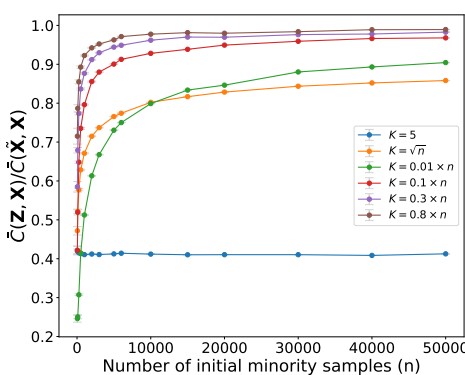

Figure 1: $\bar{C}(\mathbf{Z}, \mathbf{X})/\bar{C}(\tilde{\mathbf{X}}, \mathbf{X})$ with $\mathcal{U}([-3, 3]^2)$.

showing a lack of diversity of generated points. Besides, this diversity is asymptotically more important for $K = 0.1n$ and $K = 0.01n$. This corroborates our theoretical findings (Theorem 3.2) as these asymptotics do not satisfy $K/n \to 0$. Indeed, when $K$ is set to a fraction of $n$, the SMOTE distribution does not converge to the original distribution anymore, therefore generating data points that are not simple copies of the original uniform samples. By construction, SMOTE data points are close to central points, which may explain why the quantity of interest in Figure 1 is smaller than 1.

**Extension to real-world data sets** We extended our protocol to a real-world data set by splitting the data into two sets of equal size $\mathbf{X}$ and $\tilde{\mathbf{X}}$. The first one is used for applying SMOTE strategies to sample $\mathbf{Z}$ and the other set is used to compute the normalization factor $\bar{C}(\tilde{\mathbf{X}}, \mathbf{X})$. More details about this variant of the protocol are available on Appendix A.

**Results** We apply the adapted protocol to Phoneme data set, described in Table 1. Figure 2 displays the quantity $\bar{C}(\mathbf{Z}, \mathbf{X})/\bar{C}(\tilde{\mathbf{X}}, \mathbf{X})$ as a function of the size $n$ of the minority class. As above, we observe in Figure 2 that the average normalized distance $\bar{C}(\mathbf{Z}, \mathbf{X})/\bar{C}(\tilde{\mathbf{X}}, \mathbf{X})$ increases for all strategies but the one with $K =$

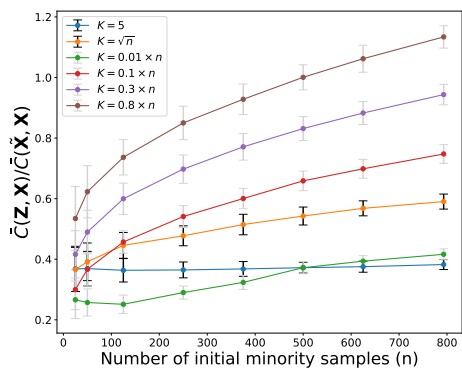

Figure 2: $\bar{C}(\mathbf{Z}, \mathbf{X})/\bar{C}(\tilde{\mathbf{X}}, \mathbf{X})$ with Phoneme data.

5. The strategies using a value of hyperparameter $K$ such that $K/n \to 0$ seem to converge to a value smaller than all the strategies with $K$ such that $K/n \not\to 0$.

## 4 Predictive evaluation on real-world data sets

In this section, we first describe the different rebalancing strategies and the two new ones we propose. Then, we describe our experimental protocol before discussing our results.

## 4.1 REBALANCING STRATEGIES

**Class-weight (CW) [Model-level strategy]** The class weighting strategy assigns the same weight (choosen as hyperparameter) to each minority samples. The default setting for this strategy is to choose a weight $\rho$ such that $\rho n = N - n$, where $n$ and $N - n$ are respectively the number of minority and majority samples in the data set.

**Over/Under Sampling strategies [Non-synthetic data-level strategies]** Random Under Sampling (RUS) acts on the majority class by selecting uniformly without replacement several samples in order to obtain a prespecified size for the majority class. Similarly, Random Over Sampling (ROS) acts on the minority class by selecting uniformly with replacement several samples to be copied in order to obtain a prespecified size for the minority class.

**NearMissOne [Non-synthetic data-level strategy]** NearMissOne is an undersampling procedure. For each sample $X_i$ in the majority class, the averaged distance of $X_i$ to its $K$ nearest neighbors in the minority class is computed. Then, the samples $X_i$ are ordered according to this averaged distance. Finally, iteratively, the first $X_i$ is dropped until the given number/ratio is reached. Consequently, the $X_i$ with the smallest mean distance are dropped firstly.

**Borderline SMOTE 1 and 2 [Synthetic data-level strategies]** Borderline SMOTE 1 (Han et al., 2005) procedure works as follows. For each individual $X_i$ in the minority class, let $m_-(X_i)$ be the number of samples of the majority class among the $m$ nearest neighbors of $X_i$, where $m$ is a hyperparameter. For all $X_i$ in the minority class such that $m/2 \leq m_-(X_i) < m$, generate $q$ successive samples $Z = WX_i + (1 - W)X_k$ where $W \sim \mathcal{U}([0, 1])$ and $X_k$ is selected among the $K$ nearest-neighbors of $X_i$ in the minority class. In Borderline SMOTE 2 (Han et al., 2005), the selected neighbor $X_k$ is chosen from the neighbors of both positive and negative classes, and $Z$ is sampled with $W \sim \mathcal{U}([0, 0.5])$.

Table 1: Initial data sets.

|               | $N$     | $n/N$  | $d$ |
|---------------|---------|--------|-----|
| Haberman      | 306     | 26%    | 3   |
| Ionosphere    | 351     | 36%    | 32  |
| Breast cancer | 630     | 36%    | 9   |
| Pima          | 768     | 35%    | 8   |
| Vehicle       | 846     | 23%    | 18  |
| Yeast         | 1 462   | 11%    | 8   |
| Abalone       | 4 177   | 1%     | 8   |
| Wine          | 4 974   | 4%     | 11  |
| Phoneme       | 5 404   | 29%    | 5   |
| MagicTel      | 13 376  | 50%    | 10  |
| House_16H     | 22 784  | 30%    | 16  |
| California    | 20 634  | 50%    | 8   |
| CreditCard    | 284 315 | 0.2%   | 29  |

The limitations of SMOTE highlighted in Section 3 drive us to two new rebalancing strategies.

**CV SMOTE [Synthetic data-level strategy]** We introduce a new algorithm, called CV SMOTE, that finds the best hyperparameter $K$ among a prespecified grid via a 5-fold cross-validation procedure. The grid is composed of the set $\{1, 2, \ldots, 15\}$ extended with the values $\lfloor 0.01 n_{train} \rfloor, \lfloor 0.1 n_{train} \rfloor, \lfloor 0.5 n_{train} \rfloor, \lfloor 0.7 n_{train} \rfloor$ and $\lfloor \sqrt{n_{train}} \rfloor$, where $n_{train}$ is the number of minority samples in the training set. Recall that through Theorem 3.5, we show that SMOTE procedure with the default value $K = 5$ asymptotically copies the original samples. The idea of CV SMOTE is then to test several (larger) values of $K$ in order to avoid duplicating samples, therefore improving predictive performances. CV SMOTE is one of the simplest ideas to solve some SMOTE limitations, which were highlighted theoretically in Section 3.

**Multivariate Gaussian SMOTE($K$) (MGS) [Synthetic data-level strategy]** We introduce a new oversampling strategy in which new samples are generated from the distribution $\mathcal{N}(\hat{\mu}, \hat{\Sigma})$, where the empirical mean $\hat{\mu}$ and covariance matrix $\hat{\Sigma}$ are estimated using the $K$ neighbors and the central point (see Algorithm 2 for details). By default, we choose $K = d + 1$, so that estimated covariance matrices can be of full rank. MGS produces more diverse synthetic observations than SMOTE as they are spread in all directions (in case of full-rank covariance matrix) around the central point. Besides, sampling from a normal distribution may generate points outside the convex hull of the nearest neighbors.

## 4.2 PRELIMINARY RESULTS

**Initial data sets** We employ classical tabular data sets already used in Grinsztajn et al. (2022). We also used some data sets from UCI Irvine (see Dua & Graff, 2017; Grinsztajn et al.,

2022) and other public data sets such as Phoneme (Alinat, 1993) and Credit Card (Dal Pozzolo et al., 2015). All data sets are described in Table 1 and we call them *initial data sets*. As we want to compare several rebalancing methods including SMOTE, originally designed to handle continuous variables only, we have removed all categorical variables in each data set.

**Protocol** We compare the different rebalancing strategies on the initial data sets of Table 1. We employ a 5-fold stratified cross-validation, and apply each rebalancing strategy on four training folds, in order to obtain the same number of minority/majority samples. Then, we train a Random Forest classifier (showing good predictive performance, see Grinsztajn et al., 2022) on the same folds, and evaluate its performance on the remaining fold, via the ROC AUC. Results are averaged over the five test folds and over 20 repetitions of the cross-validation. We use the `RandomForestClassifier` module in *scikit-learn* (Pedregosa et al., 2011) and tune the tree depth (when desired) via nested cross-validation Cawley & Talbot (2010). We use the implementation of *imb-learn* (Lemaître et al., 2017) for the state-of-the-art rebalancing strategies (see Appendix A.2 for implementation details).

---

**Algorithm 2** Multivariate Gaussian SMOTE iteration.

---

**Input:** Minority class samples $X_1, \ldots, X_n$, number $K$ of nearest-neighbors.

Select uniformly $X_c$ among $X_1, \ldots, X_n$.

Denote by $I$ the set composed of the $K + 1$ nearest-neighbors of $X_c$ among $X_1, \ldots, X_n$ including $X_c$ (w.r.t. $L_2$ norm).

$\hat{\mu} \leftarrow \frac{1}{K+1} \sum\limits_{x \in I} x$

$\hat{\Sigma} \leftarrow \frac{1}{K+1} \sum\limits_{x \in I} (x - \hat{\mu})^T (x - \hat{\mu})$

Sample $Z \sim \mathcal{N}\left(\hat{\mu}, \hat{\Sigma}\right)$

**Return** $Z$

---

**None is competitive for low imbalanced data sets** For 10 initial data sets out of 13, applying no strategy is the best, probably highlighting that the imbalance ratio is not high enough or the learning task not difficult enough to require a tailored rebalancing strategy. Therefore, considering only continuous input variables, and measuring the predictive performance with ROC AUC, we observe that dedicated rebalancing strategies are not required for most data sets. While the performance without applying any strategy was already perceived in the literature (see, e.g., Han et al., 2005; He et al., 2008), we believe that our analysis advocates for its broad use in practice, at least as a default method. Note that for these 10 data sets, qualified as low imbalanced, applying no rebalancing strategy is on par with the CW strategy, one of the most common rebalancing strategies (regardless of tree depth tuning, see Table 5 and Table 7).

### 4.3 Experiments on highly imbalanced data sets

**Strengthening the imbalance** To analyze what could happen for data sets with higher imbalance ratio, we subsample the minority class for each one of the initial data sets mentioned above, so that the resulting imbalance ratio is set to 20%, 10% or 1% (when possible, taking into account dimension $d$). By doing so, we reproduce the high imbalance that is often encountered in practice (see He & Garcia, 2009). We apply our subsampling strategy once for each data set and each imbalance ratio in a nested fashion, so that the minority samples of the 1% data set are included in the minority samples of the 10% data set. The new data sets thus obtained are called *subsampled data sets* and presented in Table 4 in Appendix A.2. For the sake of brevity, we display in Table 2 the data sets among the initial and subsampled for which the None strategy is not the best (up to its standard deviation). The others are presented in Table 5 in Appendix A.3.

Hereafter, we discuss the performances of rebalancing methods presented in Table 2. We remark that the included data sets correspond to the most imbalanced subsampling for each data set, or simply the initial data set in case of high initial imbalance. Therefore, in the following, we refer to them as *highly imbalanced data sets*.

**Performances on highly imbalanced data sets** Whilst in the vast majority of experiments, applying no rebalancing is among the best approaches to deal with imbalanced data (see Table 5), it seems to be outperformed by dedicated rebalancing strategies for highly imbalanced data sets (Table 2). Surprisingly, most rebalancing strategies do not benefit drastically from tree depth tuning, with the notable exceptions of applying no rebalancing and CW (see the differences between Table 2 and Table 6).

Table 2: Highly imbalanced data sets ROC AUC (max_depth tuned with ROC AUC). Only data sets whose ROC AUC of at least one rebalancing strategy is larger than that of None strategy plus its standard deviation are displayed. Undersampled data sets are in italics. Standard deviations are displayed in Table 10.

| Strategy | None | CW | RUS | ROS | Near Miss1 | BS1 | BS2 | SMOTE | CV SMOTE | MGS $(d+1)$ |
|---|---|---|---|---|---|---|---|---|---|---|
| CreditCard (0.2%) | 0.966 | 0.967 | **0.970** | 0.935 | 0.892 | 0.949 | 0.944 | 0.947 | 0.954 | 0.952 |
| Abalone (1%) | 0.764 | 0.748 | 0.735 | 0.722 | 0.656 | 0.744 | 0.753 | 0.741 | 0.791 | **0.802** |
| *Phoneme* (1%) | 0.897 | 0.868 | 0.868 | 0.858 | 0.698 | 0.867 | 0.869 | 0.888 | **0.924** | 0.915 |
| *Yeast* (1%) | 0.925 | 0.920 | 0.938 | 0.908 | 0.716 | 0.949 | 0.954 | **0.955** | 0.942 | 0.945 |
| Wine (4%) | 0.928 | 0.925 | 0.915 | 0.924 | 0.682 | 0.933 | 0.927 | 0.934 | 0.938 | **0.941** |
| *Pima* (20%) | 0.798 | **0.808** | 0.799 | 0.790 | 0.777 | 0.793 | 0.788 | 0.789 | 0.787 | 0.787 |
| *Haberman* (10%) | 0.708 | 0.709 | 0.720 | 0.704 | 0.697 | 0.723 | 0.721 | 0.719 | 0.742 | **0.744** |
| *MagicTel* (20%) | 0.917 | 0.921 | 0.917 | **0.922** | 0.649 | 0.920 | 0.905 | 0.921 | 0.919 | 0.913 |
| *California* (1%) | 0.887 | 0.877 | 0.880 | 0.883 | 0.630 | 0.885 | 0.874 | 0.906 | 0.916 | **0.923** |

**Re-weighting strategies** RUS, ROS and CW are similar strategies in that they are equivalent to applying weights to the original samples. When random forests with default parameters are applied, we see that ROS and CW have the same predictive performances (see Table 6). This was expected, as ROS assigns random weights to minority samples, whose expectation is that of the weights produced by CW. More importantly, RUS has better performances than both ROS and CW. This advocates for the use of RUS among these three rebalancing methods, as RUS produces smaller data sets, thus resulting in faster learning phases. We describe another benefit of RUS in the next paragraph.

**Implicit regularization** The good performances of RUS, compared to ROS and CW, may result from the implicit regularization of the maximum tree depth. Indeed, fewer samples are available after the undersampling step, which makes the resulting trees shallower, as by default, each leaf contains at least one observation. When the maximum tree depth is fixed, RUS, ROS and CW strategies have the same predictive performances (see Table 8 or Table 9). Similarly, when the tree depth is tuned, the predictive performances of RUS, ROS and CW are smoothed out (see Table 2). This highlights the importance of implicit regularization on RUS good performances.

**SMOTE and CV-SMOTE** Default SMOTE ($K = 5$) has a tendency to duplicate original observations, as shown by Theorem 3.5. This behavior is illustrated through our experiments when the tree depth is fixed. In this context, SMOTE ($K = 5$) has the same behavior as ROS, a method that copies original samples (see Table 8 or Table 9). When the tree depth is tuned, SMOTE may exhibit better performances compared to reweighting methods (ROS, RUS, CW), probably due to a higher tree depth. Indeed, even if synthetic data are close to the original samples, they are distinct and thus allow for more splits in the tree structure. However, as expected, CV SMOTE performances are higher than default SMOTE ($K = 5$) on most data sets (see Table 2).

**MGS** Our second new publicly available [3] strategy exhibits good predictive performances (best performance in 4 out of 9 data sets in Table 2). This could be explained by the Gaussian sampling of synthetic observations that allows generating data points outside the convex hull of the minority class, therefore limiting the border phenomenon, established in Theorem 3.7. Note that with MGS, there is no need of tuning the tree depth: predictive performances of default RF are on par with tuned RF. Thus, MGS is a promising new strategy.

## 4.4 SUPPLEMENTARY RESULTS

**Logistic Regression** When replacing random forests with Logistic regression in the above protocol (see Table 15), we still do not observe strong benefits of using a rebalancing strategies for most data sets. We compared in Table 17 the LDAM and Focal losses intended for long-tailed learning, using PyTorch. Table 17 shows that Focal loss performances are on par with the None strategy ones, while the performances of LDAM are significantly lower. Such methods do not seem promising for binary classification on tabular data, for which they were not initially intended.

---

[3] https://github.com/anonymous8880/smote_study

**LightGBM - ROC AUC**   We apply the same protocol as in Section 4.2, using LightGBM (a second-order boosting algorithm, see Ke et al., 2017) instead of random forests. Again, only data sets such that None strategy is not competitive (in terms of ROC AUC) are displayed in Table 3 (the remaining ones can be found in Table 20). In Table 3, we note that our introduced strategies, CV-SMOTE and MGS, display good predictive results.

Table 3: LightGBM ROC AUC. Only data sets whose ROC AUC of at least one rebalancing strategy is larger than that of None strategy plus its standard deviation are displayed. Undersampled data sets are in italics. Standard deviations are displayed in Table 20.

| Strategy | None | CW | RUS | ROS | Near Miss1 | BS1 | BS2 | SMOTE | CV SMOTE | MGS $(d+1)$ |
|---|---|---|---|---|---|---|---|---|---|---|
| CreditCard $(0.2\%)$ | 0.761 | 0.938 | **0.970** | 0.921 | 0.879 | 0.941 | 0.932 | 0.937 | 0.950 | 0.956 |
| Abalone $(1\%)$ | 0.738 | 0.738 | 0.726 | 0.738 | 0.700 | 0.750 | 0.757 | 0.748 | **0.775** | 0.745 |
| *Haberman* $(10\%)$ | 0.691 | 0.689 | 0.575 | 0.643 | 0.564 | 0.710 | 0.674 | 0.712 | 0.726 | **0.729** |
| *House_16H* $(1\%)$ | 0.903 | 0.896 | 0.899 | 0.896 | 0.605 | 0.907 | 0.909 | 0.894 | 0.894 | **0.912** |

**PR AUC**   As above, we apply exactly the same protocol as described in Section 4.2 using the PR AUC metric instead of the ROC AUC. The results are displayed in Table 13 and Table 14 for tuned random forests. For LightGBM classifiers, results are available in Table 18 and Table 19. Again, we only focus on data sets such that None strategy is not competitive (in terms of PR AUC). In Table 13, for random forests tuned on PR AUC, we remark that CV-SMOTE exhibits good performances, being among the best rebalancing strategy for 3 out of 4 data sets. For LightGBM classifier, in Table 18, we note that our introduced strategies, CV-SMOTE and MGS, display good predictive results.

## 5   CONCLUSION AND PERSPECTIVES

In this paper, we analyzed the impact of rebalancing strategies on predictive performance for binary classification tasks on tabular data. First, we prove that default SMOTE tends to copy the original minority samples asymptotically, and exhibits boundary artifacts, thus justifying several SMOTE variants. From a computational perspective, we show that applying no rebalancing is competitive for most datasets, when used in conjunction with a tuned random forest/Logistic regression/LightGBM, whether considering the ROC AUC or PR AUC as metric. For highly imbalanced data sets, rebalancing strategies lead to improved predictive performances, with or without tuning the maximum tree depth. The SMOTE variants we propose, CV-SMOTE and MGS, appear promising, with good predictive performances regardless of the hyperparameter tuning of random forests. Besides, our analysis sheds some lights on the performances of reweighting strategies (ROS, RUS, CW) and an implicit regularization phenomenon occurring when such rebalancing methods are used with random forests.

More analyses need to be carried out in order to understand the influence of MGS parameters (regularization of the covariance matrices, number of nearest neighbors...). We also plan to extend our new MGS method to handle categorical features, and compare the different rebalancing strategies in presence of continuous and categorical input variables.

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

## A  Experiments

**Hardware**  For all the numerical experiments, we use the following processor : AMD Ryzen Threadripper PRO 5955WX: 16 cores, 4.0 GHz, 64 MB cache, PCIe 4.0. We also add access to 250GB of RAM.

### A.1  Numerical illustrations

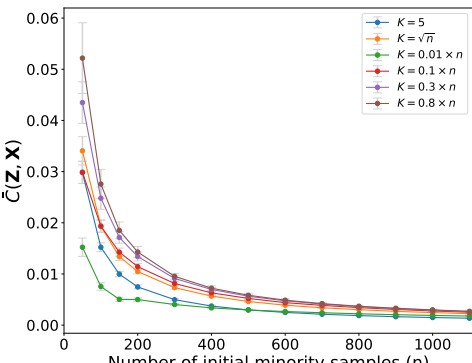

Figure 3: Average distance $\bar{C}(\mathbf{Z}, \mathbf{X})$.

**Results with $\bar{C}(\mathbf{Z}, \mathbf{X})$**  Figure 3 depicts the quantity $\bar{C}(\mathbf{Z}, \mathbf{X})$ as a function of the size of the minority class, for different values of $K$. The metric $\bar{C}(\mathbf{Z}, \mathbf{X})$ is consistently smaller for $K = 5$ than for other values of $K$, therefore highlighting that data generated by SMOTE with $K = 5$ are closer to the original data sample. This phenomenon is strengthened as $n$ increases. This is an artifact of the simulation setting as the original data samples fill the input space as $n$ increases.

**More details on the numerical illustrations protocol applied to real-world data sets**  We apply SMOTE on real-world data and compare the distribution of the generated data points to the original distribution, using the metric $\bar{C}(\mathbf{Z}, \mathbf{X})/\bar{C}(\tilde{\mathbf{X}}, \mathbf{X})$.

For each value of $n$, we subsample $n$ data points from the minority class. Then,

1. We uniformly split the data set into $X_1, \ldots, X_{n/2}$ (denoted by $\mathbf{X}$) and $\tilde{X}_1, \ldots, \tilde{X}_{n/2}$ (denoted by $\tilde{\mathbf{X}}$).

2. We generate a data set $\mathbf{Z}$ composed of $m = n/2$ i.i.d new observations $Z_1, \ldots, Z_m$ by applying SMOTE procedure on the original data set $\mathbf{X}$, with different values of $K$. We compute $C(\mathbf{Z}, \mathbf{X})$.

3. We use $\tilde{\mathbf{X}}$ in order to compute $C(\tilde{\mathbf{X}}, \mathbf{X})$.

This procedure is repeated $B = 100$ times to compute averages values as in Section 3.3.

### A.2  Binary classification protocol

**General comment on the protocol**  For each data set, the ratio hyperparameters of each rebalancing strategy are chosen so that the minority and majority class have the same weights in the training phase.The main purpose is to apply the strategies exactly on the same data points ($X_{train}$), then train the chosen classifier and evaluate the strategies on the same $X_{test}$. This objective is achieved by selecting each time 4 fold for the training, apply each of the strategies to these 4 exact same fold.

The state-of-the-art rebalancing strategies(see Lemaître et al., 2017) are used with their default hyperparameter values.

Table 4: Subsampled data sets.

|  | $N$ | $n/N$ | $d$ |
|---|---|---|---|
| *Haberman* (10%) | 250 | 10% | 3 |
| *Ionosphere* (20%) | 281 | 20% | 32 |
| *Ionosphere* (10%) | 250 | 10% | 32 |
| *Breast cancer* (20%) | 500 | 20% | 9 |
| *Breast cancer* (10%) | 444 | 10% | 9 |
| Pima (20%) | 625 | 20% | 8 |
| *Vehicle* (10%) | 718 | 10% | 18 |
| *Yeast* (1%) | 1 334 | 1% | 8 |
| *Phoneme* (20%) | 4 772 | 20% | 5 |
| *Phoneme* (10%) | 4 242 | 10% | 5 |
| *Phoneme* (1%) | 3 856 | 1% | 5 |
| *MagicTel* (20%) | 8 360 | 20% | 10 |
| *House_16H* (20%) | 20 050 | 20% | 16 |
| *House_16H* (10%) | 17 822 | 10% | 16 |
| *House_16H* (1%) | 16 202 | 1% | 16 |
| *California* (20%) | 12 896 | 20% | 8 |
| *California* (10%) | 11 463 | 10% | 8 |
| *California* (1%) | 10 421 | 1% | 8 |

The subsampled data sets (see Table 4) can be obtained through the repository (the functions and the seeds are given in a jupyter notebook). For the CreditCard data set, a Time Series split is performed instead of a Stratified $5-$fold, because of the temporality of the data. Furthermore, a group out is applied on the different scope time value.

For MagicTel and California data sets, the initial data sets are already balanced, leading to no opportunity for applying a rebalancing strategy. This is the reason why we do not include these original data sets in our study but only their subsampled associated data sets.

The max_depth hyperparameter is tuned using GridSearch function from scikit-learn. The grids minimum is 5 and the grid maximum is the mean depth of the given strategy for the given data set (when random forest is used without tuning depth hyperparameter). Then, using numpy (Harris et al., 2020), a list of integer of size 8 between the minimum value and the maximum is value is built. Finally, the "None" value is added to this list.

**Mean standard deviation**    For each protocol run, we computed the standard deviation of the ROC AUC over the 5-fold. Then, all of these 100 standard deviation are averaged in order to get what we call in some of our tables the mean standard deviations. On Table 10, Table 11 and Table 17 means standard deviation over 100 runs are displayed for each strategy (no averaging is performed).

**CV SMOTE**    We also apply our protocol for SMOTE with values of hyperparameter $K$ depending on the number of minority inside the training set. The results are shown on both Table 12 and Table 16. As expected, CV SMOTE is most of the time the best strategy among the SMOTE variants for highly imbalanced data sets. This another illustration of our Theorem 3.5.

**More results about None strategy from seminal papers**    Several seminal papers already noticed that the None strategy was competitive in terms of predictive performances. He et al. (2008) compare the None strategy, ADASYN and SMOTE, followed by a decision tree classifier on 5 data sets (including Vehicle, Pima, Ionosphere and Abalone). In terms of Precision and F1 score, the None strategy is on par with the two other rebalancing methods. Han et al. (2005) study the impact of Borderline SMOTE and others SMOTE variant on 4 data sets (including Pima and Haberman). The None strategy is competitive (in terms of F1 score) on two of these data sets.

**Random forests - PR AUC**    We apply exactly the same protocol as described in Section 4.2 but using the PR AUC metric instead of the ROC AUC. Data sets such that the None strategy is not competitive (in terms of PR AUC) are displayed in Table 13, others can be found in Table 14. As

for the ROC AUC metric, None and CW strategies are competitive for a large number of data sets (see Table 14). Furthermore, in Table 13, CV-SMOTE exhibits good performances, being among the best rebalancing strategy for 3 out of 4 data sets.

**LightGBM - PR AUC**    As above, we apply the same protocol as in Section 4.2, using the PR AUC metric instead of the ROC AUC and LightGBM (a second-order boosting algorithm, see Ke et al., 2017) instead of random forests. Again, only data sets such that None strategy is not competitive (in terms of PR AUC) are displayed in Table 18 (the remaining ones can be found in Table 19). In Table 18, we note that our introduced strategies, CV-SMOTE and MGS, display good predictive results.

The classification experiments needed 2 months of calculation time.

### A.3    ADDITIONAL EXPERIMENTS

**Tables**    The tables in appendix can be divided into 3 categories. First, we have the tables related to random forests. Then the tables related to logistic regression. Finally, we have the tables of LightGBM classifiers. Here are some details fore each group:

- **Random Forest:** In Table 5, ROC AUC of the data sets not presented Table 2 are displayed (using tuned forest on ROC AUC for both). In Table 6 and Table 7, ROC AUC of default random forests are displayed for all the data sets. In Table 8 and Table 9 are displayed default random forests ROC AUC with a max tree depth fixed to respectively 5 and the value of RUS depth. Table 10 and Table 11 illustrate respectively the same setting as Table 2 and Table 5 with the standard deviation displayed. In Table 12, the ROC AUC of several SMOTE strategies with various $K$ hyperparameter value are displayed using defaults random forests for all data sets. PR AUC of tuned random forests on PR AUC are displayed in Table 13 and Table 14.

- **Logistic Regression:** Table 15 display ROC AUC of several rebalancing strategies when using Logistic regression. In Table 16, the ROC AUC of several SMOTE strategies with various $K$ hyperparameter value are displayed using logistic regression for all data sets. Table 17 shows ROC AUC of None, LDAM and Focal loss strategies when using a logistic regression reimplemented using PyTorch.

- **LightGBM:** The PR AUC and ROC AUC of the remaining data sets when using Light-GBM classifiers are displayed respectively in Table 18/Table 19 and Table 20.

Table 5: Remaining data sets (without those of Table 2). Random Forest (max_depth= tuned with ROC AUC) ROC AUC for different rebalancing strategies and different data sets. Other data sets are presented in Table 2. The best strategy is highlighted in bold for each data set. Standard deviations are available on Table 11.

| Strategy | None | CW | RUS | ROS | Near Miss1 | BS1 | BS2 | SMOTE | CV SMOTE | MGS $(d+1)$ |
|---|---|---|---|---|---|---|---|---|---|---|
| Phoneme | **0.962** | 0.961 | 0.951 | **0.962** | 0.910 | 0.960 | 0.961 | **0.962** | 0.961 | 0.959 |
| *Phoneme* (20%) | 0.952 | 0.952 | 0.935 | **0.953** | 0.793 | 0.950 | 0.951 | **0.953** | 0.953 | 0.949 |
| *Phoneme* (10%) | **0.936** | 0.935 | 0.909 | **0.936** | 0.664 | 0.933 | 0.932 | 0.935 | 0.938 | 0.932 |
| Pima | **0.833** | 0.832 | 0.828 | 0.823 | 0.817 | 0.814 | 0.811 | 0.820 | 0.824 | 0.826 |
| Yeast | 0.968 | **0.971** | **0.971** | 0.968 | 0.921 | 0.964 | 0.965 | 0.968 | 0.969 | 0.968 |
| Haberman | 0.686 | 0.686 | 0.685 | 0.673 | 0.686 | 0.682 | 0.670 | 0.681 | 0.690 | **0.698** |
| *California* (20%) | **0.956** | 0.955 | 0.951 | **0.956** | 0.850 | 0.953 | 0.947 | 0.955 | **0.956** | 0.954 |
| *California* (10%) | 0.948 | 0.946 | 0.940 | 0.948 | 0.775 | 0.945 | 0.934 | 0.947 | **0.950** | 0.948 |
| House_16H | **0.950** | **0.950** | 0.948 | **0.950** | 0.899 | 0.945 | 0.942 | 0.948 | 0.949 | 0.948 |
| *House_16H* (20%) | **0.950** | 0.949 | 0.946 | 0.949 | 0.835 | 0.943 | 0.938 | 0.946 | 0.947 | 0.946 |
| *House_16H* (10%) | **0.945** | 0.943 | 0.940 | 0.944 | 0.717 | 0.939 | 0.931 | 0.939 | 0.942 | 0.937 |
| *House_16H* (1%) | **0.906** | 0.893 | 0.902 | 0.885 | 0.600 | 0.894 | 0.896 | 0.898 | 0.905 | 0.889 |
| Vehicle | **0.995** | 0.994 | 0.990 | 0.994 | 0.978 | 0.994 | 0.993 | 0.994 | **0.995** | **0.995** |
| Vehicle (10%) | 0.992 | 0.991 | 0.982 | 0.989 | 0.863 | 0.991 | 0.989 | 0.992 | 0.993 | **0.994** |
| Ionosphere | **0.978** | **0.978** | 0.974 | **0.978** | 0.945 | **0.978** | **0.978** | **0.978** | 0.977 | 0.976 |
| *Ionosphere* (20%) | **0.988** | 0.986 | 0.974 | 0.987 | 0.881 | 0.981 | 0.974 | 0.981 | 0.983 | 0.983 |
| *Ionosphere* (10%) | **0.988** | 0.983 | 0.944 | 0.981 | 0.822 | 0.972 | 0.962 | 0.966 | 0.967 | 0.968 |
| Breast Cancer | **0.994** | 0.993 | 0.993 | 0.993 | **0.994** | 0.992 | 0.992 | 0.993 | **0.994** | 0.993 |
| *Breast Cancer* (20%) | 0.996 | 0.995 | 0.994 | 0.995 | **0.997** | 0.994 | 0.993 | 0.995 | 0.996 | 0.996 |
| *Breast Cancer* (10%) | **0.997** | 0.996 | 0.994 | 0.996 | **0.997** | 0.993 | 0.992 | 0.996 | **0.997** | **0.997** |

Table 6: Highly imbalanced data sets. Random Forest (max_depth=$\infty$) ROC AUC for different rebalancing strategies and different data sets. Data sets artificially undersampled for minority class are in italics. Other data sets are presented in Table 7. Mean standard deviations are computed.

| Strategy | None | CW | RUS | ROS | Near Miss1 | BS1 | BS2 | SMOTE | CV SMOTE | MGS $(d+1)$ |
|---|---|---|---|---|---|---|---|---|---|---|
| CreditCard (0.2%)(±0.003) | 0.930 | 0.927 | **0.968** | 0.932 | 0.887 | 0.933 | 0.941 | 0.950 | 0.961 | 0.953 |
| Abalone (1%)(±0.018) | 0.716 | 0.698 | 0.732 | 0.699 | 0.652 | 0.745 | 0.754 | 0.744 | 0.777 | **0.805** |
| *Phoneme* (1%)(±0.020) | 0.852 | 0.851 | 0.864 | 0.840 | 0.690 | 0.859 | 0.863 | 0.883 | 0.893 | **0.913** |
| *Yeast* (1%)(±0.020) | 0.914 | 0.926 | 0.922 | 0.919 | 0.711 | 0.936 | **0.954** | 0.936 | **0.954** | 0.932 |
| Wine (4%)(±0.008) | 0.926 | 0.923 | 0.917 | 0.927 | 0.693 | 0.934 | 0.927 | 0.934 | 0.935 | **0.939** |
| *Pima* (20%)(±0.009) | 0.777 | 0.791 | **0.796** | 0.787 | 0.767 | 0.791 | 0.790 | 0.789 | 0.786 | 0.786 |
| *Haberman* (10%)(±0.028) | 0.680 | 0.685 | 0.709 | 0.688 | 0.697 | 0.716 | 0.713 | 0.721 | 0.735 | **0.736** |
| *MagicTel* (20%)(±0.001) | 0.917 | **0.921** | 0.917 | **0.921** | 0.650 | 0.920 | 0.905 | **0.921** | **0.921** | 0.913 |
| *California* (1%)(±0.009) | 0.857 | 0.871 | 0.881 | 0.637 | 0.883 | 0.876 | 0.904 | 0.908 | **0.921** | 0.874 |

Table 7: Remaining data sets (without those of Table 2). Random Forest (max_depth=∞) ROC AUC for different rebalancing strategies and different data sets. Only datasets such that the None strategy is on par with the best strategies are displayed. Other data sets are presented in Table 6. Mean standard deviations are computed. The best strategy is highlighted in bold for each data set.

| Strategy | None | CW | RUS | ROS | Near Miss1 | BS1 | BS2 | SMOTE | CV SMOTE | MGS $(d+1)$ |
|---|---|---|---|---|---|---|---|---|---|---|
| Phoneme (±0.001) | 0.961 | 0.962 | 0.951 | **0.963** | 0.909 | 0.961 | 0.961 | 0.962 | 0.961 | 0.959 |
| *Phoneme* (20%) (±0.002) | 0.952 | 0.952 | 0.935 | **0.953** | 0.793 | 0.950 | 0.951 | **0.953** | **0.953** | 0.949 |
| *Phoneme* (10%) (±0.004) | **0.937** | 0.936 | 0.911 | **0.937** | 0.668 | 0.933 | 0.932 | 0.935 | 0.915 | 0.933 |
| Pima (±0.005) | **0.824** | **0.824** | 0.823 | 0.821 | 0.808 | 0.813 | 0.812 | 0.820 | 0.821 | 0.822 |
| Yeast (±0.003) | 0.965 | 0.969 | **0.970** | 0.968 | 0.919 | 0.964 | 0.967 | 0.967 | 0.968 | 0.966 |
| Haberman (±0.017) | 0.674 | 0.674 | 0.675 | 0.672 | **0.691** | 0.678 | 0.668 | 0.684 | 0.680 | 0.679 |
| *California* (20%)(±0.001) | **0.956** | 0.955 | 0.951 | **0.956** | 0.850 | 0.954 | 0.947 | 0.955 | 0.954 | 0.954 |
| *California* (10%)(±0.001) | **0.948** | 0.947 | 0.939 | **0.948** | 0.775 | 0.945 | 0.935 | 0.947 | 0.947 | **0.948** |
| House_16H (±0.001) | **0.951** | 0.950 | 0.948 | 0.950 | 0.900 | 0.945 | 0.942 | 0.948 | 0.948 | 0.948 |
| *House_16H* (20%)(±0.001) | **0.950** | 0.949 | 0.946 | 0.949 | 0.835 | 0.943 | 0.938 | 0.946 | 0.945 | 0.946 |
| *House_16H* (10%)(±0.001) | **0.945** | 0.943 | 0.941 | 0.944 | 0.718 | 0.939 | 0.931 | 0.939 | 0.939 | 0.937 |
| *House_16H* (1%)(±0.001) | 0.888 | 0.875 | **0.902** | 0.880 | 0.599 | 0.893 | 0.898 | 0.899 | 0.896 | 0.890 |
| Vehicle (±0.001) | **0.995** | 0.994 | 0.990 | **0.995** | 0.977 | 0.994 | 0.994 | 0.994 | 0.994 | **0.995** |
| Vehicle (10%) (±0.003) | 0.992 | 0.992 | 0.983 | 0.991 | 0.867 | 0.991 | 0.989 | 0.992 | 0.992 | **0.993** |
| Ionosphere (±0.003) | 0.978 | 0.978 | 0.974 | 0.978 | 0.946 | 0.978 | **0.979** | **0.979** | **0.979** | 0.976 |
| *Ionosphere* (20%) (±0.004) | **0.989** | 0.987 | 0.977 | 0.988 | 0.883 | 0.982 | 0.974 | 0.981 | 0.982 | 0.985 |
| *Ionosphere* (10%) (±0.008) | **0.989** | 0.983 | 0.946 | 0.982 | 0.825 | 0.973 | 0.961 | 0.965 | 0.965 | 0.967 |
| Breast Cancer (±0.001) | **0.994** | 0.993 | 0.993 | 0.993 | **0.994** | 0.992 | 0.992 | 0.993 | 0.993 | 0.993 |
| *Breast Cancer* (20%)(±0.001) | **0.996** | **0.996** | 0.994 | **0.996** | **0.996** | 0.994 | 0.993 | 0.995 | **0.996** | **0.996** |
| *Breast Cancer* (10%)(±0.001) | **0.997** | 0.996 | 0.994 | 0.996 | **0.997** | 0.994 | 0.993 | 0.996 | 0.996 | **0.997** |

Table 8: Highly imbalanced data sets. ROC AUC Random Forest with max_depth=5.

| Strategy | None | CW | RUS | ROS | Near Miss1 | BS1 | BS2 | SMOTE | CV SMOTE | MGS $(d+1)$ |
|---|---|---|---|---|---|---|---|---|---|---|
| CreditCard (0.2%)(±0.002) | 0.954 | 0.970 | 0.970 | 0.971 | 0.898 | 0.960 | 0.962 | **0.971** | **0.971** | 0.964 |
| Abalone (1%)(±0.017) | 0.775 | 0.756 | 0.735 | 0.731 | 0.653 | 0.760 | 0.754 | 0.744 | 0.757 | **0.780** |
| *Phoneme* (1%)(±0.012) | **0.891** | 0.871 | 0.870 | 0.867 | 0.697 | 0.865 | 0.851 | 0.882 | 0.878 | 0.886 |
| *Yeast* (1%)(±0.023) | 0.923 | 0.921 | 0.934 | 0.887 | 0.709 | 0.933 | 0.922 | **0.945** | 0.940 | 0.944 |
| Wine (4%)(±0.005) | 0.900 | 0.905 | 0.900 | **0.907** | 0.587 | 0.895 | 0.880 | 0.902 | 0.899 | 0.885 |
| *Pima* (20%) (±0.007) | 0.802 | 0.811 | 0.805 | **0.809** | 0.778 | 0.804 | 0.805 | 0.805 | 0.806 | 0.804 |
| *Haberman* (10%)(±0.029) | 0.714 | 0.722 | 0.708 | 0.723 | 0.699 | 0.749 | 0.738 | 0.751 | 0.750 | **0.759** |
| *MagicTel* (20%) (±0.001) | **0.893** | 0.892 | 0.893 | 0.891 | 0.604 | 0.888 | 0.874 | 0.891 | 0.891 | 0.885 |
| *California* (1%)(±0.008) | **0.880** | 0.877 | 0.875 | 0.874 | 0.631 | 0.852 | 0.838 | 0.867 | 0.866 | 0.878 |

Table 9: Highly imbalanced data sets. ROC AUC Random Forest with max_depth=RUS. On the last column, the value of maximal depth when using Random forest (max_depth=$\infty$) with RUS strategy for each data set.

| Strategy | None | CW | RUS | ROS | Near Miss1 | BS1 | BS2 | SMOTE | CV SMOTE | MGS $(d+1)$ | depth |
|---|---|---|---|---|---|---|---|---|---|---|---|
| CreditCard $(0.2\%)(\pm 0.002)$ | 0.954 | 0.950 | 0.970 | 0.970 | 0.893 | 0.960 | 0.962 | **0.972** | **0.972** | 0.962 | 10 |
| Abalone $(1\%)(\pm 0.017)$ | 0.770 | 0.750 | 0.733 | 0.729 | 0.656 | 0.762 | 0.758 | 0.744 | 0.761 | **0.795** | 7 |
| *Phoneme* $(1\%)(\pm 0.014)$ | **0.897** | 0.874 | 0.872 | 0.869 | 0.695 | 0.869 | 0.858 | 0.887 | 0.880 | 0.894 | 6 |
| *Yeast* $(1\%)(\pm 0.021)$ | 0.928 | 0.927 | 0.928 | 0.893 | 0.725 | 0.924 | 0.919 | 0.934 | 0.925 | **0.945** | 3 |
| Wine $(4\%)(\pm 0.006)$ | **0.927** | 0.922 | 0.915 | 0.925 | 0.665 | 0.923 | 0.913 | 0.923 | 0.925 | 0.923 | 10 |
| *Pima* $(20\%)(\pm 0.009)$ | 0.784 | 0.797 | 0.793 | 0.790 | 0.768 | 0.792 | 0.789 | 0.792 | 0.792 | 0.790 | 10 |
| *Haberman* $(10\%)(\pm 0.028)$ | 0.696 | 0.711 | 0.713 | 0.721 | 0.690 | 0.737 | 0.729 | 0.740 | 0.748 | **0.752** | 7 |
| *MagicTel* $(20\%)(\pm 0.001)$ | 0.917 | 0.920 | 0.917 | 0.921 | 0.651 | 0.919 | 0.905 | **0.921** | **0.921** | 0.913 | 20 |
| *California* $(1\%)(\pm 0.009)$ | 0.895 | 0.871 | 0.877 | 0.875 | 0.639 | 0.876 | 0.859 | 0.884 | 0.903 | **0.913** | 10 |

Table 10: Table 2 with standard deviations over 100 runs. Random Forest (max_depth=tuned with ROC AUC) ROC AUC for different rebalancing strategies and different data sets. The best strategies are displayed in bold are displayed.

| Strategy | None | CW | RUS | ROS | Near Miss1 | BS1 | BS2 | SMOTE | CV SMOTE | MGS $(d+1)$ |
|---|---|---|---|---|---|---|---|---|---|---|
| CreditCard $(0.2\%)$ | 0.966 | 0.967 | **0.970** | 0.935 | 0.892 | 0.949 | 0.944 | 0.947 | 0.954 | 0.952 |
| std | $\pm 0.003$ | $\pm 0.003$ | $\pm 0.003$ | $\pm 0.003$ | $\pm 0.005$ | $\pm 0.005$ | $\pm 0.006$ | $\pm 0.004$ | $\pm 0.003$ | $\pm 0.003$ |
| Abalone $(1\%)$ | 0.764 | 0.748 | 0.735 | 0.722 | 0.656 | 0.744 | 0.753 | 0.741 | 0.791 | **0.802** |
| std | $\pm 0.021$ | $\pm 0.021$ | $\pm 0.021$ | $\pm 0.021$ | $\pm 0.033$ | $\pm 0.025$ | $\pm 0.019$ | $\pm 0.019$ | $\pm 0.018$ | $\pm 0.012$ |
| *Phoneme* $(1\%)$ | 0.897 | 0.868 | 0.868 | 0.858 | 0.698 | 0.867 | 0.869 | 0.888 | **0.924** | 0.915 |
| std | $\pm 0.015$ | $\pm 0.018$ | $\pm 0.015$ | $\pm 0.02$ | $\pm 0.030$ | $\pm 0.026$ | $\pm 0.023$ | $\pm 0.020$ | $\pm 0.014$ | $\pm 0.013$ |
| *Yeast* $(1\%)$ | 0.925 | 0.920 | 0.938 | 0.908 | 0.716 | 0.949 | 0.954 | **0.955** | 0.942 | 0.945 |
| std | $\pm 0.017$ | $\pm 0.030$ | $\pm 0.026$ | $\pm 0.021$ | $\pm 0.069$ | $\pm 0.0220$ | $\pm 0.009$ | $\pm 0.016$ | $\pm 0.021$ | $\pm 0.018$ |
| Wine $(4\%)$ | 0.928 | 0.925 | 0.915 | 0.924 | 0.682 | 0.933 | 0.927 | 0.934 | 0.938 | **0.941** |
| std | $\pm 0.007$ | $\pm 0.008$ | $\pm 0.005$ | $\pm 0.008$ | $\pm 0.013$ | $\pm 0.007$ | $\pm 0.008$ | $\pm 0.006$ | $\pm 0.006$ | $\pm 0.005$ |
| *Pima* $(20\%)$ | 0.798 | **0.808** | 0.799 | 0.790 | 0.777 | 0.793 | 0.788 | 0.789 | 0.787 | 0.787 |
| std | $\pm 0.009$ | $\pm 0.008$ | $\pm 0.010$ | $\pm 0.009$ | $\pm 0.007$ | $\pm 0.009$ | $\pm 0.008$ | $\pm 0.008$ | $\pm 0.007$ | $\pm 0.008$ |
| *Haberman* $(10\%)$ | 0.708 | 0.709 | 0.720 | 0.704 | 0.697 | 0.723 | 0.721 | 0.719 | 0.742 | **0.744** |
| std | $\pm 0.027$ | $\pm 0.029$ | $\pm 0.040$ | $\pm 0.024$ | $\pm 0.038$ | $\pm 0.027$ | $\pm 0.027$ | $\pm 0.024$ | $\pm 0.022$ | $\pm 0.026$ |
| *MagicTel* $(20\%)$ | 0.917 | 0.921 | 0.917 | **0.922** | 0.649 | 0.920 | 0.905 | 0.921 | 0.919 | 0.913 |
| std | $\pm 0.001$ | $\pm 0.001$ | $\pm 0.001$ | $\pm 0.001$ | $\pm 0.005$ | $\pm 0.001$ | $\pm 0.002$ | $\pm 0.001$ | $\pm 0.001$ | $\pm 0.001$ |
| *California* $(1\%)$ | 0.887 | 0.877 | 0.880 | 0.883 | 0.630 | 0.885 | 0.874 | 0.906 | 0.916 | **0.923** |
| std | $\pm 0.010$ | $\pm 0.013$ | $\pm 0.010$ | $\pm 0.011$ | $\pm 0.012$ | $\pm 0.014$ | $\pm 0.013$ | $\pm 0.011$ | $\pm 0.007$ | $\pm 0.006$ |
| *House_16H* $(1\%)$ | **0.906** | 0.893 | 0.902 | 0.885 | 0.600 | 0.894 | 0.896 | 0.898 | 0.905 | 0.889 |
| std | $\pm 0.006$ | $\pm 0.006$ | $\pm 0.006$ | $\pm 0.007$ | $\pm 0.018$ | $\pm 0.008$ | $\pm 0.006$ | $\pm 0.006$ | $\pm 0.005$ | $\pm 0.005$ |

Table 11: Table 5 with standard deviations over 100 runs. Random Forest (max_depth=tuned with ROC AUC) ROC AUC for different rebalancing strategies and different data sets. The best strategies are displayed in bold.

| Strategy | None | CW | RUS | ROS | Near Miss1 | BS1 | BS2 | SMOTE | CV SMOTE | MGS $(d+1)$ |
|---|---|---|---|---|---|---|---|---|---|---|
| Phoneme | **0.962** | 0.961 | 0.951 | **0.962** | 0.910 | 0.960 | 0.961 | **0.962** | 0.961 | 0.959 |
| std | ±0.001 | ±0.001 | ±0.001 | ±0.001 | ±0.003 | ±0.001 | ±0.001 | ±0.001 | ±0.001 | ±0.001 |
| *Phoneme* (20%) | 0.952 | 0.952 | 0.935 | **0.953** | 0.793 | 0.950 | 0.951 | **0.953** | 0.953 | 0.949 |
| std | ±0.001 | ±0.001 | ±0.002 | ±0.001 | ±0.014 | ±0.001 | ±0.001 | ±0.001 | ±0.001 | ±0.001 |
| *Phoneme* (10%) | **0.936** | 0.935 | 0.909 | **0.936** | 0.664 | 0.933 | 0.932 | 0.935 | 0.938 | 0.932 |
| std | ±0.003 | ±0.003 | ±0.005 | ±0.003 | ±0.013 | ±0.003 | ±0.004 | ±0.003 | ±0.003 | ±0.003 |
| Pima | **0.833** | 0.832 | 0.828 | 0.823 | 0.817 | 0.814 | 0.811 | 0.820 | 0.824 | 0.826 |
| std | ±0.004 | ±0.004 | ±0.004 | ±0.005 | ±0.004 | ±0.005 | ±0.005 | ±0.007 | ±0.006 | ±0.006 |
| Yeast | 0.968 | **0.971** | **0.971** | 0.968 | 0.921 | 0.964 | 0.965 | 0.968 | 0.969 | 0.968 |
| std | ±0.003 | ±0.002 | ±0.002 | ±0.004 | ±0.005 | ±0.004 | ±0.003 | ±0.004 | ±0.004 | ±0.003 |
| Haberman | 0.686 | 0.686 | 0.685 | 0.673 | 0.686 | 0.682 | 0.670 | 0.681 | 0.690 | **0.698** |
| std | ±0.020 | ±0.015 | ±0.025 | ±0.015 | ±0.012 | ±0.016 | ±0.014 | ±0.012 | ±0.015 | ±0.014 |
| *California* (20%) | **0.956** | 0.955 | 0.951 | **0.956** | 0.850 | 0.953 | 0.947 | 0.955 | **0.956** | 0.954 |
| std | ±0.001 | ±0.001 | ±0.001 | ±0.001 | ±0.002 | ±0.001 | ±0.001 | ±0.001 | ±0.001 | ±0.001 |
| *California* (10%) | 0.948 | 0.946 | 0.940 | 0.948 | 0.775 | 0.945 | 0.934 | 0.947 | **0.950** | 0.948 |
| std | ±0.002 | ±0.002 | ±0.002 | ±0.001 | ±0.004 | ±0.001 | ±0.002 | ±0.001 | ±0.001 | ±0.001 |
| House_16H | **0.950** | **0.950** | 0.948 | **0.950** | 0.899 | 0.945 | 0.942 | 0.948 | 0.949 | 0.948 |
| std | ±0.001 | ±0.001 | ±0.001 | ±0.001 | ±0.001 | ±0.001 | ±0.001 | ±0.001 | ±0.001 | ±0.001 |
| *House_16H* (20%) | **0.950** | 0.949 | 0.946 | 0.949 | 0.835 | 0.943 | 0.938 | 0.946 | 0.947 | 0.946 |
| std | ±0.001 | ±0.001 | ±0.001 | ±0.001 | ±0.001 | ±0.001 | ±0.001 | ±0.001 | ±0.001 | ±0.001 |
| *House_16H* (10%) | **0.945** | 0.943 | 0.940 | 0.944 | 0.717 | 0.939 | 0.931 | 0.939 | 0.942 | 0.937 |
| std | ±0.001 | ±0.001 | ±0.001 | ±0.001 | ±0.003 | ±0.001 | ±0.001 | ±0.001 | ±0.001 | ±0.001 |
| *House_16H* (1%) | **0.906** | 0.893 | 0.902 | 0.885 | 0.600 | 0.894 | 0.896 | 0.898 | 0.905 | 0.889 |
| std | ±0.006 | ±0.006 | ±0.006 | ±0.007 | ±0.018 | ±0.008 | ±0.006 | ±0.006 | ±0.005 | ±0.005 |
| Vehicle | **0.995** | 0.994 | 0.990 | 0.994 | 0.978 | 0.994 | 0.993 | 0.994 | **0.995** | **0.995** |
| std | ±0.001 | ±0.001 | ±0.001 | ±0.001 | ±0.003 | ±0.001 | ±0.001 | ±0.001 | ±0.001 | ±0.001 |
| Vehicle (10%) | 0.992 | 0.991 | 0.982 | 0.989 | 0.863 | 0.991 | 0.989 | 0.992 | 0.993 | **0.994** |
| std | ±0.002 | ±0.002 | ±0.005 | ±0.002 | ±0.010 | ±0.002 | ±0.003 | ±0.002 | ±0.001 | ±0.001 |
| Ionosphere | **0.978** | **0.978** | 0.974 | **0.978** | 0.945 | **0.978** | **0.978** | **0.978** | 0.977 | 0.976 |
| std | ±0.003 | ±0.003 | ±0.003 | ±0.003 | ±0.003 | ±0.003 | ±0.003 | ±0.003 | ±0.002 | ±0.002 |
| *Ionosphere* (20%) | **0.988** | 0.986 | 0.974 | 0.987 | 0.881 | 0.981 | 0.974 | 0.981 | 0.983 | 0.983 |
| std | ±0.002 | ±0.003 | ±0.005 | ±0.002 | ±0.013 | ±0.003 | ±0.004 | ±0.003 | ±0.004 | ±0.003 |
| *Ionosphere* (10%) | **0.988** | 0.983 | 0.944 | 0.981 | 0.822 | 0.972 | 0.962 | 0.966 | 0.967 | 0.968 |
| std | ±0.004 | ±0.005 | ±0.016 | ±0.005 | ±0.026 | ±0.007 | ±0.005 | ±0.005 | ±0.006 | ±0.006 |
| Breast Cancer | **0.994** | 0.993 | 0.993 | 0.993 | **0.994** | 0.992 | 0.992 | 0.993 | **0.994** | 0.993 |
| std | ±0.001 | ±0.001 | ±0.001 | ±0.001 | ±0.001 | ±0.001 | ±0.001 | ±0.001 | ±0.001 | ±0.001 |
| *Breast Cancer* (20%) | 0.996 | 0.995 | 0.994 | 0.995 | **0.997** | 0.994 | 0.993 | 0.995 | 0.996 | 0.996 |
| std | ±0.001 | ±0.001 | ±0.001 | ±0.001 | ±0.001 | ±0.002 | ±0.001 | ±0.001 | ±0.001 | ±0.001 |
| *Breast Cancer* (10%) | **0.997** | 0.996 | 0.994 | 0.996 | **0.997** | 0.993 | 0.992 | 0.996 | **0.997** | **0.997** |
| std | ±0.001 | ±0.001 | ±0.001 | ±0.001 | ±0.001 | ±0.001 | ±0.001 | ±0.001 | ±0.001 | ±0.001 |

Table 12: Highly imbalanced data sets at the top and remaining ones at the bottom. Random Forest (max_depth=∞) ROC AUC. The best strategy is highlighted in bold for each data set.

| SMOTE Strategy | $K = 5$ | $K = \sqrt{n}$ | $K = 0.01n$ | $K = 0.1n$ | CV SMOTE |
|---|---|---|---|---|---|
| CreditCard $(\pm0.004)$ | 0.949 | 0.959 | 0.941 | **0.961** | **0.961** |
| Abalone $(1\%)(\pm0.021)$ | 0.744 | 0.745 | 0.727 | 0.729 | **0.777** |
| *Phoneme* $(1\%)(\pm0.019)$ | 0.883 | 0.880 | 0.872 | 0.871 | **0.893** |
| *Yeast* $(1\%)(\pm0.016)$ | 0.940 | 0.935 | 0.932 | 0.931 | **0.954** |
| Wine $(4\%)(\pm0.006)$ | 0.934 | **0.935** | 0.930 | 0.934 | **0.935** |
| *Pima* $(20\%)(\pm0.008)$ | 0.789 | 0.786 | **0.790** | 0.788 | 0.786 |
| *Haberman* $(10\%)(\pm0.024)$ | 0.721 | 0.723 | 0.715 | 0.725 | **0.735** |
| *MagicTel* $(20\%)(\pm0.001)$ | **0.921** | 0.921 | 0.921 | 0.920 | 0.921 |
| *California* $(1\%)(\pm0.009)$ | 0.904 | 0.905 | 0.893 | 0.905 | **0.908** |
| Phoneme $(\pm0.001)$ | **0.962** | 0.961 | **0.962** | 0.961 | 0.961 |
| *Phoneme* $(20\%)(\pm0.001)$ | **0.953** | 0.952 | **0.953** | 0.952 | **0.953** |
| *Phoneme* $(10\%)(\pm0.003)$ | 0.935 | 0.938 | 0.936 | **0.939** | 0.915 |
| Pima $(\pm0.005)$ | 0.820 | 0.819 | **0.821** | 0.819 | **0.821** |
| Yeast $(\pm0.003)$ | 0.967 | **0.970** | 0.968 | 0.969 | 0.968 |
| Haberman $(\pm0.016)$ | **0.684** | **0.684** | 0.674 | 0.680 | 0.680 |
| *California* $(20\%)(\pm0.001)$ | **0.955** | 0.954 | 0.954 | 0.953 | 0.954 |
| *California* $(10\%)(\pm0.001)$ | **0.947** | **0.947** | **0.947** | 0.946 | **0.947** |
| House_16H $(\pm0.001)$ | **0.948** | 0.947 | 0.947 | 0.947 | **0.948** |
| *House_16H* $(20\%)(\pm0.001)$ | **0.946** | 0.944 | 0.945 | 0.944 | 0.945 |
| *House_16H* $(10\%)(\pm0.001)$ | **0.939** | 0.938 | **0.939** | 0.937 | **0.939** |
| *House_16H* $(1\%)(\pm0.005)$ | **0.899** | 0.898 | 0.896 | 0.898 | 0.896 |
| Vehicle $(\pm0.001)$ | **0.994** | **0.994** | **0.994** | **0.994** | **0.994** |
| Vehicle $(10\%)(\pm0.002)$ | **0.992** | **0.992** | **0.992** | **0.992** | **0.992** |
| Ionosphere $(\pm0.003)$ | **0.979** | 0.977 | 0.978 | 0.978 | **0.979** |
| *Ionosphere* $(20\%)(\pm0.003)$ | 0.981 | 0.981 | **0.984** | 0.982 | 0.982 |
| *Ionosphere* $(10\%)(\pm0.005)$ | **0.965** | 0.964 | **0.965** | 0.966 | **0.965** |
| Breast Cancer $(\pm0.001)$ | **0.993** | **0.993** | **0.993** | **0.993** | **0.993** |
| *Breast Cancer* $(20\%)(\pm0.001)$ | 0.995 | 0.995 | **0.996** | 0.995 | **0.996** |
| *Breast Cancer* $(10\%)(\pm0.001)$ | **0.996** | **0.996** | **0.996** | **0.996** | **0.996** |

Table 13: Random Forest (max_depth=tuned with PR AUC) PR AUC for different rebalancing strategies and different data sets.

| Strategy | None | CW | RUS | ROS | Near Miss1 | BS1 | BS2 | SMOTE | CV SMOTE | MGS $(d+1)$ |
|---|---|---|---|---|---|---|---|---|---|---|
| Abalone $(1\%)$ | 0.048 | **0.055** | 0.047 | 0.049 | 0.022 | 0.045 | 0.041 | 0.039 | **0.055** | 0.035 |
| *Phoneme* $(1\%)$ | 0.198 | 0.215 | 0.081 | 0.146 | 0.054 | 0.226 | 0.236 | 0.236 | **0.260** | 0.101 |
| *Haberman* $(10\%)$ | 0.246 | 0.264 | 0.275 | 0.227 | 0.274 | 0.274 | 0.287 | 0.278 | **0.292** | 0.283 |
| *MagicTel* $(20\%)$ | 0.755 | 0.759 | 0.742 | **0.760** | 0.336 | 0.740 | 0.701 | 0.756 | 0.756 | 0.748 |

Table 14: Random Forest (max_depth=tuned with PR AUC) PR AUC for different rebalancing strategies and different data sets.

| Strategy | None | CW | RUS | ROS | Near Miss1 | BS1 | BS2 | SMOTE | CV SMOTE | MGS $(d+1)$ |
|---|---|---|---|---|---|---|---|---|---|---|
| CreditCard (0.2%) | **0.849** | 0.845 | 0.739 | 0.846 | 0.614 | 0.817 | 0.808 | 0.845 | 0.842 | 0.837 |
| std | 0.003 | 0.003 | 0.005 | 0.003 | 0.051 | 0.006 | 0.009 | 0.003 | 0.000 | 0.002 |
| Phoneme | **0.919** | 0.917 | 0.885 | **0.919** | 0.846 | 0.911 | 0.911 | 0.916 | 0.914 | 0.917 |
| std | 0.003 | 0.002 | 0.005 | 0.003 | 0.005 | 0.004 | 0.003 | 0.004 | 0.004 | 0.003 |
| *Phoneme* (20%) | **0.863** | 0.857 | 0.776 | 0.861 | 0.573 | 0.842 | 0.844 | 0.855 | 0.854 | 0.855 |
| std | 0.004 | 0.005 | 0.007 | 0.005 | 0.022 | 0.006 | 0.005 | 0.004 | 0.005 | 0.005 |
| *Phoneme* (10%) | **0.724** | 0.707 | 0.533 | 0.710 | 0.268 | 0.675 | 0.677 | 0.695 | 0.693 | 0.691 |
| std | 0.011 | 0.012 | 0.016 | 0.012 | 0.018 | 0.015 | 0.011 | 0.010 | 0.014 | 0.012 |
| Yeast | **0.837** | 0.843 | 0.825 | 0.831 | 0.712 | 0.786 | 0.765 | 0.831 | 0.833 | 0.835 |
| std | 0.011 | 0.006 | 0.014 | 0.013 | 0.019 | 0.014 | 0.015 | 0.008 | 0.013 | 0.013 |
| *Yeast* (1%) | 0.351 | 0.304 | 0.208 | 0.269 | 0.126 | 0.354 | 0.301 | 0.351 | **0.373** | 0.316 |
| std | 0.055 | 0.045 | 0.067 | 0.055 | 0.047 | 0.054 | 0.051 | 0.055 | 0.053 | 0.060 |
| Wine (4%) | **0.602** | 0.598 | 0.400 | 0.588 | 0.140 | 0.580 | 0.572 | 0.589 | 0.588 | 0.536 |
| std | 0.023 | 0.027 | 0.018 | 0.028 | 0.017 | 0.024 | 0.024 | 0.025 | 0.025 | 0.027 |
| Pima | **0.718** | 0.709 | 0.703 | 0.696 | 0.701 | 0.673 | 0.672 | 0.689 | 0.687 | 0.710 |
| std | 0.011 | 0.008 | 0.011 | 0.011 | 0.009 | 0.016 | 0.011 | 0.015 | 0.013 | 0.009 |
| *Pima* (20%) | **0.525** | 0.522 | 0.514 | 0.498 | 0.490 | 0.476 | 0.465 | 0.484 | 0.482 | 0.516 |
| std | 0.016 | 0.020 | 0.024 | 0.019 | 0.013 | 0.019 | 0.017 | 0.017 | 0.015 | 0.015 |
| Haberman | 0.465 | 0.479 | 0.457 | 0.411 | 0.468 | 0.417 | 0.421 | 0.431 | 0.436 | **0.483** |
| std | 0.029 | 0.024 | 0.031 | 0.022 | 0.017 | 0.019 | 0.025 | 0.024 | 0.024 | 0.026 |
| *California* (20%) | **0.888** | 0.885 | 0.871 | 0.886 | 0.672 | 0.874 | 0.862 | 0.882 | 0.880 | 0.879 |
| std | 0.001 | 0.001 | 0.001 | 0.002 | 0.004 | 0.002 | 0.002 | 0.002 | 0.002 | 0.002 |
| *California* (10%) | **0.802** | 0.795 | 0.760 | 0.797 | 0.384 | 0.774 | 0.738 | 0.787 | 0.784 | 0.767 |
| std | 0.003 | 0.003 | 0.004 | 0.003 | 0.010 | 0.004 | 0.006 | 0.003 | 0.003 | 0.003 |
| *California* (1%) | **0.297** | 0.290 | 0.208 | 0.210 | 0.019 | 0.227 | 0.210 | 0.249 | 0.267 | 0.196 |
| std | 0.018 | 0.018 | 0.020 | 0.020 | 0.002 | 0.014 | 0.018 | 0.021 | 0.015 | 0.012 |
| House_16H | 0.901 | 0.896 | 0.890 | 0.897 | 0.799 | 0.885 | 0.881 | 0.892 | 0.891 | **0.902** |
| std | 0.001 | 0.001 | 0.001 | 0.001 | 0.002 | 0.001 | 0.001 | 0.001 | .001 | 0.001 |
| House_16H (20%) | 0.856 | 0.847 | 0.832 | 0.847 | 0.578 | 0.827 | 0.814 | 0.837 | 0.836 | **0.857** |
| std | 0.001 | 0.001 | 0.002 | 0.002 | 0.005 | 0.002 | 0.003 | 0.001 | 0.001 | 0.001 |
| *House_16H* (10%) | **0.757** | 0.729 | 0.691 | 0.731 | 0.242 | 0.703 | 0.680 | 0.711 | 0.709 | 0.756 |
| std | 0.003 | 0.002 | 0.006 | 0.003 | 0.008 | 0.003 | 0.006 | 0.003 | 0.003 | 0.002 |
| *House_16H* (1%) | **0.312** | 0.242 | 0.167 | 0.185 | 0.032 | 0.208 | 0.201 | 0.203 | 0.212 | 0.265 |
| std | 0.013 | 0.014 | 0.018 | 0.013 | 0.005 | 0.011 | 0.010 | 0.010 | 0.011 | 0.013 |
| Vehicle | 0.981 | 0.978 | 0.963 | 0.981 | 0.957 | 0.979 | 0.979 | 0.978 | 0.978 | **0.982** |
| std | 0.003 | 0.003 | 0.008 | 0.002 | 0.006 | 0.003 | 0.003 | 0.003 | 0.002 | 0.003 |
| *Vehicle* (10%) | **0.949** | 0.942 | 0.869 | 0.921 | 0.699 | 0.932 | .935 | 0.947 | 0.942 | 0.944 |
| std | 0.010 | 0.009 | 0.028 | 0.014 | 0.034 | 0.010 | 0.012 | 0.009 | 0.008 | 0.009 |
| Ionosphere | **0.971** | 0.970 | 0.965 | **0.971** | 0.932 | 0.968 | 0.970 | 0.968 | 0.967 | 0.969 |
| std | 0.003 | 0.003 | 0.003 | 0.003 | 0.007 | 0.004 | 0.005 | 0.005 | 0.005 | 0.004 |
| *Ionosphere* (20%) | **0.964** | 0.955 | 0.927 | 0.958 | 0.730 | 0.921 | 0.877 | 0.925 | 0.919 | 0.963 |
| std | 0.005 | 0.007 | 0.015 | 0.006 | 0.022 | 0.012 | 0.014 | 0.010 | 0.013 | 0.006 |
| *Ionosphere* (10%) | **0.945** | 0.917 | 0.808 | 0.920 | 0.526 | 0.845 | 0.761 | 0.820 | 0.838 | 0.941 |
| std | 0.017 | 0.019 | 0.065 | 0.020 | 0.065 | 0.028 | 0.031 | 0.033 | 0.031 | 0.017 |
| Breast Cancer | 0.988 | 0.986 | 0.984 | 0.986 | 0.987 | 0.983 | 0.981 | 0.986 | 0.986 | **0.989** |
| std | 0.003 | 0.003 | 0.004 | 0.003 | 0.003 | 0.003 | 0.004 | 0.004 | 0.003 | 0.002 |
| *Breast Cancer* (20%) | 0.984 | 0.980 | 0.968 | 0.979 | 0.984 | 0.971 | 0.967 | 0.978 | 0.980 | **0.985** |
| std | 0.005 | 0.005 | 0.011 | 0.008 | 0.005 | 0.007 | 0.009 | 0.007 | 0.006 | 0.005 |
| *Breast Cancer* (10%) | 0.975 | 0.962 | 0.939 | 0.960 | 0.976 | 0.936 | 0.921 | 0.957 | 0.954 | **0.978** |
| std | 0.008 | 0.009 | 0.014 | 0.009 | 0.009 | 0.016 | 0.014 | 0.015 | 0.015 | 0.006 |

Table 15: Highly imbalanced data sets at the top and remaning ones at the bottom. Logistic Regression ROC AUC. For each data set, the best strategy is highlighted in bold and the mean of the standard deviation is computed (and rounded to $10^{-3}$).

| Strategy | None | CW | RUS | ROS | Near Miss1 | BS1 | BS2 | SMOTE | CV SMOTE | MGS $(d+1)$ |
|---|---|---|---|---|---|---|---|---|---|---|
| CreditCard (±0.001) | 0.951 | 0.953 | **0.963** | 0.951 | 0.888 | 0.903 | 0.919 | 0.946 | 0.955 | 0.926 |
| Abalone (1%)(±0.009) | 0.848 | 0.876 | 0.814 | 0.878 | 0.761 | 0.859 | 0.853 | 0.878 | **0.879** | 0.872 |
| *Phoneme* (1%)(±0.013) | 0.800 | 0.804 | 0.792 | 0.804 | 0.695 | 0.783 | 0.779 | 0.805 | **0.806** | 0.805 |
| *Yeast* (1%)(±0.006) | **0.975** | 0.974 | 0.965 | 0.972 | 0.920 | 0.974 | 0.973 | 0.973 | 0.974 | 0.970 |
| Wine (4%)(±0.003) | 0.836 | **0.840** | 0.835 | 0.839 | 0.576 | 0.837 | 0.831 | 0.838 | 0.839 | 0.833 |
| *Pima* (20%)(±0.005) | **0.821** | 0.820 | 0.813 | 0.819 | 0.797 | 0.818 | 0.820 | 0.819 | 0.819 | 0.818 |
| *Haberman* (10%)(±0.028) | 0.751 | **0.760** | 0.726 | 0.758 | 0.750 | 0.750 | 0.746 | 0.753 | 0.754 | 0.743 |
| *MagicTel* (20%)(±0.001) | **0.844** | 0.841 | 0.841 | 0.841 | 0.490 | 0.815 | 0.814 | 0.841 | 0.842 | 0.838 |
| *California* (1%)(±0.004) | 0.909 | 0.922 | 0.892 | 0.923 | 0.648 | 0.918 | 0.914 | **0.925** | 0.924 | 0.923 |
| Phoneme (±0.001) | **0.813** | 0.811 | 0.811 | 0.811 | 0.576 | 0.801 | 0.805 | 0.810 | 0.812 | 0.808 |
| *Phoneme* (20%)(±0.001) | **0.810** | 0.808 | 0.807 | 0.808 | 0.505 | 0.801 | 0.805 | 0.807 | 0.809 | 0.805 |
| *Phoneme* (10%)(±0.002) | **0.802** | 0.800 | 0.799 | 0.800 | 0.426 | 0.796 | 0.799 | 0.799 | 0.801 | 0.794 |
| Pima (±0.003) | **0.831** | **0.831** | 0.828 | **0.831** | 0.822 | 0.829 | 0.830 | 0.830 | 0.830 | 0.830 |
| Yeast (±0.001) | **0.968** | 0.967 | 0.966 | 0.967 | 0.945 | 0.966 | 0.965 | 0.967 | 0.967 | 0.965 |
| Haberman (±0.019) | 0.674 | 0.678 | 0.672 | 0.674 | **0.702** | 0.663 | 0.661 | 0.674 | 0.670 | 0.674 |
| *California* (20%)(±0.001) | 0.927 | 0.927 | 0.926 | **0.928** | 0.903 | **0.928** | 0.925 | **0.928** | **0.928** | **0.928** |
| *California* (10%)(±0;001) | 0.923 | 0.925 | 0.921 | 0.925 | 0.855 | 0.925 | 0.919 | **0.926** | **0.926** | 0.925 |
| House_16H (±0.001) | 0.886 | **0.889** | **0.889** | **0.889** | 0.867 | 0.888 | 0.888 | **0.889** | **0.889** | **0.889** |
| *House_16H* (20%)(±0.001) | 0.881 | **0.887** | **0.887** | **0.887** | 0.826 | 0.886 | 0.886 | **0.887** | **0.887** | 0.886 |
| *House_16H* (10%)(±0.001) | 0.871 | **0.885** | 0.884 | **0.885** | 0.764 | **0.885** | **0.885** | **0.885** | **0.885** | 0.883 |
| *House_16H* (1%)(±0.006) | 0.822 | **0.862** | 0.856 | **0.862** | 0.694 | 0.849 | 0.854 | 0.861 | 0.860 | 0.848 |
| Vehicle (±0.001) | **0.994** | 0.993 | 0.990 | **0.994** | 0.990 | 0.993 | 0.992 | **0.994** | **0.994** | **0.994** |
| Vehicle (10%)(±0.001) | **0.994** | 0.993 | 0.985 | **0.994** | 0.984 | 0.993 | 0.991 | **0.994** | **0.994** | **0.994** |
| Ionosphere (±0.012) | 0.901 | 0.899 | **0.904** | 0.893 | 0.872 | 0.889 | 0.889 | 0.894 | 0.895 | 0.897 |
| *Ionosphere* (20%)(±0.021) | 0.894 | 0.886 | **0.896** | 0.879 | 0.872 | 0.882 | 0.888 | 0.881 | 0.879 | 0.885 |
| *Ionosphere* (10%)(±0.018) | 0.862 | 0.856 | 0.857 | 0.858 | 0.812 | 0.868 | **0.878** | 0.860 | 0.858 | 0.862 |
| Breast Cancer (±0.001) | **0.996** | **0.996** | **0.996** | **0.996** | **0.996** | **0.996** | **0.996** | **0.996** | **0.996** | **0.996** |
| *Breast Cancer* (20%)(±0.001) | **0.997** | **0.997** | **0.997** | **0.997** | **0.997** | 0.996 | 0.994 | **0.997** | **0.997** | **0.997** |
| *Breast Cancer* (10%)(±0.001) | **0.997** | **0.997** | **0.997** | **0.997** | 0.996 | **0.997** | **0.997** | **0.997** | **0.997** | **0.997** |

Table 16: Highly imbalanced data sets at the top and remaining ones at the bottom. Logistic regression ROC AUC. For each data set, the best strategy is highlighted in bold and the mean of the standard deviation is computed (and rounded to $10^{-3}$).

| SMOTE Strategy | $K = 5$ | $K = \sqrt{n}$ | $K = 0.01n$ | $K = 0.1n$ | CV SMOTE |
|---|---|---|---|---|---|
| CreditCard $(\pm 0.001)$ | 0.946 | 0.947 | 0.947 | 0.949 | **0.955** |
| Abalone $(1\%)(\pm 0.001)$ | 0.878 | 0.878 | **0.881** | 0.877 | 0.879 |
| *Phoneme* $(1\%)(\pm 0.001)$ | 0.805 | 0.805 | **0.806** | **0.806** | **0.806** |
| *Yeast* $(1\%)(\pm 0.001)$ | 0.973 | **0.974** | 0.973 | 0.973 | **0.974** |
| Wine $(4\%)(\pm 0.003)$ | 0.838 | 0.837 | 0.838 | 0.837 | **0.839** |
| *Pima* $(20\%)(\pm 0.005)$ | **0.819** | 0.818 | **0.819** | **0.819** | **0.819** |
| *Haberman* $(10\%)(\pm 0.028)$ | 0.753 | 0.749 | **0.756** | 0.753 | 0.754 |
| *MagicTel* $(20\%)(\pm 0.001)$ | 0.841 | 0.840 | 0.841 | 0.841 | **0.842** |
| *California* $(1\%)(\pm 0.003)$ | **0.925** | **0.925** | **0.925** | **0.925** | 0.924 |
| Phoneme $(\pm 0.001)$ | 0.810 | 0.810 | 0.810 | 0.810 | **0.812** |
| *Phoneme* $(20\%)(\pm 0.01)$ | 0.807 | 0.807 | 0.807 | 0.808 | **0.809** |
| *Phoneme* $(10\%)(\pm 0.001)$ | 0.799 | 0.799 | 0.799 | 0.799 | **0.801** |
| Pima $(\pm 0.003)$ | **0.830** | **0.830** | **0.830** | **0.830** | **0.830** |
| Yeast $(\pm 0.001)$ | **0.967** | **0.967** | **0.967** | **0.967** | **0.967** |
| Haberman $(\pm 0.018)$ | 0.674 | 0.677 | **0.678** | 0.677 | 0.670 |
| *California* $(20\%)(\pm 0.001)$ | **0.928** | **0.928** | **0.928** | **0.928** | **0.928** |
| *California* $(10\%)(\pm 0.001)$ | **0.926** | **0.926** | **0.926** | 0.925 | **0.926** |
| House_16H $(\pm 0.001)$ | **0.889** | **0.889** | **0.889** | **0.889** | **0.889** |
| *House_16H* $(20\%)(\pm 0.001)$ | **0.887** | **0.887** | **0.887** | 0.886 | **0.887** |
| *House_16H* $(10\%)(\pm 0.001)$ | **0.885** | **0.885** | **0.885** | 0.884 | **0.885** |
| *House_16H* $(1\%)(\pm 0.005)$ | **0.861** | 0.860 | 0.859 | 0.859 | 0.860 |
| Vehicle $(\pm 0.001)$ | **0.994** | **0.994** | **0.994** | **0.994** | **0.994** |
| Vehicle $(10\%)(\pm 0.001)$ | **0.994** | **0.994** | **0.994** | **0.994** | **0.994** |
| *Ionosphere* $(\pm 0.011)$ | 0.894 | **0.896** | 0.895 | 0.894 | 0.895 |
| *Ionosphere* $(20\%)(\pm 0.20)$ | **0.881** | **0.881** | 0.879 | 0.880 | 0.879 |
| Ionosphere $(10\%)(\pm 0.017)$ | 0.860 | 0.857 | **0.861** | 0.859 | 0.858 |
| Breast Cancer $(\pm 0.001)$ | **0.996** | **0.996** | **0.996** | **0.996** | **0.996** |
| *Breast Cancer* $(20\%)(\pm 0.001)$ | **0.997** | **0.997** | **0.997** | **0.997** | **0.997** |
| *Breast Cancer* $(10\%)(\pm 0.001)$ | **0.997** | **0.997** | **0.997** | **0.997** | **0.997** |

Table 17: Highly imbalanced data sets ROC AUC. Logistic regression reimplemented in PyTorch using the implementation of Cao et al. (2019).

| Strategy | None | LDAM loss | Focal loss |
|---|---|---|---|
| CreditCard | **0.968** $\pm 0.002$ | 0.934 $\pm 0.003$ | 0.967 $\pm 0.002$ |
| Abalone | 0.790 $\pm 0.008$ | 0.735 $\pm 0.046$ | **0.799** $\pm 0.009$ |
| *Phoneme* $(1\%)$ | 0.806 $\pm 0.008$ | 0.656 $\pm 0.091$ | **0.807** $\pm 0.008$ |
| *Yeast* $(1\%)$ | **0.977** $\pm 0.002$ | 0.942 $\pm 0.002$ | **0.977** $\pm 0.002$ |
| Wine | 0.827 $\pm 0.002$ | 0.675 $\pm 0.087$ | **0.831** $\pm 0.002$ |
| *Pima* $(20\%)$ | **0.821** $\pm 0.005$ | 0.697 $\pm 0.036$ | **0.821** $\pm 0.005$ |
| *Haberman* $(10\%)$ | 0.749 $\pm 0.030$ | 0.611 $\pm 0.077$ | **0.750** $\pm 0.029$ |
| *MagicTel* $(20\%)$ | 0.843 $\pm 0.001$ | 0.785 $\pm 0.20$ | **0.844** $\pm 0.001$ |
| *California* $(1\%)$ | 0.833 $\pm 0.006$ | **0.922** $\pm 0.003$ | 0.841 $\pm 0.007$ |

Table 18: LightGBM PR AUC for different rebalancing strategies and different data sets.

| Strategy | None | CW | RUS | ROS | Near Miss1 | BS1 | BS2 | SMOTE | CV SMOTE | MGS $(d+1)$ |
|---|---|---|---|---|---|---|---|---|---|---|
| CreditCard (0.2%) | 0.276 | **0.772** | 0.729 | 0.757 | 0.334 | 0.627 | 0.620 | 0.724 | 0.720 | 0.731 |
| *Phoneme* (1%) | 0.228 | 0.230 | 0.054 | 0.223 | 0.040 | 0.263 | 0.255 | 0.267 | **0.278** | 0.157 |
| *House_16H* (1%) | 0.343 | 0.362 | 0.180 | 0.356 | 0.023 | 0.330 | 0.344 | 0.312 | 0.321 | **0.367** |

Table 19: LightGBM PR AUC for different rebalancing strategies and different data sets.

| Strategy | None | CW | RUS | ROS | Near Miss1 | BS1 | BS2 | SMOTE | CV SMOTE | MGS $(d+1)$ |
|---|---|---|---|---|---|---|---|---|---|---|
| Abalone (1%) | **0.056** | 0.054 | 0.047 | 0.053 | 0.034 | 0.050 | 0.049 | 0.044 | 0.045 | 0.040 |
| std | 0.016 | 0.012 | 0.015 | 0.012 | 0.008 | 0.011 | 0.010 | 0.008 | 0.008 | 0.006 |
| Phoneme | **0.898** | 0.895 | 0.864 | 0.895 | 0.733 | 0.883 | 0.884 | 0.894 | 0.893 | 0.889 |
| std | 0.003 | 0.003 | 0.004 | 0.003 | 0.014 | 0.003 | 0.005 | 0.003 | 0.003 | 0.003 |
| *Phoneme* (20%) | **0.836** | 0.830 | 0.757 | 0.829 | 0.492 | 0.814 | 0.812 | 0.830 | 0.828 | 0.816 |
| std | 0.003 | 0.004 | 0.008 | 0.006 | 0.024 | 0.007 | 0.006 | 0.004 | 0.006 | 0.005 |
| *Phoneme* (10%) | **0.683** | 0.679 | 0.519 | 0.680 | 0.237 | 0.653 | 0.657 | 0.670 | 0.671 | 0.643 |
| std | 0.012 | 0.011 | 0.018 | 0.013 | 0.017 | 0.012 | 0.011 | 0.013 | 0.011 | 0.014 |
| Yeast | 0.795 | 0.797 | 0.785 | **0.801** | 0.697 | 0.768 | 0.761 | 0.792 | 0.791 | 0.793 |
| std | 0.017 | 0.017 | 0.023 | 0.016 | 0.020 | 0.019 | 0.020 | 0.018 | 0.017 | 0.017 |
| *Yeast* (1%) | 0.296 | 0.299 | 0.010 | 0.296 | 0.010 | 0.330 | 0.293 | **0.337** | 0.334 | 0.322 |
| std | 0.076 | 0.080 | 0.000 | 0.078 | 0.000 | 0.072 | 0.058 | 0.074 | 0.068 | 0.064 |
| Wine (4%) | **0.603** | 0.596 | 0.269 | 0.595 | 0.081 | 0.545 | 0.567 | 0.546 | 0.534 | 0.560 |
| std | 0.026 | 0.024 | 0.019 | 0.026 | 0.010 | 0.022 | 0.025 | 0.027 | 0.022 | 0.028 |
| Pima | 0.666 | 0.666 | 0.665 | 0.672 | **0.673** | 0.651 | 0.658 | 0.667 | 0.667 | 0.672 |
| std | 0.014 | 0.015 | 0.015 | 0.012 | 0.010 | 0.014 | 0.017 | 0.014 | 0.017 | 0.012 |
| *Pima* (20%) | 0.475 | 0.480 | 0.473 | 0.473 | **0.483** | 0.466 | 0.470 | 0.471 | 0.471 | 0.466 |
| std | 0.019 | 0.017 | 0.026 | 0.016 | 0.018 | 0.019 | 0.021 | 0.022 | 0.017 | 0.019 |
| Haberman | 0.433 | 0.434 | **0.481** | 0.410 | 0.493 | 0.422 | 0.423 | 0.425 | 0.429 | 0.418 |
| std | 0.026 | 0.023 | 0.027 | 0.022 | 0.017 | 0.021 | 0.021 | 0.019 | 0.024 | 0.027 |
| *Haberman* (10%) | 0.267 | 0.262 | 0.140 | 0.209 | 0.133 | 0.255 | 0.233 | 0.259 | 0.272 | **0.274** |
| std | 0.029 | 0.035 | 0.028 | 0.031 | 0.029 | 0.035 | 0.029 | 0.033 | 0.030 | 0.039 |
| *MagicTel* (20%) | **0.761** | 0.765 | 0.735 | 0.765 | 0.259 | 0.728 | 0.729 | 0.760 | 0.760 | 0.750 |
| std | 0.003 | 0.004 | 0.005 | 0.004 | 0.008 | 0.006 | 0.005 | 0.004 | 0.004 | 0.004 |
| *California* (20%) | **0.906** | 0.905 | 0.891 | 0.904 | 0.562 | 0.895 | 0.896 | 0.901 | 0.902 | 0.898 |
| std | 0.001 | 0.002 | 0.002 | 0.001 | 0.013 | 0.002 | 0.002 | 0.001 | 0.001 | 0.001 |
| *California* (10%) | 0.830 | **0.831** | 0.786 | 0.830 | 0.314 | 0.816 | 0.818 | 0.823 | 0.823 | 0.810 |
| std | 0.003 | 0.003 | 0.006 | 0.002 | 0.012 | 0.003 | 0.003 | 0.003 | 0.003 | 0.004 |
| *California* (1%) | 0.359 | 0.368 | 0.234 | 0.343 | 0.041 | 0.342 | 0.322 | 0.347 | **0.372** | 0.285 |
| std | 0.019 | 0.019 | 0.028 | 0.023 | 0.010 | 0.017 | 0.018 | 0.017 | 0.019 | 0.020 |
| House_16H | **0.911** | 0.909 | 0.906 | 0.909 | 0.674 | 0.902 | 0.901 | 0.907 | 0.907 | 0.910 |
| std | 0.001 | 0.001 | 0.001 | 0.001 | 0.004 | 0.001 | 0.001 | 0.001 | 0.001 | 0.001 |
| *House_16H* (20%) | **0.869** | 0.867 | 0.857 | 0.866 | 0.417 | 0.855 | 0.854 | 0.862 | 0.861 | 0.868 |
| std | 0.002 | 0.001 | 0.002 | 0.001 | 0.005 | 0.001 | 0.002 | 0.001 | 0.001 | 0.001 |
| *House_16H* (10%) | **0.776** | 0.770 | 0.735 | 0.769 | 0.174 | 0.751 | 0.752 | 0.757 | 0.755 | 0.775 |
| std | 0.002 | 0.003 | 0.006 | 0.002 | 0.004 | 0.003 | 0.003 | 0.003 | 0.003 | 0.002 |
| Vehicle | 0.989 | 0.989 | 0.974 | 0.988 | 0.903 | 0.989 | **0.990** | 0.988 | 0.988 | 0.980 |
| std | 0.003 | 0.003 | 0.008 | 0.003 | 0.011 | 0.003 | 0.003 | 0.003 | 0.002 | 0.004 |
| *Vehicle* (10%) | **0.958** | 0.954 | 0.857 | 0.948 | 0.392 | 0.954 | 0.942 | 0.954 | 0.955 | 0.954 |
| std | 0.012 | 0.011 | 0.033 | 0.013 | 0.060 | 0.012 | 0.012 | 0.011 | 0.011 | 0.010 |
| Ionosphere | **0.968** | 0.967 | 0.958 | 0.965 | 0.937 | 0.962 | 0.967 | 0.963 | 0.963 | 0.967 |
| std | 0.004 | 0.005 | 0.004 | 0.005 | 0.009 | 0.006 | 0.005 | 0.006 | 0.006 | 0.004 |
| *Ionosphere* (20%) | **0.953** | 0.953 | 0.898 | 0.952 | 0.798 | 0.937 | 0.943 | 0.930 | 0.933 | **0.953** |
| std | 0.013 | 0.011 | 0.016 | 0.012 | 0.022 | 0.012 | 0.010 | 0.011 | 0.013 | 0.008 |
| *Ionosphere* (10%) | 0.895 | 0.895 | 0.456 | 0.882 | 0.431 | 0.865 | 0.868 | 0.827 | 0.840 | **0.903** |
| std | 0.027 | 0.024 | 0.080 | 0.024 | 0.085 | 0.027 | 0.027 | 0.026 | 0.028 | 0.019 |
| Breast cancer | 0.989 | 0.990 | 0.986 | 0.989 | 0.987 | 0.988 | 0.987 | 0.990 | 0.989 | **0.991** |
| std | 0.003 | 0.002 | 0.004 | 0.002 | 0.004 | 0.002 | 0.003 | 0.002 | 0.003 | 0.002 |
| *Breast cancer* (20%) | 0.980 | 0.981 | 0.974 | 0.980 | 0.981 | 0.976 | 0.972 | 0.979 | 0.977 | **0.984** |
| std | 0.005 | 0.005 | 0.008 | 0.007 | 0.005 | 0.008 | 0.008 | 0.006 | 0.007 | 0.005 |
| *Breast cancer* (10%) | 0.972 | 0.973 | 0.926 | 0.973 | 0.967 | 0.965 | 0.962 | 0.973 | 0.970 | **0.975** |
| std | 0.011 | 0.010 | 0.019 | 0.010 | 0.008 | 0.014 | 0.015 | 0.011 | 0.013 | 0.010 |

Table 20: LightGBM ROC AUC for different rebalancing strategies and different data sets.

| Strategy | None | CW | RUS | ROS | Near Miss1 | BS1 | BS2 | SMOTE | CV SMOTE | MGS $(d+1)$ |
|---|---|---|---|---|---|---|---|---|---|---|
| CreditCard (0.2%) | 0.761 | 0.938 | **0.970** | 0.921 | 0.879 | 0.941 | 0.932 | 0.937 | 0.950 | 0.956 |
| std | 0.000 | 0.000 | 0.000 | 0.000 | 0.000 | 0.000 | 0.000 | 0.017 | 0.000 | 0.002 |
| Abalone (1%) | 0.738 | 0.738 | 0.726 | 0.738 | 0.700 | 0.750 | 0.757 | 0.748 | **0.775** | 0.745 |
| std | 0.029 | 0.029 | 0.018 | 0.023 | 0.030 | 0.019 | 0.021 | 0.020 | 0.015 | 0.016 |
| Phoneme | **0.954** | 0.953 | 0.943 | 0.953 | 0.863 | 0.949 | 0.950 | 0.952 | 0.952 | 0.951 |
| std | 0.001 | 0.001 | 0.001 | 0.001 | 0.006 | 0.001 | 0.002 | 0.001 | 0.001 | 0.001 |
| Phoneme (20%) | **0.946** | 0.945 | 0.929 | 0.944 | 0.761 | 0.942 | 0.942 | 0.945 | 0.942 | 0.942 |
| std | 0.001 | 0.001 | 0.002 | 0.002 | 0.014 | 0.002 | 0.002 | 0.001 | 0.002 | 0.001 |
| Phoneme (10%) | **0.930** | 0.928 | 0.907 | 0.929 | 0.628 | 0.923 | 0.926 | 0.925 | 0.926 | 0.925 |
| std | 0.003 | 0.003 | 0.005 | 0.004 | 0.014 | 0.003 | 0.003 | 0.004 | 0.003 | 0.003 |
| Phoneme (1%) | **0.898** | 0.878 | 0.828 | 0.836 | 0.706 | 0.889 | 0.883 | 0.895 | 0.885 | 0.888 |
| std | 0.014 | 0.022 | 0.021 | 0.028 | 0.042 | 0.013 | 0.022 | 0.014 | 0.017 | 0.018 |
| Yeast | 0.966 | 0.966 | 0.966 | **0.968** | 0.923 | **0.968** | **0.968** | **0.968** | **0.968** | 0.967 |
| std | 0.003 | 0.003 | 0.004 | 0.003 | 0.006 | 0.002 | 0.003 | 0.002 | 0.002 | 0.003 |
| Yeast (1%) | 0.930 | **0.933** | 0.500 | 0.847 | 0.500 | 0.927 | 0.928 | 0.927 | 0.923 | 0.915 |
| std | 0.025 | 0.023 | 0.000 | 0.069 | 0.000 | 0.025 | 0.027 | 0.025 | 0.022 | 0.028 |
| Wine (4%) | **0.927** | 0.922 | 0.906 | 0.918 | 0.682 | 0.920 | 0.924 | 0.920 | 0.920 | 0.923 |
| std | 0.006 | 0.008 | 0.007 | 0.010 | 0.013 | 0.008 | 0.007 | 0.008 | 0.006 | 0.008 |
| Pima | 0.803 | 0.802 | 0.798 | 0.806 | 0.789 | 0.800 | 0.801 | 0.804 | 0.805 | **0.807** |
| std | 0.008 | 0.007 | 0.008 | 0.007 | 0.006 | 0.008 | 0.008 | 0.009 | 0.009 | 0.008 |
| Pima (20%) | 0.773 | 0.772 | 0.772 | 0.772 | 0.762 | 0.782 | **0.784** | 0.780 | 0.780 | 0.771 |
| std | 0.011 | 0.010 | 0.014 | 0.009 | 0.009 | 0.012 | 0.010 | 0.012 | 0.010 | 0.010 |
| Haberman | 0.684 | 0.687 | **0.707** | 0.668 | 0.707 | 0.680 | 0.677 | 0.685 | 0.681 | 0.666 |
| std | 0.018 | 0.018 | 0.018 | 0.018 | 0.013 | 0.019 | 0.019 | 0.017 | 0.017 | 0.019 |
| Haberman (10%) | 0.691 | 0.689 | 0.575 | 0.643 | 0.564 | 0.710 | 0.674 | 0.712 | 0.726 | **0.729** |
| std | 0.031 | 0.034 | 0.063 | 0.030 | 0.052 | 0.030 | 0.040 | 0.026 | 0.030 | 0.025 |
| MagicTel (20%) | 0.923 | **0.925** | 0.917 | 0.924 | 0.622 | 0.918 | 0.918 | 0.922 | 0.922 | 0.917 |
| std | 0.001 | 0.001 | 0.001 | 0.001 | 0.004 | 0.001 | 0.001 | 0.001 | 0.001 | 0.001 |
| California (20%) | **0.964** | 0.963 | 0.960 | 0.963 | 0.833 | 0.960 | 0.961 | 0.962 | 0.962 | 0.960 |
| std | 0.001 | 0.001 | 0.001 | 0.001 | 0.004 | 0.001 | 0.001 | 0.001 | 0.001 | 0.001 |
| California (10%) | **0.957** | **0.957** | 0.949 | 0.956 | 0.771 | 0.954 | 0.954 | 0.955 | 0.955 | 0.949 |
| std | 0.001 | 0.001 | 0.002 | 0.001 | 0.005 | 0.001 | 0.001 | 0.001 | 0.001 | 0.002 |
| California (1%) | 0.911 | 0.908 | 0.890 | 0.906 | 0.663 | 0.910 | 0.907 | 0.907 | **0.912** | 0.884 |
| std | 0.007 | 0.006 | 0.010 | 0.007 | 0.017 | 0.009 | 0.009 | 0.007 | 0.005 | 0.012 |
| House_16H | **0.954** | **0.954** | 0.953 | 0.953 | 0.874 | 0.950 | 0.950 | 0.953 | 0.953 | **0.954** |
| std | 0.000 | 0.000 | 0.001 | 0.000 | 0.001 | 0.000 | 0.000 | 0.000 | 0.000 | 0.000 |
| House_16H (20%) | **0.953** | **0.953** | 0.951 | **0.953** | 0.794 | 0.949 | 0.949 | 0.951 | 0.951 | **0.953** |
| std | 0.000 | 0.000 | 0.001 | 0.000 | 0.002 | 0.000 | 0.000 | 0.000 | 0.000 | 0.00 |
| House_16H (10%) | **0.950** | 0.949 | 0.945 | 0.949 | 0.686 | 0.945 | 0.946 | 0.945 | 0.944 | **0.950** |
| std | 0.001 | 0.001 | 0.001 | 0.001 | 0.003 | 0.001 | 0.001 | 0.001 | 0.001 | 0.001 |
| House_16H (1%) | 0.903 | 0.896 | 0.899 | 0.896 | 0.605 | 0.907 | 0.909 | 0.894 | 0.894 | **0.912** |
| std | 0.005 | 0.007 | 0.005 | 0.006 | 0.013 | 0.005 | 0.006 | 0.005 | 0.004 | 0.005 |
| Vehicle | 0.996 | 0.996 | 0.992 | 0.996 | 0.949 | 0.996 | **0.997** | 0.996 | 0.996 | 0.996 |
| std | 0.001 | 0.001 | 0.002 | 0.001 | 0.006 | 0.001 | 0.001 | 0.001 | 0.001 | 0.001 |
| Vehicle (10%) | **0.993** | 0.992 | 0.978 | 0.990 | 0.794 | **0.993** | 0.991 | 0.992 | 0.992 | 0.992 |
| std | 0.003 | 0.004 | 0.005 | 0.008 | 0.020 | 0.003 | 0.003 | 0.006 | 0.005 | 0.002 |
| Ionosphere | 0.973 | 0.973 | 0.967 | 0.972 | 0.944 | 0.971 | **0.975** | 0.972 | 0.972 | 0.974 |
| std | 0.004 | 0.005 | 0.005 | 0.004 | 0.006 | 0.005 | 0.004 | 0.005 | 0.004 | 0.005 |
| Ionosphere (20%) | **0.981** | 0.980 | 0.962 | 0.980 | 0.887 | 0.978 | 0.980 | 0.974 | 0.975 | **0.981** |
| std | 0.006 | 0.005 | 0.007 | 0.007 | 0.012 | 0.006 | 0.006 | 0.005 | 0.007 | 0.004 |
| Ionosphere (10%) | 0.972 | 0.972 | 0.777 | 0.959 | 0.753 | 0.954 | 0.951 | 0.927 | 0.946 | **0.975** |
| std | 0.010 | 0.013 | 0.048 | 0.017 | 0.057 | 0.013 | 0.013 | 0.015 | 0.013 | 0.007 |
| Breast cancer | **0.995** | **0.995** | 0.994 | 0.995 | 0.994 | 0.994 | 0.994 | **0.995** | **0.995** | **0.995** |
| std | 0.001 | 0.001 | 0.001 | 0.001 | 0.001 | 0.001 | 0.001 | 0.001 | 0.001 | 0.001 |
| Breast cancer (20%) | 0.996 | 0.996 | 0.994 | 0.996 | 0.996 | 0.995 | 0.995 | 0.996 | 0.996 | **0.997** |
| std | 0.001 | 0.001 | 0.001 | 0.001 | 0.001 | 0.001 | 0.001 | 0.001 | 0.001 | 0.001 |

## B  MAIN PROOFS

This section contains the main proof of our theoretical results. The technicals lemmas used by several proofs are available on Appendix C.

### B.1  PROOF OF LEMMA 3.1

*Proof of Lemma 3.1.* Let $\mathcal{X}$ be the support of $P_X$. SMOTE generates new points by linear interpolation of the original minority sample. This means that for all $x, y$ in the minority samples or generated by SMOTE procedure, we have $(1-t)x + ty \in Conv(\mathcal{X})$ by definition of $Conv(\mathcal{X})$. This leads to the fact that precisely, all the new SMOTE samples are contained in $Conv(\mathcal{X})$. This implies $Supp(P_Z) \subseteq Conv(\mathcal{X})$.

$\square$

### B.2  PROOF OF THEOREM 3.2

*Proof of Theorem 3.2.* For any event $A, B$, we have

$$1 - \mathbb{P}[A \cap B] = \mathbb{P}[A^c \cup B^c] \leq \mathbb{P}[A^c] + \mathbb{P}[B^c], \tag{8}$$

which leads to

$$\mathbb{P}[A \cap B] \geq 1 - \mathbb{P}[A^c] - \mathbb{P}[B^c] \tag{9}$$
$$= \mathbb{P}[A] - \mathbb{P}[B^c]. \tag{10}$$

By construction,

$$\|X_c - Z\| \leq \|X_c - X_{(K)}(X_c)\|. \tag{11}$$

Let $x \in \mathcal{X}$ and $\eta > 0$. Let $\alpha, \varepsilon > 0$. We have,

$$\mathbb{P}[X_c \in B(x, \alpha - \varepsilon)] - \mathbb{P}[\|X_c - X_{(K)}(X_c)\| > \varepsilon] \tag{12}$$
$$\leq \mathbb{P}[X_c \in B(x, \alpha - \varepsilon), \|X_c - X_{(K)}(X_c)\| \leq \varepsilon] \tag{13}$$
$$\leq \mathbb{P}[X_c \in B(x, \alpha - \varepsilon), \|X_c - Z\| \leq \varepsilon] \tag{14}$$
$$\leq \mathbb{P}[Z \in B(x, \alpha)]. \tag{15}$$

Similarly, we have

$$\mathbb{P}[Z \in B(x, \alpha)] - \mathbb{P}[\|X_c - X_{(K)}(X_c)\| > \varepsilon] \tag{16}$$
$$\leq \mathbb{P}[Z \in B(x, \alpha), \|X_c - X_{(K)}(X_c)\| \leq \varepsilon] \tag{17}$$
$$\leq \mathbb{P}[Z \in B(x, \alpha), \|X_c - Z\| \leq \varepsilon] \tag{18}$$
$$\leq \mathbb{P}[X_c \in B(x, \alpha + \varepsilon)]. \tag{19}$$

Since $X_c$ admits a density, for all $\varepsilon > 0$ small enough

$$\mathbb{P}[X_c \in B(x, \alpha + \varepsilon)] \leq \mathbb{P}[X_c \in B(x, \alpha)] + \eta, \tag{20}$$

and

$$\mathbb{P}[X_c \in B(x, \alpha)] - \eta \leq \mathbb{P}[X_c \in B(x, \alpha - \varepsilon)]. \tag{21}$$

Let $\varepsilon$ such that equation 20 and equation 21 are verified. According to Lemma 2.3 in Biau & Devroye (2015), since $X_1, \ldots, X_n$ are i.i.d., if $K/n$ tends to zero as $n \to \infty$, we have

$$\mathbb{P}[\|X_c - X_{(K)}(X_c)\| > \varepsilon] \to 0. \tag{22}$$

Thus, for all $n$ large enough,

$$\mathbb{P}[X_c \in B(x, \alpha)] - 2\eta \leq \mathbb{P}[Z \in B(x, \alpha)] \tag{23}$$

and

$$\mathbb{P}[Z \in B(x, \alpha)] \leq 2\eta + \mathbb{P}[X_c \in B(x, \alpha)]. \tag{24}$$

Finally, for all $\eta > 0$, for all $n$ large enough, we obtain

$$\mathbb{P}[X_c \in B(x, \alpha)] - 2\eta \leq \mathbb{P}[Z \in B(x, \alpha)] \leq 2\eta + \mathbb{P}[X_c \in B(x, \alpha)], \tag{25}$$

which proves that

$$\mathbb{P}[Z \in B(x, \alpha)] \to \mathbb{P}[X_c \in B(x, \alpha)]. \tag{26}$$

Therefore, by the Monotone convergence theorem, for all Borel sets $B \subset \mathbb{R}^d$,

$$\mathbb{P}[Z \in B] \to \mathbb{P}[X_c \in B]. \tag{27}$$

$\square$

### B.3 PROOF OF LEMMA 3.3

*Proof of Lemma 3.3.* We consider a single SMOTE iteration. Recall that the central point $X_c$ (see Algorithm 1) is fixed, and thus denoted by $x_c$.

The random variables $X_{(1)}(x_c), \ldots, X_{(n-1)}(x_c)$ denote a reordering of the initial observations $X - 1, X_2, \ldots, X_n$ such that

$$||X_{(1)}(x_c) - x_c|| \le ||X_{(2)}(x_c) - x_c|| \le \ldots \le ||X_{(n-1)}(x_c) - x_c||.$$

For clarity, we remove the explicit dependence on $x_c$. Recall that SMOTE builds a linear interpolation between $x_c$ and one of its $K$ nearest neighbors chosen uniformly. Then the newly generated point $Z$ satisfies

$$Z = (1 - W)x_c + W \sum_{k=1}^{K} X_{(k)} \mathbb{1}_{\{I=k\}}, \tag{28}$$

where $W$ is a uniform random variable over $[0, 1]$, independent of $I, X_1, \ldots, X_n$, with $I$ distributed as $\mathcal{U}(\{1, \ldots, K\})$.

From now, consider that the $k$-th nearest neighbor of $x_c$, $X_{(k)}(x_c)$, has been chosen (that is $I = k$). Then $Z$ satisfies

$$Z = (1 - W)x_c + W X_{(k)} \tag{29}$$
$$= x_c - W x_c + W X_{(k)}, \tag{30}$$

which implies

$$Z - x_c = W(X_{(k)} - x_c). \tag{31}$$

Let $f_{Z-x_c}$, $f_W$ and $f_{X_{(k)}-x_c}$ be respectively the density functions of $Z - x_c$, $W$ and $X_{(k)} - x_c$. Let $z, z_1, z_2 \in \mathbb{R}^d$. Recall that $z \le z_1$ means that each component of $z$ is lower than the corresponding component of $z_1$. Since $W$ and $X_{(k)} - x_c$ are independent, we have,

$$\mathbb{P}(z_1 \le Z - x_c \le z_2) = \int_{w \in \mathbb{R}} \int_{x \in \mathbb{R}^d} f_{W, X_{(k)}-x_c}(w, x) \mathbb{1}_{\{z_1 \le wx \le z_2\}} \mathrm{d}w \mathrm{d}x \tag{32}$$

$$= \int_{w \in \mathbb{R}} \int_{x \in \mathbb{R}^d} f_W(w) f_{X_{(k)}-x_c}(x) \mathbb{1}_{\{z_1 \le wx \le z_2\}} \mathrm{d}w \mathrm{d}x \tag{33}$$

$$= \int_{w \in \mathbb{R}} f_W(w) \left( \int_{x \in \mathbb{R}^d} f_{X_{(k)}-x_c}(x) \mathbb{1}_{\{z_1 \le wx \le z_2\}} \mathrm{d}x \right) \mathrm{d}w. \tag{34}$$

Besides, let $u = wx$. Then $x = (\frac{u_1}{w}, \ldots, \frac{u_d}{w})^T$. The Jacobian of such transformation equals:

$$\begin{vmatrix} \frac{\partial x_1}{\partial u_1} & \cdots & \frac{\partial x_1}{\partial u_d} \\ \vdots & \ddots & \vdots \\ \frac{\partial x_d}{\partial u_1} & \cdots & \frac{\partial x_d}{\partial u_d} \end{vmatrix} = \begin{vmatrix} \frac{1}{w} & & 0 \\ & \ddots & \\ 0 & \cdots & \frac{1}{w} \end{vmatrix} = \frac{1}{w^d} \tag{35}$$

Therefore, we have $x = u/w$ and $\mathrm{d}x = \mathrm{d}u/w^d$, which leads to

$$\mathbb{P}(z_1 \le Z - x_c \le z_2) \tag{36}$$

$$= \int_{w \in \mathbb{R}} \frac{1}{w^d} f_W(w) \left( \int_{u \in \mathbb{R}^d} f_{X_{(k)}-x_c} \left( \frac{u}{w} \right) \mathbb{1}_{\{z_1 \le u \le z_2\}} \mathrm{d}u \right) \mathrm{d}w. \tag{37}$$

Note that a random variable $Z'$ with density function

$$f_{Z'}(z') = \int_{w \in \mathbb{R}} \frac{1}{w^d} f_W(w) f_{X_{(k)}-x_c} \left( \frac{z'}{w} \right) \mathrm{d}w \tag{38}$$

satisfies, for all $z_1, z_2 \in \mathbb{R}^d$,

$$\mathbb{P}(z_1 \le Z - x_c \le z_2) = \int_{w \in \mathbb{R}} \frac{1}{w^d} f_W(w) \left( \int_{u \in \mathbb{R}^d} f_{X_{(k)}-x_c} \left( \frac{u}{w} \right) \mathbb{1}_{\{z_1 \le u \le z_2\}} \mathrm{d}u \right) \mathrm{d}w. \tag{39}$$

Therefore, the variable $Z - x_c$ admits the following density

$$f_{Z-x_c}(z'|X_c = x_c, I = k) = \int_{w \in \mathbb{R}} \frac{1}{w^d} f_W(w) f_{X_{(k)}-x_c}\left(\frac{z'}{w}\right) \mathrm{d}w. \tag{40}$$

Since $W$ follows a uniform distribution on $[0, 1]$, we have

$$f_{Z-x_c}(z'|X_c = x_c, I = k) = \int_0^1 \frac{1}{w^d} f_{X_{(k)}-x_c}\left(\frac{z'}{w}\right) \mathrm{d}w. \tag{41}$$

The density $f_{X_{(k)}-x_c}$ of the $k$-th nearest neighbor of $x_c$ can be computed exactly (see, Lemma 6.1 in Berrett, 2017), that is

$$f_{X_{(k)}-x_c}(u) = (n-1)\binom{n-2}{k-1} f_X(x_c + u)\left[\mu_X\left(B(x_c, \|u\|)\right)\right]^{k-1}$$
$$\times \left[1 - \mu_X\left(B(x_c, \|u\|)\right)\right]^{n-k-1}, \tag{42}$$

where

$$\mu_X\left(B(x_c, \|u\|)\right) = \int_{B(x_c, \|u\|)} f_X(x)\mathrm{d}x. \tag{43}$$

We recall that $B(x_c, \|u\|)$ is the ball centered on $x_c$ and of radius $\|u\|$. Hence we have

$$f_{X_{(k)}-x_c}(u) = (n-1)\binom{n-2}{k-1} f_X(x_c + u)\mu_X\left(B(x_c, \|u\|)\right)^{k-1}\left[1 - \mu_X\left(B(x_c, \|u\|)\right)\right]^{n-k-1}. \tag{44}$$

Since $Z - x_c$ is a translation of the random variable $Z$, we have

$$f_Z(z|X_c = x_c, I = k) = f_{Z-x_c}(z - x_c|X_c = x_c, I = k). \tag{45}$$

Injecting Equation (44) in Equation (41), we obtain

$$f_Z(z|X_c = x_c, I = k) \tag{46}$$
$$= f_{Z-x_c}(z - x_c|X_c = x_c, I = k) \tag{47}$$
$$= \int_0^1 \frac{1}{w^d} f_{X_{(k)}-x_c}\left(\frac{z - x_c}{w}\right) \mathrm{d}w \tag{48}$$
$$= (n-1)\binom{n-2}{k-1} \int_0^1 \frac{1}{w^d} f_X\left(x_c + \frac{z - x_c}{w}\right) \mu_X\left(B\left(x_c, \frac{\|z - x_c\|}{w}\right)\right)^{k-1} \tag{49}$$
$$\times \left[1 - \mu_X\left(B\left(x_c, \frac{\|z - x_c\|}{w}\right)\right)\right]^{n-k-1} \mathrm{d}w \tag{50}$$

Recall that in SMOTE, $k$ is chosen at random in $\{1, \ldots, K\}$ through the uniform random variable $I$. So far, we have considered $I$ fixed. Taking the expectation with respect to $I$, we have

$$f_Z(z|X_c = x_c) \tag{51}$$

$$= \sum_{k=1}^{K} f_Z(z|X_c = x_c, I = k) \mathbb{P}[I = k] \tag{52}$$

$$= \frac{1}{K} \sum_{k=1}^{K} \int_0^1 \frac{1}{w^d} f_{X_{(k)} - x_c}\left(\frac{z - x_c}{w}\right) dw \tag{53}$$

$$= \frac{1}{K} \sum_{k=1}^{K} (n-1)\binom{n-2}{k-1} \int_0^1 \frac{1}{w^d} f_X\left(x_c + \frac{z-x_c}{w}\right) \mu_X\left(B\left(x_c, \frac{||z-x_c||}{w}\right)\right)^{k-1} \tag{54}$$

$$\times [1 - \mu_X\left(B\left(x_c, \frac{||z-x_c||}{w}\right)\right)]^{n-k-1} dw \tag{55}$$

$$= \frac{(n-1)}{K} \int_0^1 \frac{1}{w^d} f_X\left(x_c + \frac{z-x_c}{w}\right) \sum_{k=1}^{K} \binom{n-2}{k-1} \mu_X\left(B\left(x_c, \frac{||z-x_c||}{w}\right)\right)^{k-1} \tag{56}$$

$$\times [1 - \mu_X\left(B\left(x_c, \frac{||z-x_c||}{w}\right)\right)]^{n-k-1} dw \tag{57}$$

$$= \frac{(n-1)}{K} \int_0^1 \frac{1}{w^d} f_X\left(x_c + \frac{z-x_c}{w}\right) \sum_{k=0}^{K-1} \binom{n-2}{k} \mu_X\left(B\left(x_c, \frac{||z-x_c||}{w}\right)\right)^{k} \tag{58}$$

$$\times \left[1 - \mu_X\left(B\left(x_c, \frac{||z-x_c||}{w}\right)\right)\right]^{n-k-2} dw. \tag{59}$$

Note that the sum can be expressed as the cumulative distribution function of a Binomial distribution parameterized by $n-2$ and $\mu_X(B(x_c, ||z-x_c||/w))$, so that

$$\sum_{k=0}^{K-1} \binom{n-2}{k} \mu_X\left(B\left(x_c, \frac{||z-x_c||}{w}\right)\right)^{k} \left[1 - \mu_X\left(B\left(x_c, \frac{||z-x_c||}{w}\right)\right)\right]^{n-k-2} \tag{60}$$

$$= (n-K-1)\binom{n-2}{K-1} \mathcal{B}\left(n-K-1, K; 1 - \mu_X\left(B\left(x_c, \frac{||z-x_c||}{w}\right)\right)\right), \tag{61}$$

(see Technical Lemma C.1 for details). We inject Equation (61) in Equation (51)

$$f_Z(z|X_c = x_c) = (n-K-1)\binom{n-1}{K} \int_0^1 \frac{1}{w^d} f_X\left(x_c + \frac{z-x_c}{w}\right)$$

$$\times \mathcal{B}\left(n-K-1, K; 1 - \mu_X\left(B\left(x_c, \frac{||z-x_c||}{w}\right)\right)\right) dw. \tag{62}$$

We know that

$$f_Z(z) = \int_{x_c \in \mathcal{X}} f_Z(z|X_c = x_c) f_X(x_c) dx_c.$$

Combining this remark with the result of Equation (62) we get

$$f_Z(z) = (n-K-1)\binom{n-1}{K} \int_{x_c \in \mathcal{X}} \int_0^1 \frac{1}{w^d} f_X\left(x_c + \frac{z-x_c}{w}\right)$$

$$\times \mathcal{B}\left(n-K-1, K; 1 - \mu_X\left(B\left(x_c, \frac{||z-x_c||}{w}\right)\right)\right) f_X(x_c) dw dx_c. \tag{63}$$

**Link with Elreedy's formula**   According to the Elreedy formula

$$f_Z(z|X_c = x_c) = (n-K-1)\binom{n-1}{K} \int_{r=||z-x_c||}^{\infty} f_X\left(x_c + \frac{(z-x_c)r}{||z-x_c||}\right) \frac{r^{d-2}}{||z-x_c||^{d-1}}$$

$$\times \mathcal{B}\left(n-K-1, K; 1 - \mu_X\left(B\left(x_c, r\right)\right)\right) dr. \tag{64}$$

Now, let $r = \|z - x_c\|/w$ so that $\mathrm{d}r = -\|z - x_c\|\mathrm{d}w/w^2$. Thus,

$$f_Z(z|X_c = x_c)$$

$$= (n - K - 1)\binom{n-1}{K}\int_0^1 f_X\left(x_c + \frac{z - x_c}{w}\right)\frac{1}{w^{d-2}}\frac{1}{\|z - x_c\|} \tag{65}$$

$$\times \mathcal{B}\left(n - K - 1, K; 1 - \mu_X\left(B\left(x_c, \frac{z - x_c}{w}\right)\right)\right)\frac{\|z - x_c\|}{w^2}\mathrm{d}w \tag{66}$$

$$= (n - K - 1)\binom{n-1}{K}\int_0^1 \frac{1}{w^d}f_X\left(x_c + \frac{z - x_c}{w}\right)$$

$$\times \mathcal{B}\left(n - K - 1, K; 1 - \mu_X\left(B\left(x_c, \frac{z - x_c}{w}\right)\right)\right)\mathrm{d}w. \tag{67}$$

$$\square$$

### B.4 Proof of Theorem 3.5

*Proof of Theorem 3.5.* Let $x_c \in \mathcal{X}$ be a central point in a SMOTE iteration. From Lemma 3.3, we have,

$$f_Z(z|X_c = x_c)$$

$$= (n - K - 1)\binom{n-1}{K}\int_0^1 \frac{1}{w^d}f_X\left(x_c + \frac{z - x_c}{w}\right)$$

$$\times \mathcal{B}\left(n - K - 1, K; 1 - \mu_X\left(B\left(x_c, \frac{\|z - x_c\|}{w}\right)\right)\right)\mathrm{d}w \tag{68}$$

$$= (n - K - 1)\binom{n-1}{K}\int_0^1 \frac{1}{w^d}f_X\left(x_c + \frac{z - x_c}{w}\right)\mathbb{1}_{\{x_c + \frac{z - x_c}{w} \in \mathcal{X}\}}$$

$$\times \mathcal{B}\left(n - K - 1, K; 1 - \mu_X\left(B\left(x_c, \frac{\|z - x_c\|}{w}\right)\right)\right)\mathrm{d}w. \tag{69}$$

Let $R \in \mathbb{R}$ such that $\mathcal{X} \subset \mathcal{B}(0, R)$. For all $u = x_c + \frac{z - x_c}{w}$, we have

$$w = \frac{\|z - x_c\|}{\|u - x_c\|}. \tag{70}$$

If $u \in \mathcal{X}$, then $u \in \mathcal{B}(0, R)$. Besides, since $x_c \in \mathcal{X} \subset B(0, R)$, we have $\|u - x_c\| < 2R$ and

$$w > \frac{\|z - x_c\|}{2R}. \tag{71}$$

Consequently,

$$\mathbb{1}_{\{x_c + \frac{z - x_c}{w} \in \mathcal{X}\}} \leq \mathbb{1}_{\{w > \frac{\|z - x_c\|}{2R}\}}. \tag{72}$$

So finally

$$\mathbb{1}_{\{x_c + \frac{z - x_c}{w} \in \mathcal{X}\}} = \mathbb{1}_{\{x_c + \frac{z - x_c}{w} \in \mathcal{X}\}}\mathbb{1}_{\{w > \frac{\|z - x_c\|}{2R}\}}. \tag{73}$$

Hence,

$$f_Z(z|X_c = x_c) = (n - K - 1)\binom{n-1}{K}\int_0^1 \frac{1}{w^d}f_X\left(x_c + \frac{z - x_c}{w}\right)\mathbb{1}_{\{x_c + \frac{z - x_c}{w} \in \mathcal{X}\}}\mathbb{1}_{\{w > \frac{\|z - x_c\|}{2R}\}}$$

$$\times \mathcal{B}\left(n - K - 1, K; 1 - \mu_X\left(B\left(x_c, \frac{\|z - x_c\|}{w}\right)\right)\right)\mathrm{d}w \tag{74}$$

$$= (n - K - 1)\binom{n-1}{K}\int_{\frac{\|z - x_c\|}{2R}}^1 \frac{1}{w^d}f_X\left(x_c + \frac{z - x_c}{w}\right)$$

$$\times \mathcal{B}\left(n - K - 1, K; 1 - \mu_X\left(B\left(x_c, \frac{\|z - x_c\|}{w}\right)\right)\right)\mathrm{d}w. \tag{75}$$

Now, let $0 < \alpha \leq 2R$ and $z \in \mathbb{R}^d$ such that $||z - x_c|| > \alpha$. In such a case, $w > \frac{\alpha}{2R}$ and:

$$f_Z(z|X_c = x_c) \tag{76}$$

$$= (n - K - 1)\binom{n-1}{K} \int_{\frac{\alpha}{2R}}^1 \frac{1}{w^d} f_X\left(x_c + \frac{z - x_c}{w}\right)$$

$$\times \mathcal{B}\left(n - K - 1, K; 1 - \mu_X\left(B\left(x_c, \frac{||z - x_c||}{w}\right)\right)\right) \mathrm{d}w \tag{77}$$

$$\leq (n - K - 1)\binom{n-1}{K} \int_{\frac{\alpha}{2R}}^1 \frac{1}{w^d} f_X\left(x_c + \frac{z - x_c}{w}\right) \mathcal{B}\left(n - K - 1, K; 1 - \mu_X\left(B\left(x_c, \alpha\right)\right)\right) \mathrm{d}w. \tag{78}$$

Let $\mu \in [0, 1]$ and $S_n$ be a binomial random variable of parameters $(n - 1, \mu)$. For all $K$,

$$\mathbb{P}[S_n \leq K] = (n - K - 1)\binom{n-1}{K} \mathcal{B}\left(n - K - 1, K; 1 - \mu\right). \tag{79}$$

According to Hoeffding's inequality, we have, for all $K \leq (n - 1)\mu$,

$$\mathbb{P}[S_n \leq K] \leq \exp\left(-2(n - 1)\left(\mu - \frac{K}{n - 1}\right)^2\right). \tag{80}$$

Thus, for all $z \notin B(x_c, \alpha)$, for all $K \leq (n - 1)\mu_X\left(B\left(x_c, \alpha\right)\right)$,

$$f_Z(z|X_c = x_c) \tag{81}$$

$$\leq \exp\left(-2(n - 1)\left(\mu_X\left(B\left(x_c, \alpha\right)\right) - \frac{K}{n - 1}\right)^2\right) \int_{\frac{\alpha}{2R}}^1 \frac{1}{w^d} f_X\left(x_c + \frac{z - x_c}{w}\right) \mathrm{d}w \tag{82}$$

$$\leq C_2 \exp\left(-2(n - 1)\left(\mu_X\left(B\left(x_c, \alpha\right)\right) - \frac{K}{n - 1}\right)^2\right) \int_{\frac{\alpha}{2R}}^1 \frac{1}{w^d} \mathrm{d}w \tag{83}$$

$$\leq C_2 \eta(\alpha, R) \exp\left(-2(n - 1)\left(\mu_X\left(B\left(x_c, \alpha\right)\right) - \frac{K}{n - 1}\right)^2\right), \tag{84}$$

with

$$\eta(\alpha, R) = \begin{cases} \ln\left(\frac{2R}{\alpha}\right) & \text{if } d = 1 \\ \frac{1}{d-1}\left(\left(\frac{2R}{\alpha}\right)^{d-1} - 1\right) & \text{otherwise} \end{cases}.$$

Letting

$$\epsilon(n, \alpha, K, x_c) = C_2 \eta(\alpha, R) \exp\left(-2(n - 1)\left(\mu_X\left(B\left(x_c, \alpha\right)\right) - \frac{K}{n - 1}\right)^2\right), \tag{85}$$

we have, for all $\alpha \in (0, 2R)$, for all $K \leq (n - 1)\mu_X\left(B\left(x_c, \alpha\right)\right)$,

$$\mathbb{P}\left(|Z - X_c| \geq \alpha | X_c = x_c\right) = \int_{z \notin \mathcal{B}(x_c, \alpha), z \in \mathcal{X}} f_Z(z|X_c = x_c) \mathrm{d}z \tag{86}$$

$$\leq \int_{z \notin \mathcal{B}(x_c, \alpha), z \in \mathcal{X}} \varepsilon(n, \alpha, K, x_c) \mathrm{d}z \tag{87}$$

$$= \varepsilon(n, \alpha, K, x_c) \int_{z \notin \mathcal{B}(x_c, \alpha), z \in \mathcal{X}} \mathrm{d}z \tag{88}$$

$$\leq c_d R^d \varepsilon(n, \alpha, K, x_c), \tag{89}$$

as $\mathcal{X} \subset B(0, R)$. Since $x_c \in \mathcal{X}$, by definition of the support, we know that for all $\rho > 0$, $\mu_X(B(x_c, \rho)) > 0$. Thus, $\mu_X\left(B\left(x_c, \alpha\right)\right) > 0$. Consequently, $\varepsilon(n, \alpha, K, x_c)$ tends to zero, as $K/n$ tends to zero. $\qquad \square$

## B.5 PROOF OF COROLLARY 3.6

We adapt the proof of Theorem 2.1 and Theorem 2.4 in Biau & Devroye (2015) to the case where $X$ belongs to $B(0, R)$. We prove the following result.

**Lemma B.1.** *Let $X$ takes values in $B(0, R)$. For all $d \geq 2$,*

$$\mathbb{E}[\|X_{(1)}(X) - X\|_2^2] \leq 36R^2 \left( \frac{k}{n+1} \right)^{2/d}, \tag{90}$$

*where $X_{(1)}(X)$ is the nearest neighbor of $X$ among $X_1, \ldots, X_n$.*

*Proof of Lemma B.1.* Let us denote by $X_{(i,1)}$ the nearest neighbor of $X_i$ among $X_1, \ldots, X_{i-1}, X_{i+1}, \ldots, X_{n+1}$. By symmetry, we have

$$\mathbb{E}[\|X_{(1)}(X) - X\|_2^2] = \frac{1}{n+1} \sum_{i=1}^{n+1} \mathbb{E}\|X_{(i,1)} - X_i\|_2^2. \tag{91}$$

Let $R_i = \|X_{(i,1)} - X_i\|_2$ and $B_i = \{x \in \mathbb{R}^d : \|x - X_i\| < R_i/2\}$. By construction, $B_i$ are disjoint. Since $R_i \leq 2R$, we have

$$\cup_{i=1}^{n+1} B_i \subset B(0, 3R), \tag{92}$$

which implies,

$$\mu \left( \cup_{i=1}^{n+1} B_i \right) \leq (3R)^d c_d. \tag{93}$$

Thus, we have

$$\sum_{i=1}^{n+1} c_d \left( \frac{R_i}{2} \right)^d \leq (3R)^d c_d. \tag{94}$$

Besides, for all $d \geq 2$, we have

$$\left( \frac{1}{n+1} \sum_{i=1}^{n+1} R_i^2 \right)^{d/2} \leq \frac{1}{n+1} \sum_{i=1}^{n+1} R_i^d, \tag{95}$$

which leads to

$$\mathbb{E}[\|X_{(1)}(X) - X\|_2^2] = \frac{1}{n+1} \sum_{i=1}^{n+1} \mathbb{E}\|X_{(i,1)} - X_i\|_2^2 \tag{96}$$

$$= \mathbb{E} \left[ \frac{1}{n+1} \sum_{i=1}^{n+1} R_i^2 \right] \tag{97}$$

$$\leq \left( \frac{(6R)^d}{n+1} \right)^{2/d} \tag{98}$$

$$\leq 36R^2 \left( \frac{1}{n+1} \right)^{2/d}. \tag{99}$$

$\square$

**Lemma B.2.** *Let $X$ takes values in $B(0, R)$. For all $d \geq 2$,*

$$\mathbb{E}[\|X_{(k)}(X) - X\|_2^2] \leq (2^{1+2/d})36R^2 \left( \frac{k}{n} \right)^{2/d}, \tag{100}$$

*where $X_{(k)}(X)$ is the nearest neighbor of $X$ among $X_1, \ldots, X_n$.*

*Proof of Lemma B.2.* Set $d \geq 2$. Recall that $\mathbb{E}[\|X_{(k)}(X) - X\|_2^2] \leq 4R^2$. Besides, for all $k > n/2$, we have

$$(2^{1+2/d})36R^2 \left(\frac{k}{n}\right)^{2/d} > (2^{1+2/d})36R^2 \left(\frac{1}{2}\right)^{2/d} \tag{101}$$

$$> 72R^2 \tag{102}$$

$$> \mathbb{E}[\|X_{(k)}(X) - X\|_2^2]. \tag{103}$$

Thus, the result is trivial for $k > n/2$. Set $k \leq n/2$. Now, following the argument of Theorem 2.4 in Biau & Devroye (2015), let us partition the set $\{X_1, \ldots, X_n\}$ into $2k$ sets of sizes $n_1, \ldots, n_{2k}$ with

$$\sum_{j=1}^{2k} n_j = n \quad \text{and} \quad \left\lfloor \frac{n}{2k} \right\rfloor \leq n_j \leq \left\lfloor \frac{n}{2k} \right\rfloor + 1. \tag{104}$$

Let $X_{(1)}^\star(j)$ be the nearest neighbor of $X$ among all $X_i$ in the $j$th group. Note that

$$\|X_{(k)}(X) - X\|^2 \leq \frac{1}{k} \sum_{j=1}^{2k} \|X_{(1)}^\star(j) - X\|^2, \tag{105}$$

since at least $k$ of these nearest neighbors have values larger than $\|X_{(k)}(X) - X\|^2$. By Lemma B.1, we have

$$\|X_{(k)}(X) - X\|^2 \leq \frac{1}{k} \sum_{j=1}^{2k} 36R^2 \left(\frac{1}{n_j + 1}\right)^{2/d} \tag{106}$$

$$\leq \frac{1}{k} \sum_{j=1}^{2k} 36R^2 \left(\frac{2k}{n}\right)^{2/d} \tag{107}$$

$$\leq 2^{1+2/d} \times 36R^2 \left(\frac{k}{n}\right)^{2/d}. \tag{108}$$

$\square$

*Proof of Corollary 3.6.* Let $d \geq 2$. By Markov's inequality, for all $\varepsilon > 0$, we have

$$\mathbb{P}\left[\|X_{(k)}(X) - X\|_2 > \varepsilon\right] \leq \frac{\mathbb{E}[\|X_{(k)}(X) - X\|_2^2]}{\varepsilon^2}. \tag{109}$$

Let $\gamma \in (0, 1/d)$ and $\varepsilon = 12R(k/n)^\gamma$, we have

$$\mathbb{P}\left[\|X_{(k)}(X) - X\|_2 > 12R(k/n)^\gamma\right] \leq \left(\frac{k}{n}\right)^{2/d - 2\gamma}. \tag{110}$$

Noticing that, by construction of a SMOTE observation $Z_{K,n}$, we have

$$\|Z_{K,n} - X\|_2^2 \leq \|X_{(K)}(X) - X\|_2^2. \tag{111}$$

Thus,

$$\mathbb{P}\left[\|Z_{K,n} - X\|_2^2 > 12R(k/n)^\gamma\right] \leq \mathbb{P}\left[\|X_{(K)}(X) - X\|_2^2 > 12R(k/n)^{1/d}\right] \tag{112}$$

$$\leq \left(\frac{k}{n}\right)^{2/d - 2\gamma}. \tag{113}$$

$\square$

### B.6 PROOF OF THEOREM 3.7

*Proof of Theorem 3.7.* Let $\varepsilon > 0$ and $z \in B(0, R)$ such that $\|z\| \geq R - \varepsilon$. Let $A_\varepsilon = \{x \in B(0, R), \langle x - z, z \rangle \leq 0\}$. Let $0 < \alpha < 2R$ and $\tilde{A}_{\alpha,\varepsilon} = A_\varepsilon \cap \{x, \|z - x\| \geq \alpha\}$. An illustration is displayed in Figure 4.

We have

$$f_Z(z) = \int_{x_c \in \tilde{A}_{\alpha,\varepsilon}} f_Z(z|X_c = x_c) f_X(x_c) \mathrm{d}x_c + \int_{x_c \in \tilde{A}_{\alpha,\varepsilon}^c} f_Z(z|X_c = x_c) f_X(x_c) \mathrm{d}x_c \tag{114}$$

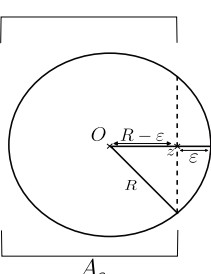

Figure 4: Illustration of Theorem 3.7.

**First term** Let $x_c \in \tilde{A}_{\alpha,\varepsilon}$. In order to have $x_c + \frac{z-x_c}{w} = z + \left(-1 + \frac{1}{w}\right)(z - x_c) \in B(0, R)$, it is necessary that

$$\left(-1 + \frac{1}{w}\right)\|z - x_c\| \leq \sqrt{2\varepsilon R} \tag{115}$$

which leads to

$$w \geq \frac{1}{1 + \frac{\sqrt{2\varepsilon R}}{\|z - x_c\|}} \tag{116}$$

Since $x_c \in \tilde{A}_{\alpha,\varepsilon}$, we have $\|x_c - z\| \geq \alpha$. Thus, according to inequality equation 116, $x_c + \frac{z-x_c}{w} \in B(0, R)$ implies

$$w \geq \frac{1}{1 + \frac{\sqrt{2\varepsilon R}}{\alpha}}. \tag{117}$$

Recall that $x_c + \frac{z-x_c}{w} \in \mathcal{X}$. Consequently, according to Lemma 3.3, for all $x_c \in \tilde{A}_{\alpha,\varepsilon}$,

$$f_Z(z|X_c = x_c) \tag{118}$$

$$= (n - K - 1)\binom{n-1}{K} \int_0^1 \frac{1}{w^d} f_X\left(x_c + \frac{z - x_c}{w}\right)$$

$$\times \mathcal{B}\left(n - K - 1, K; 1 - \mu_X\left(B\left(x_c, \frac{\|z - x_c\|}{w}\right)\right)\right) dw \tag{119}$$

$$\leq C_2(n - K - 1)\binom{n-1}{K} \int_{\frac{1}{1 + \frac{\sqrt{2\varepsilon R}}{\alpha}}}^1 \frac{1}{w^d}\mathcal{B}\left(n - K - 1, K; 1 - \mu_X\left(B\left(x_c, \frac{\|z - x_c\|}{w}\right)\right)\right) dw. \tag{120}$$

Besides,

$$(n - K - 1)\binom{n-1}{K}\mathcal{B}\left(n - K - 1, K; 1 - \mu_X\left(B\left(x_c, \frac{\|z - x_c\|}{w}\right)\right)\right) \tag{121}$$

$$= \left(\frac{n-1}{K}\right)(n - K - 1)\binom{n-2}{K-1}\mathcal{B}\left(n - K - 1, K; 1 - \mu_X\left(B\left(x_c, \frac{\|z - x_c\|}{w}\right)\right)\right) \tag{122}$$

$$\leq \frac{n-1}{K}, \tag{123}$$

according to Lemma C.1. Thus,

$$f_Z(z|X_c = x_c) \leq C_2\left(\frac{n-1}{K}\right)\int_{\frac{1}{1 + \frac{\sqrt{2\varepsilon R}}{\alpha}}}^1 \frac{1}{w^d} dw \tag{124}$$

$$\leq C_2\left(\frac{n-1}{K}\right)\eta(\alpha, R), \tag{125}$$

with

$$
\eta(\alpha, R) = \begin{cases} \ln\left(1 + \frac{\sqrt{2\varepsilon R}}{\alpha}\right) & \text{if } d = 1 \\ \frac{1}{d-1}\left(\left(1 + \frac{\sqrt{2\varepsilon R}}{\alpha}\right)^{d-1} - 1\right) & \text{otherwise} \end{cases}.
$$

**Second term**    According to Lemma 3.3, we have

$$
f_Z(z|X_c = x_c) = (n - K - 1)\binom{n-1}{K}\int_0^1 \frac{1}{w^d}f_X\left(x_c + \frac{z - x_c}{w}\right)
$$

$$
\times \mathcal{B}\left(n - K - 1, K; 1 - \mu_X\left(B\left(x_c, \frac{\|z - x_c\|}{w}\right)\right)\right)\mathrm{d}w \tag{126}
$$

$$
\leq \binom{n-1}{K}\int_0^1 \frac{1}{w^d}f_X\left(x_c + \frac{z - x_c}{w}\right)\mathrm{d}w \tag{127}
$$

Since $\mathcal{X} \subset B(0, R)$, all points $x, z \in \mathcal{X}$ satisfy $\|x - z\| \leq 2R$. Consequently, if $\|z - x_c\|/w > 2R$,

$$
x_c + \frac{\|z - x_c\|}{w} \notin \mathcal{X}. \tag{128}
$$

Hence, for all $w \leq \|z - x_c\|/2R$,

$$
f_X\left(x_c + \frac{z - x_c}{w}\right) = 0. \tag{129}
$$

Plugging this equality into equation 127, we have

$$
f_Z(z|X_c = x_c) \tag{130}
$$

$$
\leq \binom{n-1}{K}\int_{\|z-x_c\|/2R}^1 \frac{1}{w^d}f_X\left(x_c + \frac{z - x_c}{w}\right)\mathrm{d}w \tag{131}
$$

$$
\leq C_2\binom{n-1}{K}\int_{\|z-x_c\|/2R}^1 \frac{1}{w^d}\mathrm{d}w \tag{132}
$$

$$
\leq C_2\binom{n-1}{K}\left[-\frac{1}{d-1}w^{-d+1}\right]_{\|z-x_c\|/2R}^1 \tag{133}
$$

$$
\leq C_2\binom{n-1}{K}\frac{(2R)^{d-1}}{d-1}\frac{1}{\|z - x_c\|^{d-1}}. \tag{134}
$$

Besides, note that, for all $\alpha > 0$, we have

$$
\int_{B(z,\alpha)} \frac{1}{\|z - x_c\|^{d-1}}f_X(x_c)\mathrm{d}x_c \tag{135}
$$

$$
\leq C_2 \int_{B(0,\alpha)} \frac{1}{r^{d-1}}r^{d-1}\sin^{d-2}(\varphi_1)\sin^{d-3}(\varphi_2)\ldots\sin(\varphi_{d-2})\mathrm{d}r\mathrm{d}\varphi_1\ldots\mathrm{d}\varphi_{d-2}, \tag{136}
$$

where $r, \varphi_1, \ldots, \varphi_{d-2}$ are the spherical coordinates. A direct calculation leads to

$$
\int_{B(z,\alpha)} \frac{1}{\|z - x_c\|^{d-1}}f_X(x_c)\mathrm{d}x_c
$$

$$
\leq C_2 \int_0^\alpha \mathrm{d}r \int_{S(0,\alpha)} \sin^{d-2}(\varphi_1)\sin^{d-3}(\varphi_2)\ldots\sin(\varphi_{d-2})\mathrm{d}\varphi_1\ldots\mathrm{d}\varphi_{d-2} \tag{137}
$$

$$
\leq \frac{2C_2\pi^{d/2}}{\Gamma(d/2)}\alpha, \tag{138}
$$

as

$$
\int_{S(0,\alpha)} \sin^{d-2}(\varphi_1)\sin^{d-3}(\varphi_2)\ldots\sin(\varphi_{d-2})\mathrm{d}\varphi_1\ldots\mathrm{d}\varphi_{d-2} \tag{139}
$$

is the surface of the $S^{d-1}$ sphere. Finally, for all $z \in \mathcal{X}$, for all $\alpha > 0$, and for all $K, N$ such that $1 \leq K \leq N$, we have

$$
\int_{B(z,\alpha)} f_Z(z|X_c = x_c)f_X(x_c)\mathrm{d}x_c \leq \frac{2C_2^2(2R)^{d-1}\pi^{d/2}}{(d-1)\Gamma(d/2)}\binom{n-1}{K}\alpha. \tag{140}
$$

**Final result**   Using Figure 4 and Pythagore's Theorem, we have $a^2 \leq \sqrt{2\varepsilon R}$. Let $d > 1$ and $\epsilon > 0$. Then we have for all $\alpha$ such that $\alpha > a$.

$$f_Z(z) \tag{141}$$

$$= \int_{x_c \in \tilde{A}_{\alpha,\varepsilon}} f_Z(z|X_c = x_c) f_X(x_c) \mathrm{d}x_c + \int_{x_c \in \tilde{A}_{\alpha,\varepsilon}^c} f_Z(z|X_c = x_c) f_X(x_c) \mathrm{d}x_c \tag{142}$$

$$\leq \frac{C_2}{d-1} \left( \left( 1 + \frac{\sqrt{2\varepsilon R}}{\alpha} \right)^{d-1} - 1 \right) \left( \frac{n-1}{K} \right) + \frac{2C_2^2 (2R)^{d-1} \pi^{d/2}}{(d-1)\Gamma(d/2)} \left( \frac{n-1}{K} \right) \alpha \tag{143}$$

$$= \frac{C_2}{d-1} \left( \frac{n-1}{K} \right) \left[ \left( \left( 1 + \frac{\sqrt{2\varepsilon R}}{\alpha} \right)^{d-1} - 1 \right) + \frac{2C_2 (2R)^{d-1} \pi^{d/2}}{\Gamma(d/2)} \alpha \right], \tag{144}$$

But this inequality is true if $\alpha \geq a$. We know that $(1+x)^{d-1} \leq (2^{d-1} - 1)x + 1$ for $x \in [0,1]$ and $d - 1 \geq 0$. Then, for $\alpha$ such that $\frac{\sqrt{2\varepsilon R}}{\alpha} \leq 1$,

$$f_Z(z) \tag{145}$$

$$\leq \frac{C_2}{d-1} \left( \frac{n-1}{K} \right) \left[ \left( (2^{d-1} - 1) \frac{\sqrt{2\varepsilon R}}{\alpha} + 1 \right) - 1 \right) + \frac{2C_2 (2R)^{d-1} \pi^{d/2}}{\Gamma(d/2)} \alpha \right] \tag{146}$$

$$\leq \frac{C_2}{d-1} \left( \frac{n-1}{K} \right) \left[ \left( (2^{d-1} - 1) \frac{\sqrt{2\varepsilon R}}{\alpha} \right) + \frac{2C_2 (2R)^{d-1} \pi^{d/2}}{\Gamma(d/2)} \alpha \right]. \tag{147}$$

Since $\frac{\sqrt{2\varepsilon R}}{\alpha} \leq 1$, then $\alpha \geq \sqrt{2\varepsilon R} \geq a$. So our initial condition on $\alpha$ to get the upper bound of the second term is still true. Now, we choose $\alpha$ such that,

$$(2^{d-1} - 1) \frac{\sqrt{2\varepsilon R}}{\alpha} \leq \frac{2C_2 (2R)^{d-1} \pi^{d/2}}{\Gamma(d/2)} \alpha, \tag{148}$$

which leads to the following condition

$$\alpha \geq \left( \frac{\Gamma(d/2)(2^{d-1} - 1)\sqrt{2\varepsilon R}}{2C_2 (2R)^{d-1} \pi^{d/2}} \right)^{1/2}, \tag{149}$$

assuming that

$$\left( \frac{\varepsilon}{R} \right)^{1/2} \leq \frac{1}{\sqrt{2} d C_2} Vol(B_d(0,1)). \tag{150}$$

Finally, for

$$\alpha = \left( \frac{\Gamma(d/2)(2^{d-1} - 1)\sqrt{2\varepsilon R}}{2C_2 (2R)^{d-1} \pi^{d/2}} \right)^{1/2}, \tag{151}$$

we have,

$$f_Z(z) \leq \frac{C_2}{d-1} \left( \frac{n-1}{K} \right) \left[ \frac{4C_2 (2R)^{d-1} \pi^{d/2}}{\Gamma(d/2)} \alpha \right] \tag{152}$$

$$\leq \frac{C_2}{d-1} \left( \frac{n-1}{K} \right) \left[ \frac{4C_2 (2R)^{d-1} \pi^{d/2}}{\Gamma(d/2)} \left( \frac{\Gamma(d/2)(2^{d-1} - 1)\sqrt{2\varepsilon R}}{2C_2 (2R)^{d-1} \pi^{d/2}} \right)^{1/2} \right] \tag{153}$$

$$= 2^{d+2} \left( \frac{n-1}{K} \right) \left( \frac{C_2^3 Vol(B_d(0,1))}{d} \right)^{1/2} \left( \frac{\varepsilon}{R} \right)^{1/4}. \tag{154}$$

$\square$

## C  Technical lemmas

### C.0.1  Cumulative distribution function of a binomial law

**Lemma C.1** (Cumulative distribution function of a binomial distribution). *Let $X$ be a random variable following a binomial law of parameter $n \in \mathbf{N}$ and $p \in [0,1]$. The cumulative distribution function $F$ of $X$ can be expressed as Wadsworth et al. (1961):*

(i)

$$F(k; n, p) = \mathbb{P}(X \leq k) = \sum_{i=0}^{\lfloor k \rfloor} \binom{n}{i} p^i (1-p)^{n-i},$$

(ii)

$$F(k; n, p) = (n-k)\binom{n}{k} \int_0^{1-p} t^{n-k-1}(1-t)^k \mathrm{d}t$$

$$= (n-k)\binom{n}{k} \mathcal{B}(n-k, k+1; 1-p),$$

*with $\mathcal{B}(a, b; x) = \int_{t=0}^{x} t^{a-1}(1-t)^{b-1}\mathrm{d}t$, the incomplete beta function.*

*Proof.* see Wadsworth et al. (1961). □

### C.0.2  Upper bounds for the incomplete beta function

**Lemma C.2.** *Let $B(a, b; x) = \int_{t=0}^{x} t^{a-1}(1-t)^{b-1}\mathrm{d}t$, be the incomplete beta function. Then we have*

$$\frac{x^a}{a} \leq B(a, b; x) \leq x^{a-1}\left(\frac{1-(1-x)^b}{b}\right),$$

*for $a > 0$.*

*Proof.* We have

$$
\begin{aligned}
B(a, b; x) &= \int_{t=0}^{x} t^{a-1}(1-t)^{b-1}\mathrm{d}t \\
&\leq \int_{t=0}^{x} x^{a-1}(1-t)^{b-1}\mathrm{d}t \\
&= x^{a-1} \int_{t=0}^{x} (1-t)^{b-1}\mathrm{d}t \\
&= x^{a-1}\left[(-1)\frac{(1-t)^b}{b}\right]_0^x \\
&= x^{a-1}\left[-\frac{(1-x)^b}{b} + \frac{1}{b}\right] \\
&= x^{a-1}\frac{1-(1-x)^b}{b}.
\end{aligned}
$$

On the other hand,

$$
\begin{aligned}
B(a, b; x) &= \int_{t=0}^{x} t^{a-1}(1-t)^{b-1}\mathrm{d}t \\
&\geq \int_{t=0}^{t} x^{a-1}\mathrm{d}t \\
&= \left[\frac{t^a}{a}\right]_{0}^{x} \\
&= \frac{x^a}{a} - \frac{0^a}{a} \\
&= \frac{x^a}{a}.
\end{aligned}
$$

$\square$

### C.0.3 UPPER BOUNDS FOR BINOMIAL COEFFICIENT

**Lemma C.3.** *For $k, n \in \mathbb{N}$ such that $k < n$, we have*

$$
\binom{n}{k} \leq \left(\frac{en}{k}\right)^{k}. \tag{155}
$$

*Proof.* We have,

$$
\binom{n}{k} = \frac{n(n-1)\ldots(n-k+1)}{k!} \leq \frac{n^k}{k!}. \tag{156}
$$

Besides,

$$
e^k = \sum_{i=0}^{+\infty} \frac{k^i}{i!} \implies e^k \geq \frac{k^k}{k!} \implies \frac{e^k}{k^k} \geq \frac{1}{k!}. \tag{157}
$$

Hence,

$$
\binom{n}{k} = \frac{n(n-1)\ldots(n-k+1)}{k!} \leq \frac{n^k}{k!} \leq \left(\frac{en}{k}\right)^{k}. \tag{158}
$$

$\square$

### C.0.4 INEQUALITY $x \ln\left(\frac{1}{x}\right) \leq \sqrt{x}$

**Lemma C.4.** *For $x \in ]0, +\infty[$,*

$$
x \ln\left(\frac{1}{x}\right) \leq \sqrt{x}. \tag{159}
$$

*Proof.* Let,

$$
f(x) = \sqrt{x} - x \ln\left(\frac{1}{x}\right) \tag{160}
$$

$$
= \sqrt{x} + x \ln(x). \tag{161}
$$

Then,

$$f'(x) = \frac{1}{2\sqrt{x}} + \ln x + 1. \tag{162}$$

And,

$$f''(x) = \frac{1}{x} - \frac{1}{4x^{3/2}}. \tag{163}$$

We have,

$$f''(x) \geq 0 \implies \frac{1}{x} - \frac{1}{4x^{3/2}} \geq 0$$

$$\implies \frac{1}{x} \geq \frac{1}{4x^{3/2}} \tag{164}$$

Since $x \in ]0, +\infty[$,

$$Equation\ (164) \implies \frac{x^{3/2}}{x} \geq \frac{1}{4} \tag{165}$$

$$\implies \sqrt{x} \geq \frac{1}{4} \tag{166}$$

$$\implies x \geq \frac{1}{16}. \tag{167}$$

This result leads to,

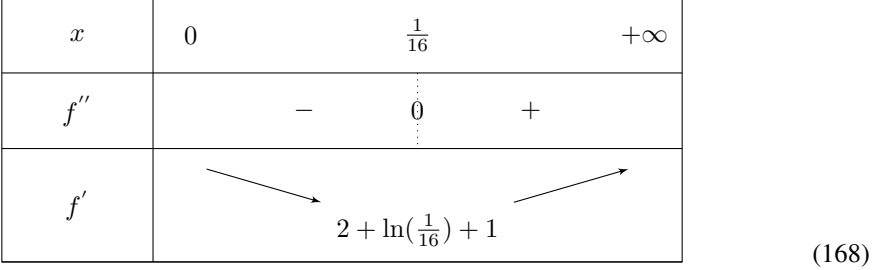

$$\tag{168}$$

We have $2 + \ln(\frac{1}{16}) + 1 > 0$. So $f'(x) > 0$ for all $x \in ]0, \infty[$. Furthermore $\lim_{x \to 0^+} f(x) = 0$, hence $f(x) > 0$ for all $x \in ]0, \infty[$, therefore $\sqrt{x} > x \ln\left(\frac{1}{x}\right)$ for all $x \in ]0, \infty[$.

$\square$

