# OpenReview forum: "Do we need rebalancing strategies? A theoretical and empirical study around SMOTE and its variants"
_ICLR.cc/2025/Conference — Submitted to ICLR 2025_

### Official Review · Reviewer_7ocJ · 2024-10-29

**Soundness:** 3
**Presentation:** 4
**Contribution:** 3
**Rating:** 8
**Confidence:** 4

**Summary:**

The paper investigates the effectiveness of rebalancing strategies for handling imbalanced datasets in binary classification, specifically focusing on the Synthetic Minority Oversampling Technique (SMOTE) and its variants. The authors provide a theoretical analysis demonstrating that default SMOTE tends to asymptotically copy original minority samples and exhibits boundary artifacts. They introduce two new strategies, CV-SMOTE and Multivariate Gaussian SMOTE (MGS), and compare their performance against traditional methods like Random Under Sampling (RUS), Random Over Sampling (ROS), and Class Weighting (CW). The empirical results indicate that for many datasets, applying no rebalancing strategy is competitive, while the proposed methods show promise for highly imbalanced datasets.

**Strengths:**

1. The paper is well-organized, with clear sections that guide the reader through the theoretical analysis, empirical evaluation, and results.
2. The paper provides a solid theoretical foundation for understanding the behavior of SMOTE, including its limitations and boundary artifacts.
3. The authors conduct a thorough empirical evaluation across multiple datasets.

**Weaknesses:**

1. The diversity in terms of data characteristics (e.g., feature types, distributions) could be expanded.
2. More recent references and baselines should be included.

**Questions:**

What criteria should practitioners use to select the hyperparameter grid for CV-SMOTE? Are there specific guidelines or best practices that can be derived from the experiments?

---

> ### Author Response · Authors · 2024-11-15
> **Response to reviewer 7ocJ**
>
> We thank you for your positive feedback, both regarding the theoretical analysis and the experimental section.
>
> You suggest applying our analysis to further data sets with different characteristics. Since we consider only tabular data sets in our study, it is difficult to consider input variable of different types (such as pixels or word embedding). We only consider continuous features as SMOTE was originally designed to handle such features. Extending our proposed method to categorical features is not straightforward, as the dependence between continuous and categorical features must be taken into account in the generative process. Thus, we leave this avenue for future works.
> Besides, all the data sets already included in the paper treats different machine learning problem. Most of them are used by several seminal works on imbalanced data (see [1,2 and 3]) or included in the open source package imbalanced learn (see [4]). Furthermore, in our study, we already analyze 13 data sets and undersample them some of them times in order to lower the imbalance ratio. If you have any suggestions on how to improve our empirical evaluation, we would be happy to incorporate it in our paper.
>
> You suggest that we should include more recent references and baselines in our work. We have implemented and compared the most common rebalancing strategies, contained in the open-source package imbalanced learn [2], that is class-weight, RUS, ROS, NearMiss1, Borderline SMOTE 1, Borderline SMOTE 2, SMOTE. Note that we implemented the Adasyn strategy but did not display the results as they were very similar to that of SMOTE. We have also used logistic regression, random forest and tree boosting as classifiers.
> Even if it does not fall into the scope of our study, we also implemented long-tail learning strategies (usually used for images and deep learning algorithms, see Table 17). We are open to suggestions if you have some other specific algorithms in mind.
>
> Finally, you ask if we can provide guidelines to choose the grid of CV-SMOTE. We originally choose the grid to test theoretical values, that is fraction of $n$ and $\sqrt{n}$. We also added some very small values for $K$, as surprisingly, those values can perform well in some data sets. However, we did not obtain a clear picture on what should be the preferred values for $K$. As this question is interesting, we will compute the best values for $K$ (the one output by the cross-validation procedure) and add them to the paper. Thank you for this suggestion!
>
> [1] Chawla, N.V., Bowyer, K.W., Hall, L.O. and Kegelmeyer, W.P., 2002. SMOTE: synthetic minority over-sampling technique. Journal of artificial intelligence research, 16, pp.321-357.
>
> [2] Han, H., Wang, W.Y. and Mao, B.H., 2005, August. Borderline-SMOTE: a new over-sampling method in imbalanced data sets learning. In International conference on intelligent computing (pp. 878-887). Berlin, Heidelberg: Springer Berlin Heidelberg.
>
> [3]He, H., Bai, Y., Garcia, E.A. and Li, S., 2008, June. ADASYN: Adaptive synthetic sampling approach for imbalanced learning. In 2008 IEEE international joint conference on neural networks (IEEE world congress on computational intelligence) (pp. 1322-1328). Ieee.
>
> [4]Lemaitre, G., Nogueira, F. and Aridas, C.K., 2017. Imbalanced-learn: A python toolbox to tackle the curse of imbalanced datasets in machine learning. Journal of machine learning research, 18(17), pp.1-5.

---

### Official Review · Reviewer_XTgD · 2024-11-02

**Soundness:** 2
**Presentation:** 2
**Contribution:** 2
**Rating:** 5
**Confidence:** 4

**Summary:**

This paper analyzes SMOTE theoretically. Its contributions include:

(1) It proves that, without tuning the hyperparameter K (usually set to 5), SMOTE asymptotically copies the original minority samples, therefore lacking the intrinsic variability required in any synthetic generative procedure. It provides numerical illustrations of this limitation.

(2) It proves that SMOTE density vanishes near the boundary of the support of the minority distribution, therefore justifying the introduction of SMOTE variants such as BorderLine SMOTE.

(3) It introduces two SMOTE alternatives, CVSMOTE and Multivariate Gaussian SMOTE (MGS). It evaluates these two new strategies and state-of-the-art rebalancing strategies on several real-world data sets using random forests/logistic regression/LightGBM. These experiments show that applying no strategy is competitive for most data sets. For the remaining data sets, the proposed strategies are among the best strategies in terms of predictive performance. The analysis of experiment results also provides some explanations about the good behavior of RUS, due to an implicit regularization in presence of random forests classifiers.

**Strengths:**

1. This paper theoretically analyzes Synthetic Minority Oversampling Technique (SMOTE), which is a widely adopted rebalancing strategy for handling imbalanced tabular data sets and has numerous variants. The theoretical analysis is comprehensive.
2. The paper proposes two SMOTE alternatives: CVSMOTE and Multivariate Gaussian SMOTE (MGS), thus making minor technical innovations.
3. Through extensive experiments on various datasets, the paper demonstrates that applying no rebalancing strategy can be competitive for some data sets. This finding is insightful.

**Weaknesses:**

Please refer to questions.

**Questions:**

1. It is recommended to clearly explain the implications of the metric \(\bar{C}(Z, X)/\bar{C}(\tilde{X}, X)\), such as what it signifies when the value approaches 1, 0, or exceeds 1, as well as the impact of the corresponding synthetic samples on the classification model performance, which currently lacks discussion.

2. Regarding Theorem 3.2, the author's claim that "This highlights a good behavior of the default setting of SMOTE (K = 5), as it can create more data points, different from the original sample, and distributed as the original sample" needs further justification. The reasoning for "create more data points" is unclear, and with finite \(n\) in practice, the conclusion "distributed as the original sample" may require additional support.

3. In the statement, “Choosing \(K\) that increases with \(n\) leads to larger characteristic distances: SMOTE observations are more distant from their central points. Corollary 3.6 leads us to choose \(K\) such that \(K/n\) does not tend too fast to zero, so that SMOTE observations are not too close to the original minority samples,” it's hard to understand “Choosing \(K\) that increases with \(n\)” and “tend too fast to zero.” These informal expressions can be confusing. It is suggested to explain the practical meaning of larger characteristic distances, such as diversity, to aid understanding.

4. In Section 3.2, THEORETICAL RESULTS ON SMOTE, could explicit ranges for \(K\) values be provided? Is there a trade-off between "regenerating the distribution of the minority class," avoiding "boundary bias," and achieving "more diversity"?

5. The term “boundary artifacts” appears only in the abstract and conclusion but not elsewhere in the paper. Furthermore, no reference is found in the original BorderLine SMOTE paper. Additionally, the author claims to “prove that SMOTE exhibits boundary artifacts, therefore justifying the introduction of SMOTE variants such as BorderLine SMOTE (Section 3),” so it is recommended to provide relevant citations and explain how BorderLine SMOTE addresses this issue.

6. Concerns regarding Section 3.3, NUMERICAL ILLUSTRATIONS:
   - (1) In the first result analysis, the author states, “Note that, for the other asymptotics in \(K\), the diversity of SMOTE observations increases with \(n\), meaning \(\bar{C}(Z, X)\) gets closer to \(\bar{C}(\tilde{X}, X)\). This behavior in terms of average distance is ideal since \(\tilde{X}\) is drawn from the same theoretical distribution as \(X\). On the contrary, \(K = 5\) maintains a lower average distance, indicating a lack of diversity in generated points.” Based on previous descriptions, higher diversity is considered better, so \(K = 5\) with low diversity should be undesirable, which conflicts with the statement “This highlights a good behavior of the default setting of SMOTE (K = 5).”
   - (2) How should “this diversity is asymptotically more important for xxx” be understood?
   - (3) How should “By construction, SMOTE data points are close to central points, which may explain why the quantity of interest in Figure 1 is smaller than 1” be interpreted? Furthermore, in the second result analysis, some values of the metric \(\bar{C}(Z, X)/\bar{C}(\tilde{X}, X)\) exceed 1; what implications would this have?

7. The conclusion that “applying no strategy is competitive for most data sets” based solely on “the imbalance ratio is not high enough or the learning task not difficult enough” is not rigorous enough. A more detailed analysis of dataset characteristics would help identify when it’s appropriate to apply no strategy versus when to use rebalancing strategies.

8. It is recommended to compare some newer SMOTE variants, as some of the rebalancing strategies used in the experiments are somewhat outdated.

9. More precise and standardized descriptions would improve readability. For example, phrases like “guarantees asymptotically” and “smoothed out” require clarification. Consistent terminology should be used instead of mixing “SMOTE data points,” “SMOTE samples,” and “SMOTE observations.” Additionally, the title of Table 2 could specify the experimental module for clarity.

---

> ### Author Response · Authors · 2024-11-15
> **Response to reviewer XTgD**
>
> We thank you very much for your careful reading and your numerous comments. We acknowledge that our phrasing is somehow imprecise when discussing the impact of our theoretical or empirical results. This was meant to ease the interpretation but, based on your feedback, we note that this can be misleading. We will correct the paper accordingly to avoid confusions.
>
> - Weakness 1 and 6 : The quantity $\bar{C}(Z, X)$ measures how far SMOTE observations $Z_i$ are from the original observations $X_i$. The quantity $\bar{C}(\tilde{X}, X)$ measures how far data $\tilde{X}_i$, distributed as the original observations $X_i$, are from these observations ($X_i$). The metric $\bar{C}(Z, X)/\bar{C}(\tilde{X}, X)$ quantify the ability of the synthetic samples to fill the space in the same way as samples generated with the true distribution.
>     - When $\bar{C}(Z, X)/\bar{C}(\tilde{X}, X)$ is close to one, SMOTE observations are comparable to observations $\tilde{X}$, in terms of the metric $\bar{C}$. In this case, data generated via SMOTE behave in the same way as the original data, with respect to the quantity $\bar{C}$.
>     - Similarly, when the quantity $\bar{C}(Z, X)/\bar{C}(\tilde{X}, X)$ is close to zero, observations generated via SMOTE are much closer to their central point than observations generated with the true distribution. In this case, this highlights that SMOTE has a tendency to generate data close to the original observations.
>     - Conversely, when  $\bar{C}(Z, X)/\bar{C}(\tilde{X}, X)$ is larger than one, data generated via SMOTE are further away from their central point than observations generated with the true distribution. In this case, this highlight that diversity of newly created samples is too high.
>
> The statement $(i)$ “This highlights a good behavior of the default setting of SMOTE (K = 5).” (line 180/181) refers to the Theorem 3.2.
>     Theorem 3.2 establishes that samples generated by SMOTE converge in distribution to the original random variable $X$, which is the asymptotic behavior that we expect from a synthetic data-level strategy. However, Theorem 3.5 and Corollary 3.6 show that SMOTE samples are close to the original data, which leads us to conclude to a lack of diversity or to the fact that SMOTE has a tendency to asymptotically copy the original samples.
>     Figure 1 illustrates this tendency for SMOTE with default value $K=5$. Our statement $(ii)$ from lines 289 to 296 (the one you quote) is a comment of Figure 1. Thus, there is no contradiction between statements $(i)$ and $(ii)$. We will clarify this ambiguity in the new version.
>
> - Weakness 2 : Any data point generated via SMOTE $Z_{K,n}$ is shown to converge in distribution to any original data point $X$, when $K/n$ tends to zero as $n$ tends to infinity. Thus, SMOTE allows us to generate new data points, which are asymptotically distributed as the original samples. As many statistical analyses, we assume that the size of the minority class tends to infinity in order to study the behavior of SMOTE. Though unrealistic, this setting tends to mimic a data set composed of a high number of observations. Note that this does not contradict the imbalanced classification setting, as we can consider an overall data set whose size tend to infinity, with a fixed proportion of minority samples, therefore allowing the size of the minority class to tend to infinity.  Practically, there exist highly imbalanced data sets with a large value of $n$. For example, the CreditCard data set, included in our study, has an imbalance ratio of 0.2\% and $n=492$. Indeed, in the banking sector, it is common to have several millions of clients and among them only a few thousands make fraudulent transactions, leading to a high value of $n$ and highly imbalanced data.
>
> - Weakness 3 and Weakness 4: It is difficult to quantify what is the right amount of diversity (for SMOTE samples) that help to improve the classifier performances. In fact, it strongly depends on the distribution of the minority class and of the chosen classifier (logistic regression, random forests...). For example, it is not likely that using SMOTE on linearly separable data improves the performance of a logistic regression, as the original logistic regression on the original data separates properly the original samples. If an oracle tells us that generated data should be distant from less than $\alpha$ from original points with probability $0.95$, our result (equation 109 in the proof) would lead us to choose
>     \begin{align}
>         K = \left( \frac{0.95 \alpha^2 }{36 R^2 2^{1 + 2/d}} \right)^{d/2} n.
>     \end{align}
>     Studying theoretically the impact of different level of diversity on different classifier is definitely a very interesting research topic.

---

> > ### Author Response · Authors · 2024-11-15
> > **Response to reviewer XTgD (part 2)**
> >
> > - Weakness 5 : It has been noted in previous work that generating new observations near the border of the minority distribution could lead to improve predictive performances. For example, Borderline-SMOTE strategies [1] generate new synthetic samples only from minority samples on the border (see page 4 of [1]). Another example is the Adasyn procedure [2] that mainly generates new samples from original minority samples mainly surrounded by majority class samples.
> >     Our theoretical analysis justifies such methods: we prove in Theorem 3.7 that the density of SMOTE observations is lower than expected near the boundary of the support. Thus, the distribution of SMOTE observations may be far from the true distribution near the boundary, for finite $K$ and $n$. This phenomenon only appears near the boundary of the support, which is what we call boundary artifacts (we will clarify the terminology). A manner to circumvent this issue is to over-sample observations near the boundary of the support, a solution proposed in [1,2] for example.
> >
> > - Weakness 7 : Identifying precisely data sets characteristics for which None performs better would definitely strengthen our paper. Unfortunately, it is often a very difficult problem to give such a characterization as there is a lot of variability in data set characteristics (number of input features, distributions, distribution of minority/majority class, number of samples...). We could try to design experiments on simulated data sets to analyze the impact of each single characteristic. However, our analysis would strongly depend on arbitrary choices (ie on the simulation design). We are open to discussion on this topic.
> >
> > - Weakness 8 : You suggest that we should include more recent references and baselines in our work. We have implemented and compared the most common rebalancing strategies, contained in the open-source package imbalanced learn [2], that is class-weight, RUS, ROS, NearMiss1, Borderline SMOTE 1, Borderline SMOTE 2, SMOTE. Note that we implemented the Adasyn strategy but did not display the results as they were very similar to that of SMOTE. We have also used logistic regression, random forest and tree boosting as classifiers.
> >     Even if it does not fall into the scope of our study, we also implemented long-tail learning strategies (usually used for images and deep learning algorithms, see Table 17). We are open to suggestions if you have some other specific algorithms in mind.
> >
> > [1] Han, H., Wang, W.Y. and Mao, B.H., 2005, August. Borderline-SMOTE: a new over-sampling method in imbalanced data sets learning. In International conference on intelligent computing (pp. 878-887). Berlin, Heidelberg: Springer Berlin Heidelberg.
> >
> > [2] He, H., Bai, Y., Garcia, E.A. and Li, S., 2008, June. ADASYN: Adaptive synthetic sampling approach for imbalanced learning. In 2008 IEEE international joint conference on neural networks (IEEE world congress on computational intelligence) (pp. 1322-1328). Ieee.

---

> > > ### Comment · Reviewer_XTgD · 2024-11-23
> > >
> > > The authors have addressed most of my concerns, but there are still two unresolved questions:
> > > 1. The statement that "default setting of SMOTE (K=5) can create more data points"—is this compared to other settings of SMOTE, such as K=6? This remains an unclear description that requires further clarification.
> > > 2. In Section 3.3, from lines 296 to 300: What does "this diversity is asymptotically more important for xxx" mean? Additionally, what does "By construction, SMOTE data points are close to central points, which may explain why the quantity of interest in Figure 1 is smaller than 1" mean? Could the authors consider adopting more formal phrasing instead of descriptions like “important” and “interest”?
> > >
> > > While it has been theoretically and experimentally demonstrated that using SMOTE requires tuning the hyperparameter K, and the experiments have validated that applying no strategy is competitive for certain datasets, this study does not provide actionable recommendations regarding parameter ranges or practical usage scenarios. Instead, everything ultimately depends on the dataset and chosen classifier. As a result, the contribution appears relatively weak because one may select a rebalancing strategy based on the dataset and classifier at hand, even without being aware of this paper.
> > >
> > > Therefore, I will keep my score unchanged.

---

> > > > ### Author Response · Authors · 2024-11-25
> > > >
> > > > Thank you for reading our response. Here is our answer to the last two points you raised:
> > > > - you are right, we were not precise enough in the following sentence '' This highlights a good behavior of the default setting of SMOTE ($K=5$), as it can create \textit{more} data points, different from the original sample, and distributed as the original sample.''. \textit{more} should be replaced by \textit{new}: we simply want to emphasize that SMOTE is able to generate new data points, that are different from the original samples, thus resulting in a data set containing more data points compared to the original data set.
> > > > - After reading again the entire paragraph about Figure 1, we acknowledge that this part "this diversity is asymptotically more important for $K=0.1n$ and $K=0.01n$" was unclear. We apologize for that. What we meant is that the metric $\bar{C}(Z,X)/\bar{C}(\tilde{X},X)$ converges to higher values (when $n$ tends to infinity) when we consider $K=\alpha n$, compared to $K=5$ and $K=\sqrt{n}$ (see Figure 1). As the metric $\bar{C}(Z,X)/\bar{C}(\tilde{X},X)$ measures how far generated data points are from the original data set, we can conclude that a choice of $K=\alpha n$ leads to higher diversity for large $n$ compared to $K=5$ and $K=\sqrt{n}$. To be exact and comprehensible, we correct our sentence as "this diversity is asymptotically more important for $K=0.01n, 0.1, 0.3, 0.8$ than for $K=5$ and $K=\sqrt{n}$".
> > > >
> > > > Following your last comment on our work, we would like to highlight,a s you mention, that two major benefits of our work are to (i) theoretically and empirically show that $K$ in SMOTE needs to be tuned and (ii) highlight that 'None strategy' performs well in most data sets (with random forests, boosting or logistic regression). Therefore, assuming that the computational budget is limited, we recommend using this method regardless of the classifier used, rather than testing more complex rebalancing strategies. We seem to disagree about the importance and practical implications of these points.
> > > > All in all, we respect your point of view and your criticism and hope that we have clarified our paper. We thank you for the time you took at reading our paper and our comments.

---

### Official Review · Reviewer_rG2v · 2024-11-02

**Soundness:** 3
**Presentation:** 3
**Contribution:** 2
**Rating:** 5
**Confidence:** 3

**Summary:**

This paper presents a comprehensive study of synthetic rebalancing strategies from theoretical and computational perspectives, further proposing two promising approaches specifically designed for highly imbalanced datasets. The findings reveal that for most mildly imbalanced datasets, applying no rebalancing strategy can be competitive, while rebalancing methods show clear benefits in scenarios with high imbalance.

**Strengths:**

1. The paper provides an in-depth theoretical analysis of SMOTE, offering valuable insights into its behavior and mechanics, which enhances the understanding of its effectiveness in handling imbalanced datasets.
2. The paper is well-written, with a clear articulation of its objectives and contributions. In particular, the Introduction and Related Works sections effectively outline the development of solutions for imbalanced datasets, providing a strong foundation for the study's motivation and purpose.

**Weaknesses:**

1. It would be incorrect and confusing about the definition of imbalanced ratio. For example, in line 147, the authors define the imbalance ratio as the proportion of minority samples to the total samples (minority samples/total samples). However, in line 400, the authors mention that "applying no strategy is the best, probably highlighting that the imbalance ratio is not high enough". This wording suggest low-imbalanced datasets while not high imbalance ratio in your definition indicates highly imbalanced datasets.
2. The novelty of the paper appears limited in several aspects. a. While the paper offers a thorough theoretical examination of SMOTE, the insights may not extend significantly beyond those of earlier works. I may not have the complete picture, however, and remain open to further clarification from the authors on their theoretical contributions. b. CV-SMOTE addresses hyperparameter selection via cross validation, while MGS introduces Gaussian sampling. Could the authors provide any unique aspects of their implementation or theoretical justifications that set the approaches apart from existing methods? c. The observation that applying no rebalancing strategy performs competitively on mildly imbalanced datasets is consistent with existing knowledge. Could the authors discuss how their empirical results add to or refine existing knowledge?
3. The study focuses solely on binary classification and tabular data. It would be beneficial to discuss how the findings might extend to multiclass problems to improve the paper's generalizability.

**Questions:**

My opinions and questions are outlined in the Weaknesses section.

---

> ### Author Response · Authors · 2024-11-15
> **Response to reviewer rG2v**
>
> Thank you for your detailed review.
>
> You noticed that our statement on line 400 is in contradiction with our definition of imbalance ratio. We thank you for your remark. **We propose to replace the statement on line 400 by "probably highlighting that the imbalance ratio is not low enough". We have also checked the whole paper and change the phrasing whenever it was necessary.**
>
> You state that the novelty of our theoretical results compared to prior existing works is not clear enough.
> To the best of our knowledge, there is only Elreedy and al. [1] and [2] that study theoretically SMOTE. In [2], the authors derive the expectation and the covariance of the samples generated by SMOTE. Elreedy and al. [1] derive the density of samples generated by SMOTE, and confirm this density with numerical experiments on simulated data. Thus, these two papers derive statistical quantities related to SMOTE algorithms, without drawing any conclusions regarding the performance/drawback of the actual SMOTE algorithm.
> In our paper, we use their derivations (see Lemma 3.3) to improve our knowledge on SMOTE algorithm. We first prove that SMOTE converges in probability to the minority class distribution as $K/n \to 0$ (Theorem 3.2), based on the concentration of the nearest-neighbors. Then, we proved that conditionally to the central point, an observation generated via SMOTE tends to the central point in probability (Theorem 3.5). Then, Corollary 3.6 gives a characteristic distance to this agglutination phenomenon using only inequality on the density of the nearest-neighbors. In particular, this result is valid in finite-sample settings, where $n$ is not assumed to tend to infinity.  In a similar finite-sample setting, we establish in Theorem 3.7 an upper bound of SMOTE density near the boundary of the support of the minority class. This result shows that SMOTE density vanishes near the boundary of the support, thus compromising the regeneration of the minority samples near the border. From those theoretical results, we deduce all our intuitions about SMOTE behavior. **Except for Lemma 3.3, all our results are new and thus absent from [1,2]. Besides, Elreedy et al. [1,2] do not use their derivations of SMOTE density in order to study SMOTE behavior.**
>
> You share some concerns on the novelty for CV-SMOTE and MGS compared to existing methods. Through our theoretical study, we proved that asymptotically, the default value $K=5$ of SMOTE leads to a lack of diversity in the samples generated by SMOTE.
> Then we show that for some values of $K$ (such that $K=0.1n$), SMOTE observations are more diverse.
> We tried different asymptotics for $K$ (like $K = \sqrt{n}$, $K = \alpha n$ for different values of $\alpha$) but we could not find a universal best choice. Thus, we introduce a grid optimization procedure to automatically choose the best value for $K$. We would like to remark that SMOTE is a widely used algorithm with its default hyperparameter value $K=5$ and to the best of our knowledge there is no paper recommending tuning $K$.  MGS is introduced in order to have a process that do not generate new samples only in a vector line. This would lead to more diverse synthetic samples again and a better reconstruction of the minority class distribution.
> It is surprising that such small modifications of existing algorithms lead to two new procedures with improved performances.
> Our goal is by no mean to say that these two methods improve drastically over existing ones. Indeed, for most data sets, None method is the best one. Thus, one of our major contribution is to establish that None is among the best strategy and that, in the remaining cases, our two proposals are competitive.
>
> You also explain that it is known that applying no rebalancing strategy performs competitively for mildly imbalanced data sets. We believe that this fact is not well-known in the ML community: many research papers as [3,4] aim at designing new rebalancing strategies to improve upon None. Many papers conclude that their strategy is better than None. For example, SMOTE is shown to outperform None in [3]  (in terms of F1-score and True Positive rate). Similarly, borderline-SMOTE strategies outperform None (see page 8 of [3]). Besides, in the Adasyn introduction article [4], it is shown that SMOTE and Adasyn offer significant improvement of predictive performances (in terms of F-measure and G-mean) over None strategy (see page 5 of [4]). Therefore, to the best of our knowledge, no article introducing new oversampling strategies explicitly conclude  that None is competitive.

---

> ### Author Response · Authors · 2024-11-15
> **Response to reviewer rG2v (part 2)**
>
> Furthermore, through our numerical experiments, we show that applying no rebalancing strategy is competitive, even for highly imbalanced data set such as Yeast which have an imbalance ratio of $11\%$ (see Table 5). Even when we lower the imbalance ratio, the None data set remains competitive for the Breast Cancer, Vehicle, Ionosphere and House\_16H data set for example. We recall that our data sets are widely used in the tabular imbalanced data community. For example, Vehicle and Ionosphere data sets are used in [3,4]. Thus, we believe that the empirical part of our study highlights an important practical fact: None is competitive, even for several low-imbalanced data sets.
>
> Finally, to answer your question, it is possible to extend our study to the multiclass framework. We provide here simulations on three data sets with multiclass target (some were already included in our study but binarized). We are willing to add more data sets to the final paper if needed. Furthermore, we look at the ROC AUCs of one class versus the rest and finally average them. We also look at the PR AUC averaged over all combinations. We remark that our introduced strategies remain competitive in multiclass setting.
>
>
> LightGBM ROC AUC computed on three data sets with multiple classes (average of one-vs-rest ROC AUC).
> | Strategy     | None           | CW    | RUS   | ROS   | NearMiss1  | BS1   | BS2   | SMOTE | CV-SMOTE       | MGS (d+1)                 |
> |--------------|----------------|-------|-------|-------|-------|-------|-------|--------------|--------------|-------------------|
> | Wine  | **0.831** | 0.825 | 0.664 | 0.820 | 0.623 | 0.816 | 0.824 | 0.804        | 0.798        | 0.830             |
> | Yeast | 0.884          | 0.884 | 0.857 | 0.876 | 0.812 | 0.881 | 0.882 | 0.882        | 0.882        | **0.887**    |
> | Ecoli | 0.965          | 0.965 | 0.952 | 0.961 | 0.941 | 0.956 | 0.959 | 0.960        | 0.960        | **0.966**    |
>
> LightGBM PR AUC computed on three data sets with multiple classes (average of one-vs-rest PR AUC).
> | Strategy     | None           | CW    | RUS   | ROS   | NearMiss1  | BS1   | BS2   | SMOTE | CV-SMOTE           | MGS (d+1)               |
> |--------------|----------------|-------|-------|-------|-------|-------|-------|--------------|--------------|-------------------|
> | Wine  | **0.496** | 0.493 | 0.252 | 0.491 | 0.225 | 0.472 | 0.479 | 0.463        | 0.458        | 0.453             |
> | Yeast | 0.676          | 0.678 | 0.626 | 0.681 | 0.563 | 0.683 | 0.684 | 0.687        | 0.687        | **0.692**    |
> | Ecoli | 0.877          | 0.875 | 0.840 | 0.866 | 0.817 | 0.845 | 0.856 | 0.862        | 0.860        | **0.886**    |
>
> [1] Elreedy, D., Atiya, A.F. and Kamalov, F., 2024. A theoretical distribution analysis of synthetic minority oversampling technique (SMOTE) for imbalanced learning. Machine Learning, 113(7), pp.4903-4923.
>
> [2]Elreedy, D. and Atiya, A.F., 2019. A comprehensive analysis of synthetic minority oversampling technique (SMOTE) for handling class imbalance. Information Sciences, 505, pp.32-64.
>
> [3] Han, H., Wang, W.Y. and Mao, B.H., 2005, August. Borderline-SMOTE: a new over-sampling method in imbalanced data sets learning. In International conference on intelligent computing (pp. 878-887). Berlin, Heidelberg: Springer Berlin Heidelberg.
>
> [4]He, H., Bai, Y., Garcia, E.A. and Li, S., 2008, June. ADASYN: Adaptive synthetic sampling approach for imbalanced learning. In 2008 IEEE international joint conference on neural networks (IEEE world congress on computational intelligence) (pp. 1322-1328). Ieee.

---

> > ### Comment · Reviewer_rG2v · 2024-11-27
> >
> > Thank you to the authors for their clarification, which has addressed most of my concerns. However, I have decided to maintain the score unchanged from the perspective of contribution and novelty. While the authors have successfully distinguished their work from related studies, the contribution remains primarily grounded in existing techniques, offering limited new insights or inspirations. Consequently, the work does not significantly advance the state of knowledge in the field.

---

### Official Review · Reviewer_aPEe · 2024-11-03

**Soundness:** 2
**Presentation:** 2
**Contribution:** 2
**Rating:** 3
**Confidence:** 4

**Summary:**

This paper makes a theretical analysis of the well-known SMOTE method for imbalance classification, and proposes two simplest variants.

**Strengths:**

This paper involves both theoretical analysis and method development

**Weaknesses:**

1) The theretical analysis is based on the assumption that $n$ is sufficiently large such that $K/n$ tends to zero, but this assumption is totally impractical in the setting of imbalanced learning where minority class has few examples.
2) The theretical analysis provides little information with respect to imbalanced learning, and it simply discusses the SMOTE method in the general learning setting. The result does not help understand the essence of imbalanced learning, and does not motivate how to solve the problem.
3) The connection between theoretical analysis and the method proposal is not well established.
4) The two proposed methods are not attractive. CV SMOTE is simply a hyperparameter tuning based on cross-validation, which is usually effective and commonly used technique as an engineering approach, without any theoretical or methodological contribution. MGS requires that there are at least $d+1$ minority-class examples, which may be impractical for high-dimensional data. In addition, even there are more than $d+1$ minority-class examples available, it has to assume that the covariance matrix is diagonal, which is also not the case in practice and thus over-generalized samples would be generated, which may do harm to the performance of imbalanced learning.
5) The experimental results are not significant.

**Questions:**

Please refer to the weaknesses 1, 2, 3 and 4.

More specifically,
1) Could you provide bounds or approximations that hold for small $n$, instead of huge $n$?
2) Could you analyze how SMOTE affects the decision boundary or classification margins in imbalanced settings?
3) Could you explain the reasons for designing these two variants based on the theoretical analysis in your paper and your responses to the questions 1) and 2)?
4) Have you considered any theoretical guarantees for CV SMOTE? Have you explored regularization techniques for MGS to handle high-dimensional data?

---

> ### Author Response · Authors · 2024-11-15
> **Response to reviewer aPEe**
>
> You share some concerns on the assumptions of Theorem 3.2 and Theorem 3.5 ($K$ and $n$ which tend to infinity). Any data point generated via SMOTE $Z_{K,n}$ is shown to converge in distribution to any original data point $X$, when $K/n$ tends to zero as $n$ tends to infinity (Theorem 3.2). Thus, SMOTE allows us to generate new data points, which are asymptotically distributed as the original samples. As many statistical analyses, we assume that the size of the minority class tends to infinity in order to study the behavior of SMOTE. Though unrealistic, this setting tends to mimic a data set composed of a high number of observations. Note that this does not contradict the imbalanced classification setting, as we can consider an overall data set whose size tend to infinity, with a fixed proportion of minority samples, therefore allowing the size of the minority class to tend to infinity.  Practically, there exist highly imbalanced data sets with a large value of $n$. For example, the CreditCard data set, included in our study, has an imbalance ratio of 0.2\% and $n=492$. Indeed, in the banking sector, it is common to have several millions of clients and among them only a few thousands make fraudulent transactions, leading to a high value of $n$ and highly imbalanced data.
> Besides, you ask if some of our theoretical results holds for a fixed $n$. Note that Corollary 3.6 is valid for any value of $n$. Besides, Theorem 3.5 and Theorem 3.7 are valid in a finite-sample setting, when $n$ is fixed but larger than a specific value (depending on $K$). Thus, our results are not only asymptotic, but leads to an understanding of SMOTE behavior in finite-sample settings.
>
> We agree that we provide an analysis of SMOTE properties, regardless of the majority class. Our analysis focuses on the ability of SMOTE to generate data according to the true distribution of the minority class, which is a first guarantee for using such a strategy in imbalance learning. We would like to emphasize that our work already leads to important conclusions on SMOTE: default value of $k=5$ should not be used by default, and the ability of SMOTE to generate samples with the correct distribution is damaged near the boundary of the support of the minority class. Including the  majority sample (and thus the imbalance setting) in our analysis cannot be done without taking into account the predictive algorithm (logistic regression, random forests...). In this case, our conclusion would depend on the predictive algorithm. We leave to future work to try to find theoretical settings in which one can establish properties of rebalancing strategies. Again, note that previous works on SMOTE simply establish the density of newly generated data points, or compute the expectation or the covariance matrix of the corresponding distribution.
>
> Through our theoretical study, we proved that asymptotically, the default value $K=5$ of SMOTE leads to a lack of diversity in the samples generated by SMOTE.
> Then we show that for some values of $K$ (such that $K=0.1n$), SMOTE observations are more diverse.
> We tried different asymptotics for $K$ (like $K = \sqrt{n}$, $K = \alpha n$ for different values of $\alpha$) but we could not find a universal best choice. Thus, we introduce a grid optimization procedure to automatically choose the best value for $K$. We would like to remark that SMOTE is a widely used algorithm with its default hyperparameter value $K=5$ and to the best of our knowledge there is no paper recommending tuning $K$.  MGS is introduced in order to have a process that do not generate new samples only in a vector line. This would lead to more diverse synthetic samples again and a better reconstruction of the minority class distribution.
> It is surprising that such small modifications of existing algorithms lead to two new procedures with improved performances.
> Our goal is by no mean to say that these two methods improve drastically over existing ones. Indeed, for most data sets, None method is the best one. Thus, one of our major contribution is to establish that None is among the best strategy and that, in the remaining cases, our two proposals are competitive.

---

> > ### Author Response · Authors · 2024-11-15
> > **Response to reviewer aPEe (part 2)**
> >
> > We would like to share that our introduced strategy MGS does not need the covariance matrix to be diagonal. In Algorithm 2 we detail the estimation of the empirical covariance matrix inside MGS procedure. Furthermore, we only set $K=d+1$ in MGS procedure in order to possibly obtain a full-rank empirical covariance matrix. In sparse high-dimension, where the true signal is assumed to belong to a variety of reasonable dimension $s \ll d$, we would prefer to choose $K \simeq s$. This can be done by implementing a cross-validation procedure. Note that we tested different strategies for estimating the covariance matrix and for regularization. More specifically, we weighted the covariance matrix by the inverse of the distance of the synthetic samples to their nearest neighbor. We also consider regularization of the covariance matrix via shrinkage (Ledoit-Wolf[1] and OAS[2]). We only presented MGS in the paper, as the above variants are more complex and did not outperform MGS.
> >
> > [1] O. Ledoit and M. Wolf, “A Well-Conditioned Estimator for Large-Dimensional Covariance Matrices”, Journal of Multivariate Analysis, Volume 88, Issue 2, February 2004, pages 365-411.
> >
> > [2] Shrinkage algorithms for MMSE covariance estimation., Chen, Y., Wiesel, A., Eldar, Y. C., and Hero, A. O. IEEE Transactions on Signal Processing, 58(10), 5016-5029, 2010.

---

> > ### Comment · Reviewer_aPEe · 2024-11-24
> >
> > As I stated in the original comment, the nature of imbalanced learning should NOT be violated for any related work to be useful and valuable. However, the analysis provided in this paper has completely abandoned the nature or constraint of imbalanced learning.
> >
> > In addition, most of my concerns have not got a satisfactory clarification, and thus I keep the score unchanged.

---

### Meta-Review · Area_Chair_w7Z5 · 2024-12-21

**Metareview:**

The Authors discuss the problem of imbalance classification. Their focus is on analyzing one of the most popular oversampling strategies, known as SMOTE. They prove, for example, that SMOTE under the default parameters tends to copy the original minority samples asymptotically. Based on this analysis they propose two new variants. The results from a large empirical studies show that rebalancing is not always needed.

The Reviewers underline as strengths the theoretical results, good presentation, and extensive empirical studies. As weaknesses, they point out limited novelty of the proposed variants, limited theoretical insights into SMOTE variants used in the empirical studies (including the introduced ones), and several remarks concerning the theoretical analysis of the standard SMOTE, which were mostly addressed by the Authors in the rebuttal.

As an AC I also posted two doubts concerning limited discussion on related work concerning (i) optimization of the complex performance metrics and (ii) relation between oversampling and regularization. It also seems that a paper is a bit contradicting and not complete. The Authors should:
- Either, study SMOTE, deliver theoretical insights, introduce new extensions, analyze them theoretically and validate them in empirical studies,
- Or, send a clear message that rebalancing is not really needed with theoretical and empirical evidence (with potential exceptions why rebalancing could be considered, e.g., to decrease computational costs, for some specific performance metrics such as balanced accuracy, or as a regularization technique).

The paper in the current form is not ready to be published at a top ML conference. I encourage the Authors to extend their theoretical studies, properly incorporating the existing results, and to improve clarity of the message they want to communicate.

**Additional Comments On Reviewer Discussion:**

The Authors succeeded to clarify many doubts of the Reviewers, but the limited practical contribution and novelty seem to be the main unresolved issues. The Reviewer with the highest score did not champion the paper.

The Authors' response to my questions has confirmed my critical remarks. The paper needs to relate properly to existing results and send a clear message to readers.

---

### Decision · Program_Chairs · 2025-01-22

Reject